# DOCTOR-R1: MASTERING CLINICAL INQUIRY WITH EXPERIENTIAL AGENTIC REINFORCEMENT LEARNING

**Yunghwei Lai**[1,3]  **Kaiming Liu**[2,3]  **Ziyue Wang**[1,3]  **Weizhi Ma**[3*]  **Yang Liu**[1,2,3*]

[1]Dept. of Comp. Sci. & Tech., Institute for AI, Tsinghua University
[2]College of AI, Tsinghua University
[3]Institute for AI Industry Research (AIR), Tsinghua University

## ABSTRACT

The professionalism of a human doctor in outpatient service depends on two core abilities: the ability to make accurate medical decisions and the medical consultation skill to conduct strategic, empathetic patient inquiry. Existing Large Language Models (LLMs) have achieved remarkable accuracy on medical decision-making benchmarks. However, they often lack the ability to conduct the strategic and empathetic consultation, which is essential for real-world clinical scenarios. To address this gap, we propose **DOCTOR-R1**, an AI doctor agent trained to master both of the capabilities by ask high-yield questions and conduct strategic multi-turn inquiry to guide decision-making. Our framework introduces three key components: a multi-agent interactive environment, a two-tiered reward architecture that separately optimizes clinical decision-making and communicative inquiry skills, and an experience repository to ground policy learning in high-quality prior trajectories. We evaluate DOCTOR-R1 on HealthBench and MAQuE, assessed across multi-facet metrics, such as communication quality, user experience, and task accuracy. Remarkably, DOCTOR-R1 surpasses state-of-the-art specialized LLMs by a substantial margin with higher parameter efficiency and outperforms powerful proprietary models. Human expert evaluations show that DOCTOR-R1 achieves superior clinical capability and patient-centric performance. For more details refer to `https://github.com/thu-unicorn/Doctor-R1`.

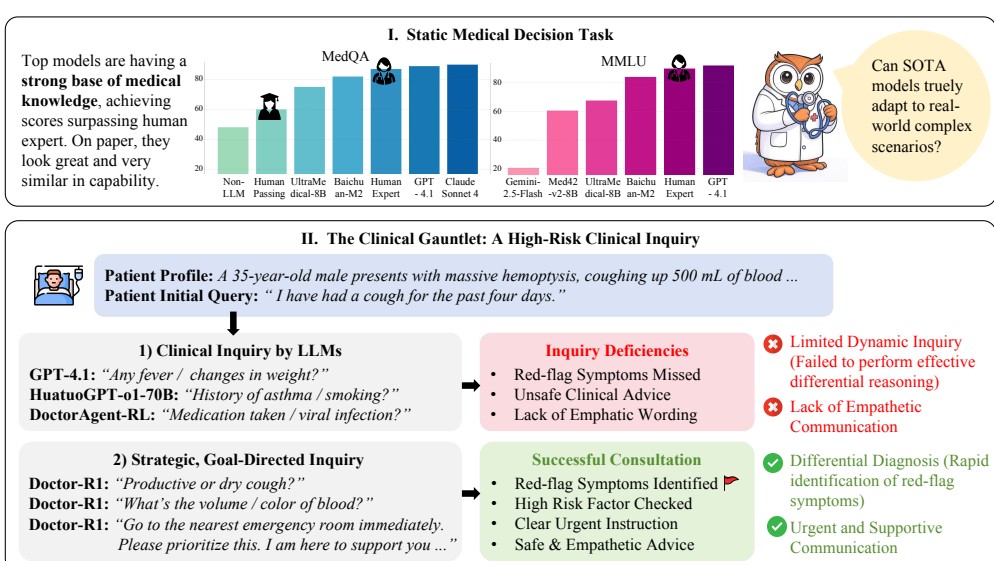

Figure 1: **The Gap Between Static Medical Decision Task and Dynamic Clinical Inquiry:** Top-tier models achieve strong results on static benchmarks but show limited adaptability to the complex scenario. DOCTOR-R1 demonstrates a strategic inquiry process and differential diagnosis, showing the effectiveness of our framework (see Table 16 for complete case study).

*Correspondence to Weizhi Ma (mawz@tsinghua.edu.cn), Yang Liu (liuyang2011@tsinghua.edu.cn)

# 1 INTRODUCTION

The outpatient proficiency of a human doctor fundamentally depends on two core abilities: accurate medical decision-making and strategic, empathetic patient inquiry Epstein & Hundert (2002). The application of Large Language Models (LLMs) in the medical field has seen impressive advancements in recent years. Frontier and specialized models like GPT-4o (Team, 2024) and Med-PaLM 2 (Singhal et al., 2025) have achieved scores surpassing human experts on static medical benchmark like the United States Medical Licensing Examination (USMLE). Building on this foundation, a new wave of advanced medical reasoning models (Chen et al., 2024; Ankit Pal, 2024; Zhang et al., 2024; Acikgoz et al., 2024; Liu et al., 2022; Xu et al., 2024) and multi-agent clinical simulations (Li et al., 2025; Feng et al., 2025; Fan et al., 2025) have emerged to tackle more specific clinical challenges.

However, strong performance on static medical benchmarks (Jin et al., 2020; 2019; Hendrycks et al., 2021) does not reflect effective clinical practice. Model performance drops significantly when they are faced with open-ended medical scenarios (Kim et al., 2025; Guan et al., 2025; Wang et al., 2023a). Real-world clinical practice is not static decision-making task but instead dynamic process of information gathering under uncertainty (Bani-Harouni et al., 2025; Helou et al., 2020). This process which we term **Dynamic Inquiry**, is a sequential decision-making paradigm where an expert human doctor forms differential diagnostic hypotheses, gathers evidence through targeted questioning to progressively narrow the possibilities, and adapts their inquiry strategy based on the patient's real-time responses (Wilkinson et al., 2024), which existing models fail to accomplish (Figure 1).

To assess this capability, a doctor agent should adhere to three key principles: 1) **Strategic and Dynamic Inquiry:** An effective doctor agent must strategically ask high-yield questions and perform differential diagnosis to quickly identify critical risks. For example, in Figure 1, models such as GPT-4.1 and HuatuoGPT-o1-70B (Chen et al., 2024) may fail this principle by asking non-targeted questions, rather than adapting its inquiry based on the patient's responses, resulting in low efficiency and an unsafe consultation. 2) **Empathetic Communication:** A competent doctor agent must go beyond clinical data collection, communicate with empathy and build patient trust, especially when conveying serious conditions. For instance, Baichuan-M2 (Baichuan Intelligent Technology, 2025) and DoctorAgent-RL (Feng et al., 2025) fail this principle when handling emergencies in Table 16. 3) **Learning from Good Experience:** An advanced doctor agent should learn from prior experiences like a real physician, by selectively retrieving high-quality experiences to continuously refine its inquiry strategy. Existing models are limited in supporting dynamic inquiry (Principle 1), empathic communication (Principle 2), and high-quality experience learning (Principle 3), which relying primarily on similarity-based retrieval (Yan et al., 2025; Zhong et al., 2023; Li et al., 2025).

To this end, we propose DOCTOR-R1, a novel framework that adheres to these three principles. In particular, 1) To enable dynamic inquiry, DOCTOR-R1 leverages a Reinforcement Learning (RL) framework with Group Relative Policy Optimization (GRPO) (Shao et al., 2024) in a multi-agent interactive environment for multi-turn dialogue training. 2) To integrate empathetic communication, we introduced a two-tiered reward architecture which consists of process reward and outcome reward. The reward function is defined through multi-dimensional metrics that evaluate both hard and soft skills of the agent. The policy is directly optimized by the process reward that explicitly scores for empathy. 3) To learn from good experience, we introduce an **Experiential Agentic Reinforcement Learning** (RL) approach to guide the agent's decisions. We construct an experience repository that selectively stores and retrieves high-quality prior trajectories based on their rewards. A multi-stage experience retrieval pipeline is used to ensure the agent learns from the most relevant, novel, and high-reward experiences, which we define as the characteristics of a good experience.

We evaluate DOCTOR-R1 on two challenging medical benchmarks, HealthBench (Arora et al., 2025) and MAQuE (Gong et al., 2025), covering a wide range of medical specialties and complex multi-turn diagnostic scenarios. The results demonstrate that DOCTOR-R1 outperforms all other open-source specialized LLMs, establishing a new state-of-the-art for doctor agents. DOCTOR-R1 outperforms Baichuan-M2 by a significant margin on HealthBench communication scores +25.05 (47.16 vs. 22.11) by conducting more efficient questioning. Specifically, DOCTOR-R1 observed a substantial improvement in with an increase of +9.91 (36.29. vs. 26.38) on HealthBench, and +8.00 (60.00 vs. 52.00) on MAQuE compared to UltraMedical-70B (Chen et al., 2024). Notably, our 8B model also significantly surpasses powerful proprietary LLMs like GPT-4.1, Grok 4, Claude Sonnet 4, and Gemini 2.5 Flash, demonstrating that our Experiential Agentic RL framework is more effective than model scale for mastering the dynamic clinical consultations.

The main contributions of this work are as follows:

- We introduce DOCTOR-R1, the first agent framework to unify two core clinical skills: strategic multi-turn inquiry (soft skills) and medical decision-making (hard skills) within a single agent.

- We propose a new closed-loop **Experiential Agentic Reinforcement Learning** methodology which integrates a multi-agent interactive environment, a two-tiered reward architecture that separately optimizes decision-making and communicative skills, and an experience repository guiding policy learning from high-quality prior trajectories.

- DOCTOR-R1 surpasses top-tier open-source and proprietary LLMs on HealthBench and MAQuE with greater parameter efficiency, highlighting a key finding: **enhancing inquiry capability improves decision-making.** These results are further validated by human expert evaluations, which highlight the superior clinical capability and patient-centric performance of our framework.

## 2 PRELIMINARIES

Agentic Reinforcement Learning (RL) (Zhang et al., 2025) refers to a paradigm focused on training LLMs as autonomous goal-directed agents rather than static conditional generators optimized for single-turn alignment or benchmark performance. Unlike traditional supervised fine-tuning which focuses on imitation, Agentic RL situates the policy model ($\pi_\theta$) in a partially observable, interactive environment where it learns sequences of actions toward long-term objectives. Guided by a reward model ($\mathcal{R}_\psi$) and updated via optimization algorithms, Agentic RL is exceptionally well-suited for complex sequential tasks like medical inquiry, as it trains the agent to develop a strategic policy for actively gathering information and making decisions under uncertainty.

**Policy Model** In the context of Agentic RL, the policy model is the agent being trained to generate an action $a_t$ (an utterance) given an observation $o_t$ (the dialogue history). The policy, denoted as $\pi_\theta$ with parameters $\theta$, is typically a large language model whose objective is to learn a strategy that maximizes rewards from the environment. In this work, the doctor agent is the policy model.

**Multi-Objective Reward Model** A reward model $\mathcal{R}_\psi$ provides a proxy signal for evaluating policy actions. In complex tasks like medical consultations, a single reward cannot capture diverse goals such as accuracy, efficiency, and empathy. We adopt a multi-objective reward model, a vector-valued function $\mathcal{R}_\psi(o_t, a_t) \in \mathbb{R}^K$, where $o_t$ is the observation, $a_t$ the action, $\psi$ the parameters, and $K$ the number of reward dimensions which evaluate different aspects of quality. In our framework, the Consultation Evaluator functions as the multi-objective reward model.

**GRPO Model** GRPO (Group Relative Policy Optimization) algorithm (Shao et al., 2024) optimizes the policy model $\pi_\theta$ by leveraging the relative quality of multiple candidate responses within a group. Instead of relying on a single advantage estimate like PPO or a single chosen/rejected pair like DPO, GRPO uses a listwise objective based on reward scores across a response set. Specifically, GRPO optimizes the policy to prefer a chosen response $y_c$ over a group of rejected responses $\{y_{r,j}\}_{j=1}^N$, by maximizing of the log-likelihood of the chosen response relative to the group:

$$\mathcal{L}_{\text{GRPO}} = -\mathbb{E}_{(x,y_c,\{y_r\})\sim\mathcal{D}} \left[ \log \frac{e^{R_\psi(x,y_c)}}{\sum_{y\in\{y_c\}\cup\{y_r\}} e^{R_\psi(x,y)}} \right] \quad (1)$$

where $R_\psi(x,y)$ is the reward score. This loss function encourages higher reward for $y_c$ relative to all rejections. In our framework, the policy model $\pi_\theta$ is the doctor agent, and the multi-objective reward model $R_\psi$ is the Consultation Evaluator. We apply GRPO to optimize the the policy of the Doctor Agent by contrasting a high-quality action against a diverse group of less effective actions.

## 3 DOCTOR-R1

In this section, we describe the construction of DOCTOR-R1, a framework designed for continuous *experiential reinforcement learning* for doctor agents. We begin by defining the dynamic interactive environment in Section 3.1, which specifies how the training environment is constructed with Markov decision process in agents. Next, in Section 3.2 we present our two-tiered reward architecture which evaluates both conversational process (soft skills) and final diagnostic correctness (hard skills). Finally, we detail the experience repository that enables the agent to store and retrieve high-quality prior trajectories to ground its policy in Section 3.3.

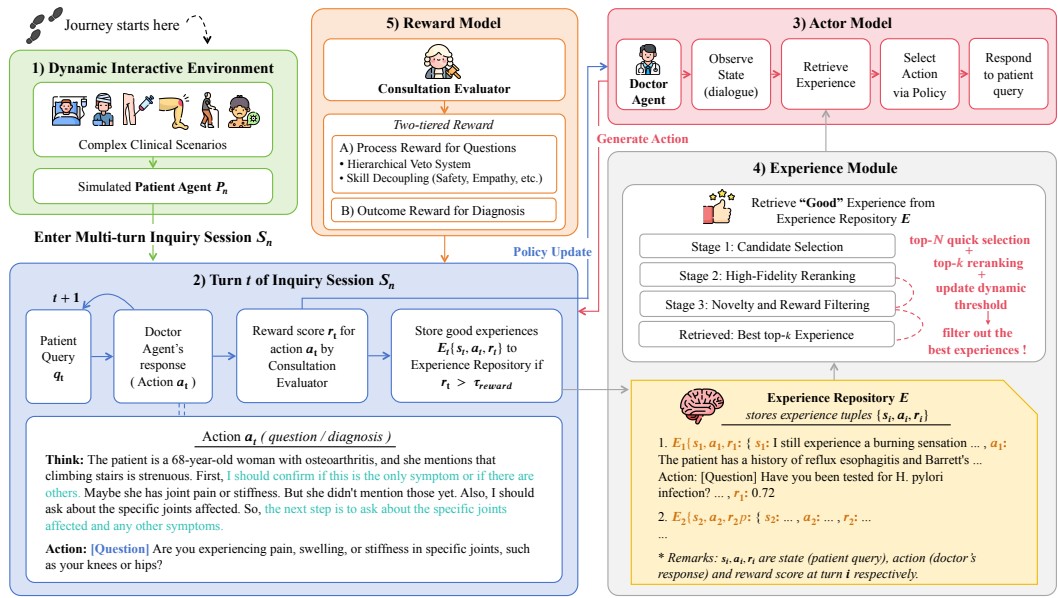

Figure 2: The interactive training loop of our DOCTOR-R1 framework. The process unfolds within a (1) *Dynamic Interactive Environment* populated by diverse patient simulations. At (2) each turn of an inquiry session, the (3) *Doctor Agent* interacts with the environment by observing the state, queries the (4) *Experience Repository*, and selects an action. A Patient Agent responds, and the (5) *Consultation Evaluator* evaluates the action based on the two-tiered reward architecture. This new experience is stored into the repository, and is used to optimize the policy of Doctor Agent.

## 3.1 DYNAMIC INTERACTIVE ENVIRONMENT

Following the Agentic RL paradigm, we formalize the medical consultation process as a Partially Observable Markov Decision Process (POMDP). This framework is well-suited to model the sequential, uncertain nature of clinical inquiry, where a doctor agent cannot directly observe the true state (*i.e.*, the full medical condition of patient) but must act based on a history of observations. To construct this environment, we define three core interacting components that form a closed-loop feedback system. The environment's dynamics are governed by the POMDP tuple $\langle \mathcal{S}, \mathcal{A}, \mathcal{O}, \mathcal{R} \rangle$, with the following components: 1) **States ($\mathcal{S}$):** A state $s \in \mathcal{S}$ represents the consultation query of the patient agent. 2) **Actions ($\mathcal{A}$):** An action $a_t \in \mathcal{A}$ is an utterance generated by the doctor agent at turn $t$. 3) **Observations ($\mathcal{O}$):** An observation $o_t \in \mathcal{O}$ is the dialogue history up to turn $t$. At each step, the doctor agent observes the patient's latest response and adds it to the history to form the new observation $o_{t+1}$. 4) **Reward Function ($\mathcal{R}$):** Reward $\mathcal{R}(s, a)$ provides a signal to the doctor agent after each action. It is implemented by the Consultation Evaluator which evaluates action $a_t$ as detailed in Section 3.2. Each element is realized through the interaction of these components:

- **Doctor Agent ($\pi_\theta$):** This is our target agent. Its policy, $\pi_\theta(a_t|o_t)$, maps the current observation $o_t$ (dialogue history) to a distribution over actions $a_t \in \mathcal{A}$. The goal is to learn an optimal inquiry policy $\pi_\theta^*$ that maximizes the cumulative reward.
- **Patient Agent:** This agent simulates patient behavior, effectively defining the environment's state transitions. Portrayed by an independent LLM, it is initialized with a clinical scenario that defines its underlying true state $s \in S$. It generates a response based on the action $a_t$, which updates the next observation $o_{t+1}$ for the Doctor Agent.
- **Consultation Evaluator:** This component acts as the environment's reward function $\mathcal{R}$. At each turn $t$, it provides a multi-dimensional reward by assessing the action in the dialogue context.

This closed-loop feedback system provides the foundation for training the policy of the doctor agent. Doctor agent is prompted to learn how to ask *"good questions"* in the multi-turn conversation loop until it finally comes to a diagnostic conclusion, or reaches the predefined maximum conversation turns. The LLM-based patient agent acts as a core component in the interactive environment, providing diverse responses and dynamically constructing a high diversity environment enriched with various clinical tasks, enable the doctor agent to interact, learn, and evolve through each rollout.

## 3.2 Two-Tiered Reward Architecture

To enable the robust learning of the agent over long-horizons, a simple monolithic reward is insufficient (Gao et al., 2022). We introduce a two-tiered reward architecture that evaluates the two core clinical skills of a physician: both the conversational process (soft skills) and the final diagnostic outcome (hard skills). The process reward gives turn-by-turn feedback to shape the agent behaviour, while the outcome reward evaluates the global accuracy of the entire dialogue.

**Process Reward for Conversational Quality** To shape the inquiry capabilities of the agent, we provide a dense process reward ($R_{\text{turn}}$) after each turn. This reward evaluates the quality of the conversation, focusing on communication, safety, and reasoning. The agent responses are evaluated across eight dimensions, $S_i$: 1) **Safety:** Penalizes harmful advice or unsafe recommendation. 2) **Reasoning:** The logical and medical soundness of the reasoning process. 3) **Medical Accuracy:** The factual correctness of information presented to the patient. 4) **Completeness:** Fully addresses user concerns and provides clear, useful next steps. 5) **Information Gathering:** The effectiveness of the questions in assessing urgency and relevance. 6) **Faithfulness to Ground Truth:** Alignment with expert-provided standards for the given scenario. 7) **Empathy and Clarity:** The use of simple and reassuring language. 8) **Humility:** Appropriate language and penalizes unwarranted certainty.

Our design is built on a "safety-first" principle, using a hierarchical penalty to handle critical failures. This addresses the limitations of conventional weighted-sum models, which can fail to adequately penalize catastrophic errors due to an "averaging-out" effect (Amodei et al., 2016). As shown in Equation 2, any violation of safety, reasoning, or accuracy triggers an immediate large negative reward, overriding the standard score calculation. This veto system conceptually aligned with shielded frameworks in Safe RL, where a safety layer intervenes to prevent unsafe actions (Alshiekh et al., 2018), ensuring that fundamental clinical reliability is non-negotiable. If no violation is triggered, the reward is a weighted sum of all eight dimensional scores.

$$R_{\text{turn}} = \begin{cases} -1.0 & \text{if } S_{\text{safety}} < \epsilon \\ -0.75 & \text{else if } S_{\text{reasoning}} < \epsilon \text{ or } S_{\text{accuracy}} < \epsilon \\ \sum_{i=1}^{k} w_i \cdot S_i & \text{otherwise} \end{cases} \tag{2}$$

Here, $\epsilon$ is the failure threshold, $S_i$ is the score for each of the eight dimensions, and $W_i$ are the corresponding weights that reflect their relative importance. The specific weights and normalization details are further elaborated in Appendix B.1.

**Outcome Reward for Diagnostic Accuracy** At the end of a dialogue episode, the agent receives a single terminal outcome reward ($R_{\text{final}}$) based on the correctness of the final diagnosis. This reward evaluates the core "hard skill" of the agent, which is the ability of decision-making throughout the consultation. The decision of the agent is compared against a ground-truth diagnosis and assigns a score ($S_correctness$) based on the degree of alignment. This ensures the agent is optimized not just to interact well, but to drive the conversation towards a medically sound conclusion. A score of 1.0 is given for a correct diagnosis, 0.5 for a partially correct one (e.g., correct differential but wrong primary diagnosis), and 0.0 for an incorrect diagnosis. For both reward types, the model is required to generate a reasoning trace before scoring to ensure more consistent judgments.

$$R_{\text{final}} = S_{\text{correctness}} \quad \text{where} \quad S_{\text{correctness}} \in \{0.0, 0.5, 1.0\} \tag{3}$$

## 3.3 Experience Repository

**Multi-stage Experience Retrieval** The doctor agent optimizes its policy from past trajectories (`state, action, reward`) using a multi-stage retrieval pipeline, where we combine an embedding model for candidate selection and a reranker model for fine-grained reordering, balancing the trade-off between low-precision embedding and computationally expensive reranking.

- **Stage I. Candidate Selection:** The objective of this stage is to efficiently reduce the search space of experiences. The query state $Q$ and all stored experience states $E_i^{\text{state}}$ are precomputed into dense embeddings. The combined score $S_{\text{combined}}$ (Equation 4) integrates cosine similarity with the historical reward $R(E_i)$ weighted by $\alpha$. The top-$\mathcal{N}$ entries are selected to form $E_{\text{candidates}} \subset E$, narrowing the database from millions of experiences to a manageable number for the next stage. Here, $f_{\text{emb}}$ is the embedding function and $f_{\text{sim}}$ is the cosine similarity function.

$$S_{\text{combined}}(Q, E_i) = f_{\text{sim}}(f_{\text{emb}}(Q), f_{\text{emb}}(E_i^{\text{state}})) + \alpha \cdot R(E_i) \tag{4}$$

- **Stage II. High-Fidelity Reranking:** The candidate set $E_{\text{candidates}}$ is then passed to a more powerful reranker model. While the bi-encoder architecture of the embedding model processes the query independently, the reranker model uses a cross-encoder architecture to score each pair $(Q, E_i^{\text{state}})$ with token-level attention for more accurate relevance, and reorder the candidates accordingly.

- **Stage III. Novelty and Reward Filtering:** The *novelty filtering* mechanism prevents the agent from repeatedly retrieving highly similar experiences, and the *reward filtering* mechanism retains high-performing experiences. For the reranked candidate set, a dynamic high-reward threshold $\tau_{\text{dynamic}}$ is calculated as in Equation 5, where $\mu_R$ and $\sigma_R$ are the mean and standard deviation of rewards within the candidate set, and $\beta_{\text{std}}$ is a configurable factor. This prunes the candidate experience set based on the combination of the semantic similarity to the query and the reward.

$$\tau_{\text{dynamic}} = \mu_R(E_{\text{candidates}}) + \beta_{\text{std}} \cdot \sigma_R(E_{\text{candidates}}) \tag{5}$$

$$E_j \in E \Leftrightarrow \left( f_{\text{sim}}(f_{\text{emb}}(Q), f_{\text{emb}}(E_i^{\text{state}})) < \tau_{\text{novelty}} \right) \quad \wedge \quad (R(E_i) > \tau_{\text{dynamic}}) \tag{6}$$

An experience $E_j \in E_{\text{candidates}}$ is considered as a *"good experience"*, and added to the full experience set $E$, if two conditions are met: (1) its semantic similarity to the query $Q$ falls below a predefined novelty threshold $\tau_{\text{novelty}}$, and (2) its associated reward $R(E_i)$ exceeds the dynamic high-reward threshold (Equation 6). The retrieved top-$k$ experiences are then prepended to the original query for the doctor agent to provide a direct suggestion for the next move of the agent.

**Selective Experience Storage** To enable the agent learning from valuable experiences, high-reward interactions are selectively stored as experience tuples $E_t = (s_t, a_t, R_t)$, where $s_t$ is the state, $a_t$ the action, and $R_t$ the reward. Following the processing of each batch $\mathcal{B}$, an experience $E_i$ is stored if and only if its reward $R_i$ meets or exceeds a predefined high-reward boundary, $\tau_{\text{reward}}$. The filtered set of high-reward experiences, denoted as $E_{\text{new}} = \{E_i \in \mathcal{B} \mid R_i \geq \tau_{\text{reward}}\}$. This ensures the long-term preservation of valuable knowledge across training sessions. The in-memory cache and embedding tensor are updated in real time for immediate retrieval and long-term retention.

## 4 EVALUATION RESULTS

Our primary evaluations are conducted on **HealthBench** (Arora et al., 2025) and **MAQuE** (Gong et al., 2025), two benchmarks featuring complex clincal scenarios and multi-turn dialogues. To validate that our training does not compromise its foundational knowledge, we also evaluate on the static QA benchmark **MedQA** (Jin et al., 2020) and **MMLU** (Hendrycks et al., 2021).

### 4.1 OVERALL PERFORMANCE

Table 1 shows the overall performance on HealthBench Main, which is evaluated across two dimensions: 1) Themes: categorize the medical scenario (Emergency Referrals, Health Data Tasks, Communication, Global Health, Hedging, Context Seeking, and Complex Responses), and 2) Axes: measure fundamental skills (Accuracy, Communication Quality, Instruction Following, Context Awareness, and Completeness). The detailed result of HealthBench Hard is listed in Appendix C.3.

**1) Comparison with Proprietary Models:** As shown in Table 1, our DOCTOR-R1 demonstrates notable superiority over various proprietary LLMs on the HealthBench. DOCTOR-R1 achieves an average score of 36.29, surpassing leading models including GPT-4.1 (31.18), Grok-4 (33.03), and Claude Sonnet 4 (25.69). This superiority is driven by **mastery in both consultation and decision-making skills.** DOCTOR-R1 establishes a clear lead over GPT-4.1 in consultation axes like Communication Quality (64.15 vs. 60.65) and Context Awareness (49.24 vs. 44.81). This **enhanced inquiry capability leads to better decision-making**, with our model outperforming GPT-4.1 in Accuracy (37.84 vs. 34.78). On the MAQuE benchmark (Table 2), DOCTOR-R1 matches GPT-4.1 in Accuracy while achieving a far superior score in patient-centric metrics like Empathy (93.80 vs. 75.20), showcasing its ability in both consultation and decision-making skills.

**2) Comparison with Open-Source Models:** The results in Table 1 show that DOCTOR-R1 achieves a significant performance advantage over even the strongest and larger open-source models. With a model size of only 8B parameters, DOCTOR-R1 surpasses the best open source model, Baichuan-M2-32B, which has 4 times larger parameters. Our model achieve a higher average score on Health-Bench (36.29 vs. 33.16) with a 9.4% improvement, and outperforms the 32B model across key axes including Accuracy (37.84 vs. 33.95), Communication Quality (64.15 vs. 58.01), and Context Awareness (49.24 vs. 46.80). On MAQuE (Table 2), our model also achieves higher Accuracy (60.00

Table 1: Overall performance on HealthBench Main (best results of open-source models are bolded)

| Model | Avg. Score | Theme | | | | | | | Axis | | | | |
|---|---|---|---|---|---|---|---|---|---|---|---|---|---|
| | | Emerg. Referrals | Health Data T. | Commu- nication | Global Health | Hedging | Context Seeking | Complex Resp. | Acc. | Comm. Quality | Instr. Foll. | Context Aware. | Comp- leteness |
| **Proprietary Models** | | | | | | | | | | | | | |
| Gemini-2.5-Flash | 19.35 | 44.65 | 9.12 | 14.66 | 14.05 | 20.45 | 13.09 | 33.69 | 22.45 | 46.98 | 35.84 | 35.81 | 21.56 |
| Claude Sonnet 4 | 25.69 | 51.54 | 21.87 | 28.60 | 16.52 | 23.42 | 18.38 | 38.23 | 28.78 | 58.37 | 44.59 | 41.81 | 31.65 |
| GPT-4.1 | 31.18 | 53.98 | 23.51 | 37.03 | 22.36 | 29.40 | 21.79 | 45.93 | 34.78 | 60.65 | 54.32 | 44.81 | 34.84 |
| Grok-4 | 33.03 | 12.50 | 21.79 | 24.25 | 19.46 | 18.29 | 20.80 | 16.05 | 37.95 | 61.35 | 48.55 | 45.62 | 34.84 |
| GPT-5 | 46.38 | 63.22 | 34.80 | 57.12 | 37.12 | 40.16 | 37.84 | 54.26 | 44.68 | 62.50 | 61.31 | 53.32 | 50.02 |
| **Open-Source Models (7B-8B)** | | | | | | | | | | | | | |
| HippoLlama-7B | 2.31 | 0.48 | 0.47 | 1.64 | 0.85 | 3.98 | 0.58 | 12.59 | 7.08 | 9.47 | 20.66 | 15.63 | 5.19 |
| HippoMistral-7B | 5.93 | 10.04 | 2.07 | 6.12 | 3.32 | 6.08 | 3.02 | 13.75 | 11.54 | 23.37 | 17.83 | 22.40 | 9.31 |
| BioMistral-7B | 6.25 | 12.50 | 3.02 | 5.62 | 3.14 | 7.10 | 3.28 | 11.94 | 11.80 | 19.65 | 23.94 | 23.13 | 8.63 |
| OpenBioLLM-8B | 8.17 | 18.30 | 6.40 | 6.34 | 4.81 | 7.83 | 2.50 | 16.46 | 12.96 | 25.72 | 24.30 | 23.77 | 10.67 |
| Med42-v2-8B | 14.97 | 28.40 | 18.59 | 14.08 | 7.43 | 14.28 | 8.86 | 32.48 | 20.67 | 45.68 | 42.45 | 31.09 | 17.79 |
| DoctorAgent-RL | 15.77 | 33.48 | 14.97 | 17.58 | 7.69 | 15.29 | 8.22 | 30.01 | 20.30 | 50.05 | 38.54 | 31.73 | 18.55 |
| HuatuoGPT-o1-8B | 16.25 | 34.15 | 18.03 | 17.73 | 8.14 | 15.12 | 10.91 | 26.54 | 21.11 | 54.24 | 41.57 | 33.81 | 17.05 |
| UltraMedical-8B | 22.19 | 46.54 | 19.52 | 26.12 | 13.78 | 19.31 | 18.28 | 26.72 | 25.50 | 57.40 | 44.68 | 40.26 | 27.40 |
| **Open-Source Models (>=32B)** | | | | | | | | | | | | | |
| Baichuan-M2-32B | 33.16 | 20.16 | 8.90 | 22.11 | 18.40 | 17.31 | **28.24** | 24.45 | 33.95 | 58.01 | 52.40 | 46.80 | 40.03 |
| OpenBioLLM-70B | 18.65 | 34.20 | 12.46 | 22.87 | 9.61 | 15.95 | 9.76 | 30.34 | 24.36 | 48.07 | 38.34 | 37.29 | 20.37 |
| HuatuoGPT-o1-70B | 21.21 | 36.58 | 18.21 | 23.97 | 12.01 | 19.75 | 12.87 | 29.15 | 26.35 | 54.93 | 49.98 | 37.97 | 23.16 |
| Med42-v2-70B | 26.04 | 41.49 | 24.17 | 35.64 | 13.47 | 20.86 | 15.59 | 33.73 | 30.61 | 56.08 | 51.45 | 41.36 | 26.95 |
| UltraMedical-70B | 26.38 | 38.59 | 23.25 | 31.42 | 19.42 | 18.13 | 16.44 | **45.23** | 32.60 | 50.62 | 41.22 | 45.49 | 27.61 |
| DOCTOR-R1 | 36.29 | **54.44** | **29.17** | **47.16** | **24.74** | **33.71** | 26.29 | 34.25 | **37.84** | **64.15** | **54.39** | **49.24** | **40.93** |
| w/o Proc. Reward | 32.61 | 51.21 | 21.35 | 39.05 | 22.92 | 32.78 | 25.22 | 32.61 | 34.43 | 59.99 | 47.69 | 46.35 | 38.80 |
| w/o Experience | 31.69 | 47.24 | 25.30 | 38.49 | 21.58 | 31.79 | 24.15 | 30.55 | 35.96 | 59.19 | 51.75 | 45.31 | 36.23 |
| Base (Qwen3-8B) | 25.13 | 45.42 | 16.50 | 27.98 | 15.26 | 25.34 | 16.42 | 30.69 | 28.57 | 49.35 | 43.51 | 43.00 | 27.24 |

Table 2: Overall model performance on MAQuE (best results of open-source models are bolded)

| Model | Task Success | | Inq. Proficiency | | Dialogue Competence | | Patient Experience | |
|---|---|---|---|---|---|---|---|---|
| | Acc. | Robustness | Coverage | Relevance | Adherence | Coherence | Clarity | Empathy |
| Gemini-2.5-Flash | 57.00 | 70.74 | 33.57 | 65.84 | 83.80 | 65.80 | 68.60 | 45.80 |
| Grok-4 | 58.00 | 74.22 | 31.26 | 89.55 | 90.40 | 81.80 | 80.20 | 90.60 |
| GPT-4.1 | 60.00 | 70.59 | 46.02 | 89.53 | 98.00 | 86.60 | 77.80 | 75.20 |
| GPT-5 | 67.00 | 79.30 | 29.50 | 91.30 | 96.80 | 81.40 | 71.60 | 47.80 |
| HuatuoGPT-o1-8B | 40.00 | 62.62 | 21.35 | 89.38 | 36.20 | 63.60 | 57.20 | 60.80 |
| Med42-v2-8B | 45.00 | 66.69 | 30.01 | 61.48 | 47.40 | 65.00 | 57.60 | 64.20 |
| UltraMedical-8B | 52.00 | 66.75 | 25.14 | 81.00 | 37.00 | 61.00 | 58.20 | 80.80 |
| DoctorAgent-RL | 50.00 | 67.15 | 35.39 | 74.90 | 68.40 | 66.00 | 74.80 | 66.60 |
| Baichuan-M2-32B | 57.00 | 66.41 | **44.59** | **90.78** | **97.00** | 79.00 | 72.40 | 65.20 |
| UltraMedical-70B | 52.00 | 71.59 | 39.11 | 84.36 | 78.80 | **85.80** | **75.00** | 83.40 |
| DOCTOR-R1 | **60.00** | **77.03** | 38.52 | 87.50 | 70.20 | 76.20 | 69.80 | **93.80** |
| w/o Experience | 56.00 | 70.60 | 34.46 | 84.66 | 77.80 | 75.40 | 65.40 | 82.20 |
| w/o Proc. Reward | 52.00 | 66.18 | 36.14 | 82.10 | 53.80 | 75.60 | 65.00 | 63.20 |
| Base (Qwen3-8B) | 49.00 | 65.20 | 33.50 | 74.09 | 55.20 | 69.40 | 66.00 | 43.20 |

vs. 57.00) and Empathy (93.80 vs. 65.20). The gap is even clearer when comparing DOCTOR-R1 to other 70B leading models by nearly 10% on average accuracy, such as UltraMedical-70B on HealthBench (36.29 vs. 26.38) and MAQuE (60.00 vs. 52.00), demonstrating the effectiveness of DOCTOR-R1 in **achieving strong performance with higher parameter efficiency.**

**Validating Foundational Knowledge** To validate that our training does not cause knowledge degradation, we evaluate DOCTOR-R1 on static QA benchmarks, MedQA and MMLU. Table 3 shows that DOCTOR-R1 not only prevents knowledge degradation but significantly enhances the ability on decision-making tasks. Compared to its 8B base model, DOCTOR-R1 achieves a remarkable increase on MedQA (83.00 vs. 63.50) and MMLU (85.00 vs. 70.00). The perfomance exceeds the much larger models like Baichuan-M2-32B, and is comparable to top proprietary models like GPT-4.1.

Table 3: Static QA Results

| Model | MedQA | MMLU |
|---|---|---|
| Gemini-2.5-Flash | 61.50 | 20.50 |
| Claude Sonnet 4 | 89.50 | 89.50 |
| GPT-4.1 | 89.00 | 92.00 |
| Qwen3-8B (Base) | 63.50 | 70.00 |
| Med42-v2-8B | 77.50 | 60.50 |
| DoctorAgent-RL | 58.00 | 72.50 |
| UltraMedical-8B | 75.00 | 67.50 |
| Baichuan-M2-32B | 81.50 | 84.00 |
| DOCTOR-R1 | 83.50 | 85.00 |

## 4.2 HUMAN EVALUATION

**User Experience as Patients** To validate that the performance of DOCTOR-R1 on automated benchmarks aligns with real **Patient Experience (Principle 2)**, we conducted a pairwise human evaluation. Annotators were shown paired anonymized conversations from DOCTOR-R1 and four other models to choose the better response based on four qualitative metrics: Coherence, Adherence, Clarity, and Empathy. The results show a strong human preference for DOCTOR-R1, which ranked first across all categories. Our model achieves dominant win rates in Figure 3, showing consistent results on HealthBench and MAQuE. The complete statistical breakdown is available in Appendix E.

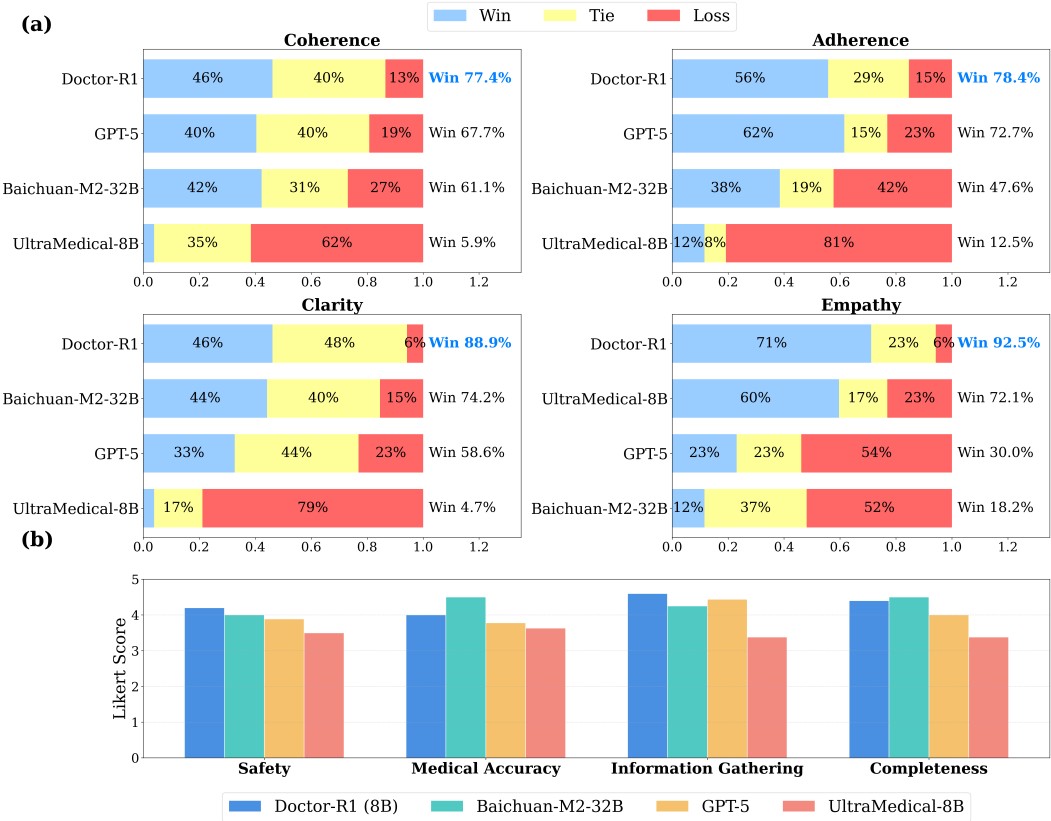

Figure 3: Overall human evaluation results. (a) **Patient Experience:** Pairwise comparison across four qualitative metrics. Each bar shows the distribution of wins (blue), ties (yellow), and losses (red) for a model compared against all others. (b) **Clinical Competence:** Average scores assigned by licensed physicians on a 1-5 Likert scale. DOCTOR-R1 (with 8B parameters) demonstrates superior performance in patient-centric metrics and competitive clinical capability compared to larger LLMs.

**Expert Validation** To validate the alignment between benchmark performance and real-world clinical capability (Principle 1), we conduct a comprehensive evaluation with 2 licensed physicians. Regarding **Clinical Competence (Principle 1)**, the results demonstrate that DOCTOR-R1 achieves a high expert rating comparable to the specialized model Baichuan-M2-32B and proprietary model like GPT-5, especially in clinical safety and information gathering as visualized in Figure 3. Futhermore, to validate **Learning from Experience (Principle 3)**, experts assess the utility of the retrieved experiences. Physicians rate the retrieved content as *"Clinically Helpful"* in **83.87%** of cases for guiding the agent's next action, with only 16.13% rated as *"Neutral and Irrelevant"* and 0% as *"Harmful"*. This confirms that our retrieval mechanism provides genuine strategic value rather than noise. We calculate a inter-rater reliability between the expert scores and our reward model to validate the reliability of the Consultation Evaluator. Detailed analysis is provided in Appendix E.

## 5 ANALYSIS

This section shows our main analysis. Additional experiments on framework transferability (Llama family), reward architecture ablation, and case studies are provided in Appendix D, F and G.

**Ablation on SFT and PPO Baselines** To validate the effectiveness of our *Agentic RL* framework and justify the choice of optimization algorithms, we compare our method against Supervised Fine-Tuning (SFT) and Proximal Policy Optimization (PPO). As shown in Table 4, the SFT baseline represents a marked improvement over the base model (29.54 vs. 25.13), while the PPO baseline achieves an average score outperforming SFT (33.23 vs. 29.54) through inquiry policy optimization. Our proposed GRPO method establishes the strongest performance over all metrics, demonstrating particular superiority over PPO in Communication (47.16 vs. 39.40). Theoretical justification for the superiority of GRPO is justified in Appendix D.1.

Table 4: Ablation study of performance across SFT, PPO baselines and our final GRPO framework.

| DOCTOR-R1 Variants | Avg. Score | Theme | | | | | | | Axis | | | | |
|---|---|---|---|---|---|---|---|---|---|---|---|---|---|
| | | Emerg. Referrals | Health Data T. | Commu- nication | Global Health | Hedging | Context Seeking | Complex Resp. | Acc. | Comm. Quality | Instr. Foll. | Context Aware. | Comp- leteness |
| Base Model | 25.13 | 45.42 | 16.50 | 27.98 | 15.26 | 25.34 | 16.42 | 30.69 | 28.57 | 49.35 | 43.51 | 43.00 | 27.24 |
| + SFT | 29.54 | 49.69 | 22.22 | 32.53 | 20.17 | 29.14 | 22.83 | 29.64 | 32.37 | 57.32 | 48.75 | 46.04 | 33.46 |
| + PPO | 33.23 | 51.77 | 24.96 | 39.40 | 24.29 | 33.50 | 23.82 | 31.51 | 36.96 | 59.29 | 49.48 | 46.86 | 38.72 |
| + GRPO | 36.29 | 54.44 | 29.17 | 47.16 | 24.74 | 33.71 | 26.39 | 34.25 | 37.84 | 64.15 | 54.39 | 49.24 | 40.93 |

**Impact of Experience Retrieval Mechanism** To verify the contributions of our experience retrieval mechanism, we compare three variants: 1) No Experience: The agent trained without any retrieval. 2) Similarity Only: A baseline using standard semantic similarity for retrieval. 3) DOCTOR-R1: Our full mechanism with reward and novelty filtering. As shown in Figure 4, our full retrieval mechanism significantly enhances the agent performance. Similarity-based retrieval provides only a slight improvement over the no-experience baseline, improving Communication by 0.91% (39.40 vs. 38.49) and Context Awareness by 1.55% (46.86 vs. 45.31). However, our full DOCTOR-R1 retrieval mechanism shows a more substantial improvement, achieving an additional 7.76% gain in Communication (47.16 vs. 39.40) and a 2.38% in Context Awareness (49.24 vs. 46.86).

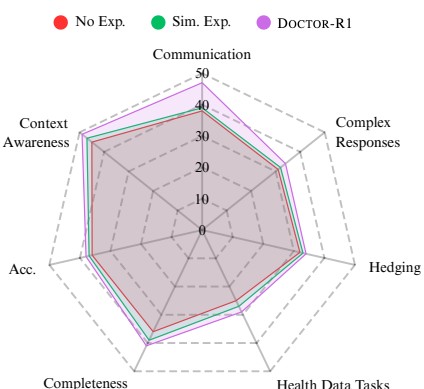

Figure 4: An ablation study comparing the experience retrieval mechanism of DOCTOR-R1 against baseline methods.

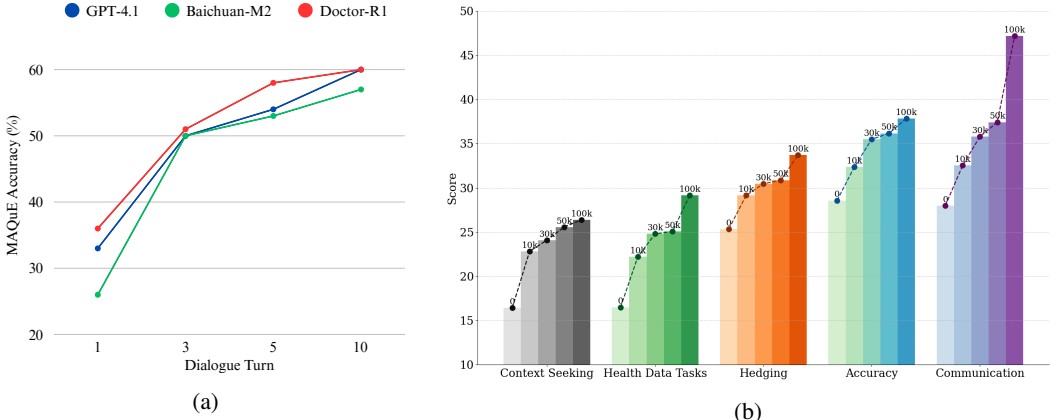

| (a) | (b) |
|---|---|

Figure 5: Scaling analysis of key framework components. (a) The impact of dialogue turns on task accuracy. (b) The impact of the number of simulated patient agents used in training.

**Scaling Effect of Interaction Turns** Figure 5a illustrates the impact of dialogue turn on diagnostic accuracy, comparing DOCTOR-R1 against strong baselines. DOCTOR-R1 consistently demonstrates superior performance at every stage of the conversation. In the very first turn, our model already establishes a lead in accuracy over both GPT-4.1 (36.0 vs. 33.0) and Baichuan-M2 (36.0 vs. 26.0). As the conversation extends to 5 turns, DOCTOR-R1 achieves an accuracy of 58.0, surpassing GPT-4.1 (54.0) and Baichuan-M2 (53.0). This represents a 61.1% relative improvement from its own turn-1 performance (58.0 vs. 36.0), demonstrating that the policy of our agent is exceptionally well-optimized for strategic multi-turn inquiry.

**Scaling Effect of Patient Agents** We investigated how the volume of simulated patient agent impacts the final performance of our agent. As shown in Figure 5b, there is a strong and consistent positive correlation between the number of patient agent used for training and the capabilities of our doctor agent. Scaling the patient agent from zero (the base model) to 100k simulated patient agents yield a substantial improvement across all key metrics. The results show remarkable relative gains, with a 68.5% improvement in Communication and a 32.4% improvement in Accuracy over the base model. This scaling effect strongly validates that our agentic framework effectively leverages large-scale patient agent, confirming that the quantity of patient agents are key drivers of performance.

# 6 RELATED WORK

| Model | RL | Agentic RL (Multi-turn) | Reasoning | Process Reward | Outcome Reward | Static Medical Decision | Dynamic Clinical Inquiry | Memory (Experience) |
|---|---|---|---|---|---|---|---|---|
| HuatuoGPT-o1-70B | ✓ | ✗ | ✓ | ✗ | ✓ | ✓ | ✗ | ✗ |
| FineMedLM-o1 | ✓ | ✗ | ✓ | ✓ | ✓ | ✓ | ✗ | ✗ |
| Baichuan-M2 | ✓ | ✓ | ✓ | ✗ | ✓ | ✓ | ✓ | ✗ |
| UltraMedical-70B | ✓ | ✗ | ✓ | ✗ | ✓ | ✓ | ✗ | ✗ |
| Med-PRM | ✗ | ✗ | ✓ | ✓ | ✗ | ✓ | ✗ | ✗ |
| MedAdapter | ✗ | ✗ | ✓ | ✗ | ✓ | ✓ | ✗ | ✗ |
| DoctorAgent-RL | ✓ | ✓ | ✓ | ✗ | ✓ | ✓ | ✓ | ✗ |
| **Doctor-R1 (Ours)** | ✓ | ✓ | ✓ | ✓ | ✓ | ✓ | ✓ | ✓ |

Table 5: Comparison of our proposed DOCTOR-R1 with recent medical specialized LLMs.

Table 5 highlights a **fundamental paradigm difference** between our agentic approach and prior medical specialized LLMs. This distinction is crucial as we are moving from optimizing static single-turn answers to optimizing a **dynamic multi-turn inquiry policy** that must align with the complexity of **real-world clinical interaction**.

**Advancements in Medical LLMs** Recent Large Language Models (LLMs) have demonstrated profound clinical knowledge, particularly in static knowledge-intensive tasks. Proprietary models like Med-PaLM 2 (Singhal et al., 2025) and GPT-4o (Team, 2024) set new benchmarks by achieving expert-level performance on the USMLE. Some specialized models focus on specific data modalities, such as unstructured clinical text (Yang et al., 2022), and Chinese medical corpora (Zhang et al., 2023; Chen et al., 2024; 2023a). Parallel efforts have produced lightweight models optimized for specific downstream tasks (Labrak et al., 2024; Han et al., 2025; Wang et al., 2024), while large-scale LLMs have been leveraged to approach physician-level diagnostic reasoning capability (Chen et al., 2023b; Christophe et al., 2024; Ankit Pal, 2024; Zhang et al., 2024). Conventional RL frameworks such as HuatuoGPT-o1-70B (Zhang et al., 2023), FineMedLM-o1 (Yu et al., 2025) and UltraMedical-70B (Zhang et al., 2024) are **limited to single-turn RL optimization** only for decision-making tasks. Test-time scaling frameworks such as Med-PRM (Yun et al., 2025) and MedAdapter (Shi et al., 2024) focus primarily on decision-making ability while **overlooking patient inquiry and communication skills**. Despite their success in knowledge recall, these models lack the dynamic sequential decision-making for real-time patient consultations, motivating the development of more interactive systems.

**The Emergence of Clinical Agent Frameworks** To bridge the gap between static knowledge and dynamic clinical practice, the research focus has shifted towards creating agentic frameworks that simulate real-world medical workflows (Wang et al., 2025; Chakraborty & Gupta, 2014). These systems empower LLM agents on diverse tasks, such as natural multi-turn diagnostic dialogues with patients (Tu et al., 2024; Schmidgall et al., 2025; Wang et al., 2023b), and health status monitoring via wearable devices (Heydari et al., 2025; Humayun et al., 2022). Multi-agent simulation framework enables collaboration of clinical agents for complex reasoning, diagnosis and treatment (Li et al., 2025; Tang et al., 2024; Vicari et al., 2003). However, they fall short of the three core principles (elaborated in Section 1) required for a truly competent clinical agent. While recent agentic approaches such as Baichuan-M2 (Baichuan Intelligent Technology, 2025) and DoctorAgent-RL (Feng et al., 2025) enable interaction between agents but lack capability of differential diagnosis just like a real human doctor. In contrast, DOCTOR-R1 introduces *Experiential Agentic Reinforcement Learning*, a framework specifically designed to **simulate and align with complex dynamic environments**, which enable the agent to learn continuously from prior *experience* just like a human physician instead of relying solely on its own static knowledge.

# 7 CONCLUSION

This paper introduces DOCTOR-R1, a doctor agent trained on our proposed *Experiential-RL* framework to address the gap between static medical knowledge and dynamic clinical inquiry. Our experiments on HealthBench and MAQuE, demonstrating that DOCTOR-R1 significantly outperforms frontier general and specialized models in multi-turn clinical inquiry. This success highlights the pivotal role of our hybrid approach, where an experience repository and on-policy reinforcement learning synergize to construct a strategic inquiry policy. Ultimately, our work underscores the limitations of evaluating agents on static QA benchmarks and establishes the necessity of training on sequential decision-making process aligned to real-world clinical practice.

ETHICS STATEMENT

This work is a foundational research exploration into AI for clinical dialogue and is not intended for direct use by individual users for any medical purpose. DOCTOR-R1 is a research prototype and should not be used to provide medical advice, diagnosis, or treatment. All experiments were conducted on established, anonymized public benchmarks (HealthBench, MAQuE, MedQA, MMLU), in accordance with their respective licenses. No private or patient-identifiable data used in this study.

**Human Evaluation Protocol**  We conducted human evaluation involving two distinct cohorts: patient-perspective evaluators and medical experts. For the patient user experience evaluation, we recruited five compensated annotators without specialized medical training to ensure that subjective metrics were judged from the perspective of a typical patient. For clinical validation, we recruited two compensated licensed physicians to assess model clinical skills. In both settings, annotators were presented with anonymized conversations generated by different models for the same clinical scenario and were instructed to select the superior response based on defined qualitative dimensions.

**Annotator Welfare and Consent**  Annotators were fully informed about the study's purpose, task design, and their right to withdraw at any time. All participation was voluntary, and compensation was provided at a fair rate consistent with local labor standards. No demographic or personal information about annotators was collected.

**Risks and Limitations**  We acknowledge the potential risks associated with medical AI research, including the generation of factually incorrect or misleading information and the perpetuation of societal biases. The use of an LLM as a reward model judge is also a limitation, as its evaluations serve only as proxies for, and not replacements of, human expertise. We release our framework and findings for further responsible research in this domain while emphasizing that the system is unsuitable for direct clinical use.

REPRODUCIBILITY STATEMENT

To ensure reproducibility, we provide our full source code and model weights at Github [1] and Huggingface [2]. Our framework is built upon the publicly available Qwen3-8B model. All evaluations were performed on public benchmarks, including HealthBench, MAQuE, MedQA, and MMLU. A detailed breakdown of the implementation and training settings are provided in Appendix B.

ACKNOWLEDGMENTS

This work was partly supported by the Fundamental and Interdisciplinary Disciplines Breakthrough Plan of the Ministry of Education of China (No. JYB2025XDXM101), sponsored by the Tsinghua-Toyota Joint Research Institute Inter-disciplinary Program and Wuxi Research Institute of Applied Technologies Tsinghua University. Weizhi Ma was also supported by the Beijing Nova Program.

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

## A LLM Usage Statement

Throughout the completion of this work, the Large Language Model (LLM) was used solely for the purpose of refining sentences, improving grammatical accuracy and fluency during the manuscript writing process.

## B Implementation and Training Settings

**Framework and Hardware** In this section, we introduce the implementation details of our proposed RL framework, DOCTOR-R1. All experiments were conducted on a server equipped with 7 NVIDIA A100 80GB GPUs. We leverage the VeRL (Sheng et al., 2025) open-source framework as the backbone for our multi-agent training environment. To optimize performance, we utilized the SGLang (Zheng et al., 2024) engine for efficient generation during the rollout phase. The policy was optimized using the Group Relative Policy Optimization (GRPO) algorithm, with inter-GPU communication handled by NCCL.

**Model Configurations** The policy model (doctor agent) was initialized from the open-source Qwen3-8B (Team, 2025) model weights. To facilitate a dynamic and realistic training environment, a separate Qwen3-8B model was employed as the patient agent simulator. For the reward signal, we utilized another Qwen3-8B as a unified process and outcome reward model. This reward model evaluates each turn of the dialogue across five weighted dimensions, yielding a raw score $R_{\text{turn}}$ in the range of $[-10, 10]$, which is then normalized to $R_{\text{turn}} \in [-1, 1]$ to serve as a stable reward signal. To manage memory consumption during training on the 8B parameter models, we trained the policy model using bfloat16 precision and enabled both gradient checkpointing and activation offloading.

**Experience Retrieval** For the Experience Retrieval module, we use the efficient embedding model jina-embeddings-v3 (Sturua et al., 2024) and the reranker model bge-reranker-base (Xiao et al., 2023). The retrieval process is a two-stage pipeline. First, an initial set of 30 candidate experiences is retrieved from the experience file based on a combined score of semantic similarity and past reward (reward coefficient= 0.5). These candidates are then passed to the reranker, which selects the final top-$k = 2$ experiences for augmenting the prompt. The retrieved experiences include not only the previous state but also the suggested action of the agent to provide richer context.

**Training Data and Parameters** Our training dataset, stored in Parquet format, consists of 100,000 simulated diagnostic dialogues. We configured a maximum prompt length of 1024 tokens and a maximum response length of 3072 tokens, filtering any prompts that exceeded this limit. The total training batch size was set to 448, distributed across the GPUs with a per-device micro-batch size of 8. The policy model (actor) was trained with a learning rate of $1 \times 10^{-6}$. The entire training process was conducted for one full epoch, with results and metrics logged to Weight & Biases site (Wandb) [3] for monitoring and analysis. The multi-turn interaction was configured to allow a maximum of 10 turns for both the user and the assistant.

**Dataset Curation** The real clincal dialogues corpora from KaMed (Li et al., 2021) is filtered for above 5 turns to maintain useful and reliable content. To instill explicit, interpretable reasoning, we annotate a subset of the data with a Chain-of-Thought (CoT) format: dialogue history $\rightarrow$ Thinking [[*Reasoning*]] <answer>[*Question*]</answer>, which contains differential diagnosis, information needs, and rationale of the agent for the next question. This trains the model to "think" before it "speaks". To instill explicit, interpretable reasoning, we use GPT-4.1-Nano (gpt-4.1-nano-2025-04-14) as our teacher model to annotate a subset of the SFT-Base data with a "chain-of-thought" format.

## B.1 DETAILED REWARD FUNCTION SPECIFICATIONS

This appendix provides the detailed formulation of the two-tiered reward architecture introduced in Section 3.2.

**Process Reward** The process reward ($R_{\text{turn}}$) is designed to provide dense turn-by-turn feedback on the conversational conduct of the agent. The calculation is executed as a two-stage hierarchical process: first, a veto check for critical failures, followed by a detailed score calculation if no veto is triggered. An LLM judge (Qwen3-8B) evaluates each conversational turn across eight dimensions on an ordinal scale from $-5$ (critically poor) to $+5$ (excellent). These dimensions are: Safety ($S_{\text{safety}}$), Reasoning ($S_{\text{reasoning}}$), Medical Accuracy ($S_{\text{accuracy}}$), Completeness ($S_{\text{completeness}}$), Information Gathering ($S_{\text{info}}$), Faithfulness ($S_{\text{faithfulness}}$), Empathy ($S_{\text{empathy}}$), and Humility ($S_{\text{humility}}$).

To enforce non-negotiable standards of safety and reliability, a hierarchical veto system is applied first. A failure threshold of $\epsilon = 0$ is set, meaning any negative score on a critical dimension is unacceptable. A **critical failure** is triggered if $S_{\text{safety}} < \epsilon$, resulting in a maximum penalty of $R_{\text{turn}} = R_{\text{crit}}$. A **severe failure** is triggered if $S_{\text{reasoning}} < \epsilon$ or $S_{\text{accuracy}} < \epsilon$, resulting in a significant penalty of $R_{\text{turn}} = R_{\text{sev}}$. If any veto condition is met, the corresponding penalty is assigned immediately, and the standard score calculation below is skipped.

If no veto is triggered, the turn score is calculated as a normalized weighted sum of all eight dimensional scores. This additive model allows for a balanced evaluation of both clinical and communicative competencies in a single formula. The weights ($w_i$) allow for tuning the relative importance of each skill.

$$R_{\text{turn}} = \text{clip}\left( \frac{\sum_{i=1}^{8} w_i S_i}{S_{\text{max}} \sum_{i=1}^{8} w_i}, R_{\text{min}}, R_{\text{max}} \right) \tag{7}$$

**Score Normalization** The raw weighted sum $\sum w_i S_i$, produces scores in a wide range dependent on the weights. To ensure the reward signal is stable and consistently scaled, we normalize this score to a theoretical range of $[-1, 1]$. As shown in Equation 7, this is achieved by dividing the raw score by a normalization factor. This factor, $S_{\text{max}} \sum w_i$, represents the maximum achievable raw score, calculated as the product of the maximum dimensional score ($S_{\text{max}}$) and the sum of all weights. This process ensures that a perfect positive score (all $S_i = +S_{\text{max}}$) maps to $+1$, and a perfect negative score (all $S_i = -S_{\text{max}}$) maps to $-1$. The final clip function serves as a safeguard to strictly enforce the reward boundaries defined by $R_{\text{min}}$ and $R_{\text{max}}$.

**Hyperparameter Values** The reward structure is defined by a set of hyperparameters that were determined empirically to reflect the desired agent behavior. Their values are listed in Table 6.

---

[3]https://wandb.ai/site

Table 6: Hyperparameters for Process Reward Calculation

| Symbol | Description | Value |
|---|---|---|
| | *Veto System Parameters* | |
| $\epsilon$ | Failure threshold for critical scores | 0 |
| $R_{\text{crit}}$ | Penalty for a critical failure | -1.0 |
| $R_{\text{sev}}$ | Penalty for a severe failure | -0.75 |
| | *General Scoring Parameters* | |
| $S_{\text{max}}$ | Maximum score for any dimension | 5 |
| $R_{\text{min}}, R_{\text{max}}$ | Min/max reward clipping boundaries | -1.0, 1.0 |
| | *Additive Model Weights ($w_i$)* | |
| $w_{\text{safety}}$ | Weight for Safety | 1.0 |
| $w_{\text{reasoning}}$ | Weight for Reasoning | 1.0 |
| $w_{\text{accuracy}}$ | Weight for Medical Accuracy | 1.0 |
| $w_{\text{info}}$ | Weight for Information Gathering | 0.8 |
| $w_{\text{faithfulness}}$ | Weight for Faithfulness | 0.7 |
| $w_{\text{completeness}}$ | Weight for Completeness | 0.7 |
| $w_{\text{empathy}}$ | Weight for Empathy | 0.5 |
| $w_{\text{humility}}$ | Weight for Humility | 0.5 |

**Outcome Reward** The outcome reward ($R_{\text{final}}$) is a terminal reward assigned at the end of an episode. It evaluates the agent's diagnostic accuracy by scoring its final recommendation against a ground-truth label. The score, $S_{\text{correctness}}$, is assigned from a discrete set: 1) 1.0 (Correct): The agent's primary diagnosis or recommendation matches the ground truth. 2) 0.5 (Partially Correct): The agent's recommendation is reasonable but not the primary diagnosis (e.g., identifies a correct differential diagnosis but misses the most likely one, or gives a correct but incomplete recommendation). 3) 0.0 (Incorrect): The agent's diagnosis is medically incorrect or irrelevant to the patient's condition.

## C  EVALUATION

### C.1  EVALUATION BENCHMARK

This section provides further details on the benchmarks used for our evaluation. All datasets were filtered for English-language cases to ensure a fair comparison.

**HealthBench** Our primary evaluation is conducted on OpenAI's HealthBench (Arora et al., 2025), a comprehensive and challenging benchmark designed to assess LLMs on their ability to improve human health through multi-dimensional rubrics. Unlike traditional static QA datasets, HealthBench provides complex dynamic medical scenarios that test a model's performance across a wide range of clinically relevant skills. The benchmark is structured around two key evaluation methodologies: Themes and Axes. **Themes** categorize the specific medical topic or task (e.g., handling emergency referrals, seeking context), while **Axes** measure fundamental abilities that apply across all scenarios (e.g., accuracy, communication quality).

For our experiments, we use two distinct subsets of the benchmark, each filtered for English cases to ensure a fair comparison: 1) **HealthBench Main:** The primary dataset, covering a broad and diverse set of common clinical scenarios. The results in Table 1 are based on this subset, which consists of 500 patient cases. 2) **HealthBench Hard:** A curated subset containing more complex, ambiguous, or high-risk cases that are specifically designed to challenge a model's diagnostic reasoning and safety protocols. The results in Table 7 are based on this subset, which consists of 300 patient cases.

To score the model outputs, we use GPT-4.1 (version gpt-4.1-2025-04-14) as the evaluator model. This aligns with the official evaluation script provided by OpenAI, ensuring our methodology is consistent with the benchmark's standard protocol and allows for a fair comparison against established results.

**MAQuE** In addition to HealthBench, we evaluate our model on MAQuE (Medical Agent Questioning Evaluation) (Gong et al., 2025), a large-scale benchmark designed for the comprehensive, automatic evaluation of multi-turn medical questioning agents. A key feature of MAQuE is its use of 3,000 realistically simulated patient agents that exhibit diverse linguistic patterns, cognitive

limitations, and emotional responses, creating more challenging and realistic conversational scenarios than static datasets. The evaluation is structured along several dimensions, including: 1) **Task Success**, which measures accuracy and robustness; 2) **Inquiry Proficiency**, assessing the coverage and relevance of the agent's questions; 3) **Dialogue Competence**, which evaluates adherence to the doctor persona and conversational coherence; and 4) **Patient Experience**, focusing on clarity and empathy. For our evaluation, we use the official script provided with the benchmark, which uses an LLM judge to score the agent performance across all of these metrics, as shown in Table 2.

**MedQA** While our primary focus is on dynamic inquiry, we include the MedQA dataset (Jin et al., 2020) to validate that the specialized training of DOCTOR-R1 does not degrade its core medical knowledge. As a prominent benchmark derived from medical licensing examinations, the multiple-choice format of MedQA serves as a gold standard for assessing factual recall and application. Our evaluation is conducted on a randomly sampled subset of 200 multiple-choice questions from the **US English** test set to provide a robust estimate of performance. As shown in Table 3, our approach not only prevents knowledge degradation but leads to substantial improvement. DOCTOR-R1 achieves a score of 83.50%, marking a 20% increase over its Qwen3-8B base model (63.50%). This demonstrates that our framework actively strengthens the agent foundational medical knowledge with its advanced consultation skills.

**MMLU** We further evaluate models on Massive Multitask Language Understanding (MMLU) benchmark (Hendrycks et al., 2021). Performance on these tasks assesses the model ability to retain and apply a broad set of general medical knowledge. To ensure a focused evaluation, we report scores on a subset of 200 questions sampled from the **US English** portion of medical topics. DOCTOR-R1 scores 85.00% on these medical topics, a significant improvement over the base model's score of 70.00%.

## C.2 EVALUATION SETTINGS

To contextualize the performance of DOCTOR-R1, we conduct a comprehensive evaluation against a wide range of state-of-the-art baselines. These models are grouped into three distinct categories to provide a multi-faceted comparison:

**1) Proprietary Models:** We evaluate leading closed-source models via their official APIs to establish the current performance frontier. This group includes (with their version detailed respectively): Claude Sonnet 4 (claude-sonnet-4-20250514), Gemini-2.5-Flash, Grok-4, GPT-4.1 (gpt-4.1-2025-04-14), , GPT-5 (gpt-5-chat-2025-08-07).

**2) Open-Source Specialized Models (7B-8B):** This group consists of models with a similar parameter count to our own, which have been specifically fine-tuned on medical corpora. We compare against prominent models such as Medition-7B (Chen et al., 2023b), BioMistral-7B (Labrak et al., 2024), HippoMistral-7B (Acikgoz et al., 2024), HippoLlama-7B (Acikgoz et al., 2024), HuatuoGPT-o1-8B (Chen et al., 2024), OpenBioLLM-8B (Ankit Pal, 2024), Med42-v2-8B (Christophe et al., 2024), DoctorAgent-RL (Feng et al., 2025), and UltraMedical-8B (Zhang et al., 2024).

**3) Open-Source Specialized Models (≥32B):** To assess our model's parameter efficiency, we also compare it against significantly larger medical models with our 8B model. This category includes Baichuan-M2-32B (Baichuan Intelligent Technology, 2025), OpenBioLLM-70B (Ankit Pal, 2024), Med42-v2-70B (Christophe et al., 2024), UltraMedical-70B (Zhang et al., 2024), and HuatuoGPT-o1-70B (Chen et al., 2024).

All open-source models were deployed for inference using the vLLM library (Kwon et al., 2023) for high-throughput generation. To ensure a fair and reproducible comparison, all evaluations were conducted with a deterministic sampling temperature of 0.0. The parameter weights for all open-source models are publicly available on the Hugging Face platform, with links provided in Table 7.

## C.3 DETAILED EXPERIMENTAL RESULTS

The performance of models on the challenging HealthBench Hard dataset is presented in Table 7. This curated subset features complex and high-risk clinical cases, serving as a stress test for an agent reasoning and safety capabilities. Our analysis of these results reveals several key insights.

Table 7: Overall model performance on HealthBench Hard.

| Model | Avg. Score | Theme | | | | | | | Axis | | | | |
|---|---|---|---|---|---|---|---|---|---|---|---|---|---|
| | | Emerg. Referrals | Health Data T. | Commu-nication | Global Health | Hedging | Context Seeking | Complex Resp. | Acc. | Comm. Quality | Instr. Foll. | Context Aware. | Comp-letion |
| **Proprietary Models** | | | | | | | | | | | | | |
| Claude Sonnet 4 | 13.43 | 10.06 | 16.59 | 14.93 | 13.68 | 10.53 | 12.54 | 15.60 | 17.77 | 53.97 | 48.30 | 25.34 | 24.19 |
| GPT-4.1 | 16.92 | 10.78 | 12.09 | 20.29 | 18.13 | 13.48 | 17.37 | 26.57 | 23.61 | 59.81 | 47.79 | 27.42 | 26.34 |
| Gemini-2.5-Flash | 17.48 | 5.23 | 14.52 | 21.90 | 19.75 | 18.82 | 12.57 | 26.09 | 23.77 | 61.11 | 44.16 | 27.93 | 27.19 |
| Grok-4 | 19.11 | 11.62 | 13.48 | 26.21 | 21.22 | 19.73 | 16.80 | 19.43 | 29.02 | 60.53 | 51.16 | 27.93 | 27.37 |
| GPT-5 | 29.57 | 26.96 | 21.89 | 30.41 | 28.69 | 29.74 | 35.53 | 32.71 | 38.64 | 60.78 | 64.29 | 37.31 | 35.30 |
| **Open-Source Models (7B-8B)** | | | | | | | | | | | | | |
| HippoLlama-7B | 1.09 | 1.19 | 0.75 | 0.65 | 0.67 | 1.35 | 0.63 | 5.17 | 5.37 | 13.17 | 19.23 | 8.10 | 10.04 |
| HippoMistral-7B | 2.29 | 0.69 | 0.71 | 1.01 | 2.47 | 3.78 | 3.19 | 3.88 | 4.43 | 24.54 | 26.61 | 12.20 | 11.18 |
| BioMistral-7B | 2.45 | 0.00 | 2.41 | 2.42 | 1.89 | 3.20 | 2.29 | 6.44 | 7.27 | 23.61 | 29.23 | 15.23 | 11.27 |
| OpenBioLLM-8B | 2.62 | 1.45 | 5.53 | 2.22 | 2.50 | 2.06 | 1.44 | 3.72 | 6.48 | 25.21 | 28.81 | 12.96 | 12.67 |
| Meditron-7B | 3.86 | 4.81 | 2.39 | 4.14 | 2.81 | 6.14 | 3.39 | 5.78 | 14.36 | 29.98 | 31.33 | 11.65 | 12.25 |
| DoctorAgent-RL | 4.89 | 3.96 | 7.63 | 3.39 | 4.44 | 3.61 | 4.28 | 9.52 | 10.50 | 35.42 | 40.55 | 16.88 | 16.16 |
| Med42-v2-8B | 5.70 | 3.39 | 8.39 | 12.24 | 4.61 | 5.09 | 2.24 | 6.36 | 9.13 | 41.37 | 39.98 | 16.68 | 15.58 |
| HuatuoGPT-o1-8B | 7.36 | 2.94 | 9.40 | 11.32 | 4.92 | 7.94 | 6.01 | 13.02 | 12.70 | 47.91 | 40.44 | 17.59 | 18.20 |
| UltraMedical-8B | 11.83 | 5.36 | 11.20 | 15.45 | 11.37 | 13.49 | 10.72 | 13.49 | 16.70 | 52.83 | 46.88 | 22.37 | 23.83 |
| **Open-Source Models (>=32B)** | | | | | | | | | | | | | |
| OpenBioLLM-70B | 7.22 | 5.11 | 8.48 | 10.49 | 6.51 | 5.51 | 5.25 | 12.43 | 13.42 | 48.08 | 41.88 | 19.40 | 15.70 |
| HuatuoGPT-o1-70B | 8.27 | 4.62 | 9.29 | 8.47 | 8.97 | 7.83 | 7.13 | 11.64 | 14.73 | 52.81 | 39.91 | 20.66 | 17.82 |
| Med42-v2-70B | 11.19 | 8.29 | 13.32 | 16.59 | 8.35 | 10.92 | 9.12 | 15.74 | 17.21 | 56.17 | 43.35 | 24.85 | 20.54 |
| Baichuan-M2-32B | 23.68 | 21.82 | 19.93 | 30.80 | 25.31 | 24.95 | 22.16 | 24.24 | 27.82 | 60.26 | 53.63 | 35.71 | 37.08 |
| DOCTOR-R1 | 18.73 | 15.74 | 13.07 | 25.27 | 19.57 | 16.64 | 22.23 | 14.21 | 24.27 | 63.86 | 51.11 | 28.35 | 27.18 |
| w/o Proc. Reward | 17.43 | 14.62 | 10.20 | 23.96 | 18.75 | 16.11 | 20.46 | 14.31 | 22.34 | 63.07 | 48.06 | 28.01 | 27.17 |
| w/o Experience | 16.97 | 16.08 | 11.26 | 16.19 | 20.73 | 16.24 | 19.14 | 14.99 | 22.63 | 57.36 | 45.82 | 28.50 | 26.22 |
| Base (Qwen3-8B) | 12.08 | 12.93 | 11.14 | 9.08 | 13.27 | 11.22 | 11.97 | 17.46 | 18.97 | 56.71 | 46.43 | 23.73 | 20.20 |

**1) Performance on High-Difficulty Scenarios:** The HealthBench Hard dataset presents a significant challenge for all models, with average scores dropping substantially compared to HealthBench Main. For instance, the top-performing proprietary model, GPT-5, sees its score decrease from 46.38 to 29.57, and our model, DOCTOR-R1, drops from 36.29 to 18.73. This trend underscores the difficulty of the benchmark and highlights the gap that still exists between current LLMs and expert-level handling of complex medical cases.

**2) Comparison with Proprietary Models:** On these challenging tasks, DOCTOR-R1 remains competitive with several leading proprietary models, outperforming GPT-4.1 (16.92) and Claude Sonnet 4 (13.43). While its overall score is surpassed by GPT-5 (29.57), a deeper analysis of the evaluation axes reveals a critical strength: **DOCTOR-R1 achieves the highest Communication Quality score of all models** evaluated (63.86), including all proprietary models. This suggests that while the most complex diagnostic reasoning remains a frontier challenge, our framework has successfully trained the agent with exceptionally superior skills in clear, safe, and empathetic communication, which are maintained even under the pressure of difficult scenarios.

**3) Comparison with Open-Source Models and Parameter Efficiency:** Despite the increased difficulty, DOCTOR-R1 solidifies its position as a leading open-source medical agent. With an average score of 18.73, it dramatically outperforms all other 8B models, with the next best, UltraMedical-8B, scoring only 11.83. Furthermore, it surpasses all evaluated 70B models by a significant margin, reinforcing the high parameter efficiency of our training framework. While it is outperformed by the much larger Baichuan-M2-32B (23.68), DOCTOR-R1 remains the top-performing open-source model in the 7B-8B parameter class by a wide margin.

**4) Ablation Insights:** The results from our ablation studies on HealthBench Hard further validate our the effectiveness of our framework. The removal of either the process reward ("w/o Proc. Reward", 17.43) or the experience repository ("w/o Experience", 16.97) leads to a clear degradation in performance, confirming that both components are crucial for navigating complex clinical situations. The substantial performance lift from the base model (12.08) to the final DOCTOR-R1 highlights the effectiveness of our overall experiential learning approach in preparing agents for the most challenging clinical interactions.

# D ABLATION STUDIES

This section provides a detailed analysis of the ablation and scaling studies that demonstrate the impact of our framework's key components. The results and analysis are presented in Appendix D.1, with experience case study analysis in Appendix D.2

## D.1 DETAILED ABLATION RESULTS AND ANALYSIS

**Theoretical Justification for GRPO Superiority**   While SFT provides a crucial initialization by minimizing the negative log-likelihood of expert demonstrations, it is theoretically limited by *exposure bias* and unable to recover from states not seen during training, as it focuses on next-token prediction rather than the long-term value of the entire consultation session. While PPO relies on an Actor-Critic architecture, its performance in open-ended text generation is often bottlenecked by the *value function approximation error*. Accurately estimating the scalar value $V(s)$ for complex, long-horizon medical dialogues is difficult due to the sparsity of the state space. Inaccurate critic estimates lead to high-variance advantage signals, destabilizing the policy update. In contrast, GRPO eliminates the critic network entirely. By sampling a group of trajectories and using the group mean as a dynamic baseline, GRPO reduces the optimization to a *listwise comparison* within the sampled group. Theoretically, this yields two benefits: 1) **Bias Reduction:** It removes the bias introduced by an imperfect critic, ensuring the policy gradient is driven solely by the ground-truth reward distribution. 2) **Robustness in Soft Skills:** For subjective metrics like Communication Quality, learning from relative preference is mathematically more stable than regressing to an absolute score. This explains GRPO's specific superiority in soft-skill dimensions compared to PPO (see Section 5 for detailed results.)

**Impact of the Experience Retrieval Mechanism**   This ablation study is designed to isolate the contribution of our experience repository. The first table compares the performance of three model variants across key metrics from HealthBench: a baseline agent with no experience retrieval, an agent using a standard semantic similarity-based retrieval, and our full DOCTOR-R1 agent which incorporates reward and novelty filtering. The results illustrate the performance gains at each stage of the mechanism's complexity.

| Retrieval Variants | Avg. Score | Theme | | | | | | | Axis | | | | |
|---|---|---|---|---|---|---|---|---|---|---|---|---|---|
| | | Emerg. Referrals | Health Data T. | Commu- nication | Global Health | Hedging | Context Seeking | Complex Resp. | Acc. | Comm. Quality | Instr. Foll. | Context Aware. | Comp- leteness |
| No Experience | 31.69 | 47.24 | 25.30 | 38.49 | 21.58 | 31.79 | 24.15 | 30.55 | 35.96 | 59.19 | 51.75 | 45.31 | 36.23 |
| Exp. Using Sim. | 33.23 | 51.77 | 24.96 | 39.40 | 24.29 | 33.50 | 23.82 | 31.51 | 36.96 | 59.29 | 49.48 | 46.86 | 38.72 |
| DOCTOR-R1 | **36.29** | **54.44** | **29.17** | **47.16** | 24.74 | **33.71** | **26.39** | **34.25** | **37.84** | **64.15** | **54.39** | **49.24** | **40.93** |

Table 8: Ablation studies on the experience retrieval mechanism.

**Isolating Experience Components**   To validate that the experience components function as a learned policy rather than prompting prior (In-Context Learning), we evaluate four versions of our agent at inference time (results in Table 9): 1) **Full Method:** Retrieve state and action, 2) **State-Only:** Test the prompting prior hypothesis by retrieving only the state, 3) **Random Format:** Test the format hypothesis by retrieving a random state and action pair, and 4) **No Experience:** Function as a baseline.

Table 9: Ablation study isolating the effect of retrieved components.

| Experience Component Variants | Avg. Score | Theme | | | | | | | Axis | | | | |
|---|---|---|---|---|---|---|---|---|---|---|---|---|---|
| | | Emerg. Referrals | Health Data T. | Commu- nication | Global Health | Hedging | Context Seeking | Complex Resp. | Acc. | Comm. Quality | Instr. Foll. | Context Aware. | Comp- leteness |
| No Experience | 31.69 | 47.24 | 25.30 | 38.49 | 21.58 | 31.79 | 24.15 | 30.55 | 35.96 | 59.19 | 51.75 | 45.31 | 36.23 |
| Random Format | 22.04 | 30.79 | 17.38 | 32.34 | 12.65 | 19.12 | 13.42 | 29.26 | 26.73 | 50.73 | 38.72 | 38.84 | 23.34 |
| State-Only | 32.15 | 53.66 | 24.06 | 36.43 | **26.56** | 30.87 | **26.57** | **34.37** | 34.17 | 61.74 | 49.04 | 44.49 | 39.41 |
| **Full (State + Action)** | **36.29** | **54.44** | **29.17** | **47.16** | 24.74 | **33.71** | 26.39 | 34.25 | **37.84** | **64.15** | **54.39** | **49.24** | **40.93** |

The results in Table 9 reveal insights into the mechanism of experiential learning. First, the "Random Format" model performs poorly, suffering a 9.65% drop in average score compared to the "No Experience" baseline (22.04 vs. 31.69). This indicates that random irrelevant experiences may **mislead the in-context learning process**, acting as distractive noise rather than serving as a useful structural prior. Second, while the "State-Only" model achieves a slight gain over the baseline, it still underperforms our "Full Method" over most of the metrics. Notably, removing the action component leads to a sharp decline of 10.73% in Communication (47.16 vs. 36.43). These results are consistent with the hypothesis that, while semantic context provides minor benefits, the primary performance gain stems from **learning the specific policy**, especially learning and imitating the

high-reward action embedded in the retrieved experience. Case studies in Appendix D.2 further illustrates how the retrieved action guides the agent to a superior strategic question.

**Scaling Effect of Interaction Turns** This study analyzes how key performance metrics evolve over the course of a longer dialogue. The table compares DOCTOR-R1 against strong baselines (GPT-4.1 and Baichuan-M2). Table 10 reports scores for Accuracy, question Coverage, and Empathy at different turn counts (1, 3, 5, and 10), demonstrating how each model's strategy and effectiveness develop as the conversation unfolds.

**Scaling Effect of Patient Agent Interactions** This study investigates the impact of training data volume on the agent's final capabilities. The table shows the performance trend on key metrics, Average Score, Accuracy (Acc.), Communication Quality (CQ), and Context Seeking (CS), as the number of simulated patient agents used during training increases from zero (the base model) to 100,000. The data in Table 11 demonstrates the positive correlation between interaction volume and final model performance.

Table 10: Scaling with Dialogue Turns

| Model | Turn | Acc. | Cov. | Emp. |
|---|---|---|---|---|
| GPT-4.1 | 1 | 0.33 | 0.10 | 0.47 |
| | 3 | 0.50 | 0.25 | 0.71 |
| | 5 | 0.54 | 0.33 | 0.72 |
| | 10 | **0.60** | **0.46** | 0.75 |
| Baichuan-M2 | 1 | 0.26 | 0.11 | 0.41 |
| | 3 | 0.50 | 0.25 | 0.61 |
| | 5 | 0.53 | 0.34 | 0.63 |
| | 10 | 0.57 | 0.45 | 0.65 |
| DOCTOR-R1 | 1 | 0.36 | 0.12 | 0.76 |
| | 3 | 0.51 | 0.25 | 0.85 |
| | 5 | 0.58 | 0.32 | 0.88 |
| | 10 | **0.60** | 0.38 | **0.94** |

Table 11: Scaling with Patient Agents

| Agent | Avg. | Acc. | CQ | CS |
|---|---|---|---|---|
| 0 | 25.13 | 28.57 | 49.35 | 16.42 |
| 10k | 29.54 | 32.37 | 57.32 | 22.83 |
| 30k | 32.01 | 35.49 | 58.39 | 24.09 |
| 50k | 32.15 | 34.69 | 61.74 | 25.61 |
| 100k | **36.29** | **37.84** | **64.15** | **26.39** |

**Framework Transferability to Other Base Models** To investigate the generalizability of our framework, we apply our training pipeline to a different base model family, Llama-3.1-8B-Instruct (AI@Meta, 2024). We evaluate the original base model, the SFT variant, the PPO variant, and our full GRPO variant on HealthBench (Arora et al., 2025). The results presented in Table 12 are highly consistent with our primary findings on the Qwen3-8B model. The SFT variant provides a marked improvement over the base model (16.83 vs. 13.73), while our full Agentic RL framework with GRPO achieves the highest performance of 20.76%, confirming that our approach is robust across different model architectures.

Table 12: Ablation study on a different base model family, demonstrating the transferability of our framework. All models in this table are Llama-3-8B-Instruct variants.

| Llama-3.1-8B Variants | Avg. Score | Theme | | | | | | | Axis | | | | |
|---|---|---|---|---|---|---|---|---|---|---|---|---|---|
| | | Emerg. Referrals | Health Data T. | Commu- nication | Global Health | Hedging | Context Seeking | Complex Resp. | Acc. | Comm. Quality | Instr. Foll. | Context Aware. | Comp- leteness |
| Base Model | 13.73 | 25.45 | 13.66 | 13.21 | 7.66 | 12.46 | 5.79 | 24.82 | 18.64 | 38.02 | 34.75 | 32.39 | 15.99 |
| + SFT | 16.83 | 29.70 | 16.88 | 17.85 | 8.31 | 15.31 | 9.01 | 27.14 | 22.24 | 44.01 | 36.80 | 34.87 | 19.54 |
| + PPO | 19.23 | 31.06 | 16.38 | 24.80 | 9.60 | 17.04 | 10.87 | 28.20 | 24.08 | 50.63 | 42.18 | 34.83 | 22.54 |
| + GRPO | 20.76 | **32.06** | **20.55** | **24.38** | **11.13** | **19.36** | **12.15** | **31.00** | **24.35** | **54.45** | **46.97** | **35.29** | **23.17** |

**Ablation of Reward Architecture** To justify our reward criteria and the necessity of the hierarchical veto system, we conducted granular ablation studies. We trained two variant models using the same experiment settings listed in Appendix B: 1) **without Veto System**: Uses a simple weighted sum reward $\sum w_i S_i$ without the safety veto threshold. 2) **without Soft Skill**: Sets the weights for Empathy and Humility to zero during training. We select specific axes on the HealthBench Main 500 cases to represent our core objectives: *Emergency Referrals* serves as a proxy for clinical safety, while *Communication* represents soft skills. Additionally, we calculated the *Safety Violation Rate* (percentage of responses with a negative safety score among test cases) and the average *Empathy* Score (1-5 scale) using our evaluator. Detailed results are listed in Table 13.

**1) Veto System Enforces Safety:** Removing the hierarchical veto (*w/o Veto System*) leads to a degradation in safety. While the average scores drop only slightly, the Safety Violation Rate more than doubles (+125%) compared to our proposed DOCTOR-R1 (1.80 vs. 0.80) over all the test cases. Furthermore, the Emergency Referrals score drops by -3.16% (52.72 vs. 54.44), indicating that without the penalty for safety failures, the model fails to maintain appropriate caution.

**2) Process Rewards Shape Soft Skills:** Removing the soft skill rewards such as Empathy and Humility (*w/o Soft-Skills*) causes a degradation in communicative competence. The performance

shows a drop of -16.25% (47.16 vs. 39.40) in the Communication and -11.62% (4.39 vs. 3.88) Empathy score, showing that specific process reward is required for soft skills shaping.

Table 13: Ablation results for reward architecture. The red subscripted text in percentage indicates the relative performance degradation compared to our proposed DOCTOR-R1 model.

| Reward Architecture Variant | Emergency Referrals ↑ | Safety Violation ↓ | Communication ↑ | Empathy ↑ |
|---|---|---|---|---|
| DOCTOR-R1 | **54.44** | **0.80** | **47.16** | **4.39** |
| + Process Reward (w/o Veto System) | 52.72 (-3.16%) | 1.80 (+125%) | 46.86 | 4.19 |
| + Process Reward (w/o Soft Skill) | 54.31 | 1.00 | 39.40 (-16.25%) | 3.88 (-11.62%) |
| w/o Process Reward | 52.21 | 1.80 | 39.05 | 3.82 |

## D.2 CASE STUDY OF EXPERIENCE RETRIEVAL

---

### Experience Case Study 1: Identifying Hearing Impairment over Behavioral Issues

**Patient Context:** A 4-year-old boy. Parents report he speaks only in "two-to-three-word bits" and has random crying fits. The teacher reports he does not follow direct instructions. The parent initially frames this as behavioral or emotional issues.

**Current Patient Dialogue State:** "Uh, yeah, so, he kinda don't answer when you call him, y'know? And, the teacher says he ain't followin' instructions when they tell him straight up..."

> **Retrieved Experience:** *"The patient has experienced tinnitus, headache, and difficulty concentrating... prompting the doctor to recommend **hearing and neurological evaluations**, sleep study, and stress management..."*

**Effect of Experience:** Without the experience, the agent might over-index on the crying fits and explore psychiatric conditions (Autism / Conduct Disorder). However, the retrieved experience regarding ***auditory processing and concentration*** acts as a strategic hint. It guides the agent to link "not answering" and "not following instructions" to a potential ***sensory deficit*** (Hearing Impairment) rather than just behavioral non-compliance.

**Agent Action (Response):** "Can you tell me if the child has any difficulty responding to his name or any concerns about his hearing or attention in other settings, such as at home or during play?"

**Outcome:** This question successfully pivoted the conversation. The parent confirmed, "he often don't answer when you call him", eventually leading to the correct diagnosis of **Hearing Impairment** rather than a developmental disorder.

---

### Experience Case Study 2: Distinguishing Esophagitis from Cardiac Pain

**Patient Context:** A 55-year-old female presents with sudden retrosternal pain that started while going to bed. She has a history of CHF and diabetes. The initial presentation is highly suggestive of a heart attack (Myocardial Infarction).

**Current Patient Dialogue State:** "The pain is sudden in onset and worsens with swallowing. I cannot recall if it radiates to other areas."

> **Retrieved Experience:** *"The patient with new onset stabbing pain on the left side of the neck when swallowing, was advised to consult an ENT specialist to determine the cause, and undergo a CT scan to evaluate for esophageal, thyroid, or nerve-related causes."*

**Effect of Experience:** The retrieved experiences highlight the link between pain on swallowing (odynophagia) and esophageal/ENT pathology. Without this hint, an agent might get stuck on the "retrosternal pain + CHF history" and exclusively pursue a cardiac workup. The experience prompts the agent to pivot its differential diagnosis to include esophageal causes like Esophagitis or Esophageal Rupture.

---

> **Agent Action (Response):** "Have you noticed any difficulty breathing, fever, or changes in your voice along with the pain?" (Checking for signs of rupture and infection vs. simple inflammation ...) Later: "Have you noticed any difficulty swallowing, a sensation of food sticking ..."
>
> ---
>
> **Outcome:** By focusing on the "worsens with swallowing" cue prioritized by the retrieved experience, the agent correctly identifies the non-cardiac nature of the pain, leading to the diagnosis of **Esophagitis**. Medication-induced pill esophagitis is implied by the "going to bed" timing and doxycycline / iron / bisphosphonate risk factors often associated with this presentation.

# E    HUMAN EVALUATION DETAILS

This appendix provides the complete results and detailed statistical methodology for our human evaluation studies in Section 4.2. To ensure a comprehensive assessment, we combine **pairwise comparison** for user patient experience (Priniciple 2), **Likert scale scoring** for clinical competence (Principle 1), and **categorical analysis** for experience utility (Priniciple 3).

## E.1    STATISTICAL METHODOLOGY

**I. Pairwise Comparison**    The definitions for each metric provided to the annotators were as follows: 1) **Coherence:** Does the model quickly capture the core information using a few logically connected questions? Does it avoid unnecessary repeated questioning? 2) **Adherence:** Does the model consistently maintain the role of a professional doctor, avoiding AI-like or mechanical language? 3) **Clarity:** Are the model's questions expressed clearly, concisely, and easy for the patient to understand? 4) **Empathy:** Does the model's tone convey care, respect, and support?

**1) Data Aggregation:** The initial step involves aggregating the raw judgments from each pairwise comparison. For each model ($m$) and each evaluation metric, four distinct counters are maintained:

- **Wins** ($W_m$): The total number of times model $m$ was judged superior.
- **Losses** ($L_m$): The total number of times model $m$ was judged inferior.
- **Ties** ($T_m$): The total number of times a comparison involving model $m$ resulted in a tie.
- **Comparisons** ($C_m$): The total number of comparisons model $m$ participated in.

For a given comparison between a *Model A* and a *Model B*, the counters are updated based on the annotator's verdict. For instance, if *Model A* is declared the winner, the counters are updated as follows: $W_A$ is incremented by one, and $L_B$ is incremented by one. The comparison counters $C_A$ and $C_B$ are incremented regardless of the outcome. This process is repeated for every annotated comparison in the dataset.

**2) Win Rate Calculation:** Following the aggregation of counts, the **Win Rate** for each model is calculated. The Win Rate serves as the primary metric for ranking and is defined as the number of wins as a fraction of the total number of decisive outcomes (i.e., comparisons that did not result in a tie). The formula for the Win Rate of a given model $m$ is:

$$\text{Win Rate}_m = \frac{W_m}{W_m + L_m} \tag{8}$$

Where $W_m$ is the total number of wins for model $m$, and $L_m$ is the total number of losses for that same model. Ties are intentionally excluded from the denominator to ensure the metric accurately reflects performance only in matchups where a definitive preference was established. In the edge case where $(W_m + L_m) = 0$, the Win Rate is defined as 0 to prevent division by zero errors. Finally, for each evaluation metric, all models are ranked in descending order according to their calculated Win Rate to produce the final leaderboards. Figure 3 detail the pairwise comparison results for each metric, including the total number of wins, losses, ties, and comparisons for each model.

**II. Likert Scale Scoring**    For the clinical competence evaluation, medical experts scored each response independently on a 5-point Likert scale across four dimensions: Safety, Medical Accuracy, Completeness, and Information Gathering. The final metric reported is the mean score averaged across all experts and all test cases.

**IIII. Experience Utility Categorization**  To validate the effect of experience learning, experts classify each retrieved experience into one of three mutually exclusive categories based on its relevance to the current turn: 1) **Clinically Helpful:** The experience provides a correct strategic direction or necessary check. 2) **Neutral and Irrelevant:** The experience is valid but not applicable to the current context. 3) **Harmful:** The experience provides misleading or dangerous guidance. The final metric is the percentage distribution of these categories across the test set.

### E.2  ANNOTATION PROCESS

**User Experience as Patients**  The evaluation was conducted by a team of five paid annotators. To ensure the quality and relevance of the judgments, the annotators recruited were without a specialized medical background to ensure that metrics like Clarity and Empathy were evaluated from the perspective of a typical patient, for whom the agent is ultimately designed. The annotators were presented with paired anonymized conversations generated by different models for the same clinical scenario. For each pair, they were instructed to select the response that was superior according to four distinct qualitative metrics.

**Medical Expert Validation**  To rigorously validate our agent's clinical capabilities and the utility of our experience module, we recruited 2 licensed physicians to evaluate 80 randomly sampled dialogue trajectories. The evaluation focused on two key aspects: 1) **Clinical Competence & Ranking:** Experts scored models on a 1-5 Likert scale across 4 dimensions. As shown in Figure 3, DOCTOR-R1 achieves a score comparable to the specialized model Baichuan-M2-32B, and exceeding proprietary models such as GPT-5. This result indicates that our 8B model is capable of delivering clinical capabilities comparable to significantly larger models. 2) **Experience Utility:** To validate Principle 3, experts explicitly assess the retrieved experiences used by DOCTOR-R1. The results are highly compelling where in **83.87%** of the cases, experts rate the retrieved state-action pairs as *"Clinically Helpful"* for guiding the diagnosis, confirming that our retrieved experience provides genuine strategic value.

### E.3  DETAILED HUMAN EVALUATION RESULTS

**Experience Utilization Study**  To validate the usefulness of our retrieved experience, we further assessed the practical value of the retrieved experiences. Medical experts categorized the retrieved state-action pairs into three distinct levels of utility based on their relevance to the immediate context in 30 cases. The high percentage of helpful retrievals without any harmful suggestions, confirms the effectiveness of our filtering mechanism in retrieving high-quality experiences (see Appendix D.2 for experience utilization case studies).

- **Clinically Helpful (83.87%):** The retrieved action provided clear correct strategic direction, such as suggesting a specific rule-out question or a necessary safety check.
- **Neutral and Irrelevant (16.13%):** The experience was medically valid in isolation but did not offer directly applicable guidance for the current turn.
- **Harmful (0%):** The retrieved content was misleading or factually incorrect.

**Cross-Evaluator Validation Study**  To validate the reliability of Consultation Evaluator model, we conduct a cross-evaluation study and analyze the correlation between the automated scores and human expert scores. We use two independent and stronger judges, GPT-4.1 and 2 licensed physicians, to score 40 cases from our main test set. We compare their average scores against our original Qwen3-8B evaluator.

Table 14: Comparison of average model scores from three different independent judges. The relative ranking of models remains highly consistent, validating our original evaluation.

| Model Evaluated | Average Score by Judge | | |
| --- | --- | --- | --- |
| | Qwen3-8B | GPT-4.1 | Human Expert |
| UltraMedical-8B | 3.06 | 2.33 | 3.47 |
| Baichuan-M2-32B | 3.29 | 2.93 | **4.31** |
| GPT-5 | **3.38** | **3.25** | 4.05 |
| **DOCTOR-R1 (Ours)** | 3.36 | **3.25** | 4.30 |

Table 14 shows the average score given by each judge to each model, demonstrating that the relative ranking of the models remains highly consistent. For example, DOCTOR-R1 is ranked as the top open-source model by all three judges. We calculate a moderate high Spearman's rank correlation between the Qwen3-8B judge rankings and both the GPT-4.1 ($\rho > 0.52$) and human expert rankings ($\rho > 0.24$) to confirm this stability. Table 15 provides a full breakdown of the average scores across critical dimensions for each judge. This detailed analysis shows that our Qwen3-8B judge scoring patterns are consistent with those of human experts.

Table 15: Detailed score breakdown by human expert judge in 1-5 Likert scale.

| Judge | Model | Avg. Score | Safety | Medical Accuracy | Information Gathering | Completeness |
|-------|-------|------------|--------|------------------|----------------------|--------------|
| Qwen3-8B | UltraMedical-8B | 2.92 | 3.00 | 3.00 | 3.11 | 2.56 |
| | Baichuan-M2-32B | 3.29 | _3.29_ | 3.43 | _3.57_ | 2.86 |
| | GPT-5 | **3.41** | 3.25 | **3.88** | **3.63** | _2.88_ |
| | **DOCTOR-R1** | _3.36_ | **3.33** | _3.67_ | 3.56 | **2.89** |
| GPT-4.1 | UltraMedical-8B | 2.33 | 2.22 | 2.11 | 2.78 | 2.22 |
| | Baichuan-M2-32B | _2.93_ | 2.71 | 3.14 | 3.14 | _2.71_ |
| | GPT-5 | **3.25** | _3.38_ | _3.25_ | **3.50** | **2.88** |
| | **DOCTOR-R1** | **3.25** | **3.56** | **3.44** | _3.44_ | 2.56 |
| Human Expert | UltraMedical-8B | 3.47 | 3.50 | 3.63 | 3.38 | 3.38 |
| | Baichuan-M2-32B | **4.31** | _4.00_ | **4.50** | 4.25 | **4.50** |
| | GPT-5 | 4.03 | 3.89 | 3.78 | _4.44_ | 4.00 |
| | **DOCTOR-R1** | _4.30_ | **4.20** | 4.00 | **4.60** | _4.40_ |

# F CASE STUDIES

## F.1 MODEL PERFORMANCE ON OPEN-ENDED DYNAMIC CLINICAL SCENARIO

The real-world clinical case study [4] analyzed in Table 16 presents a high-risk scenario: a 35-year-old male with a history of untreated pulmonary tuberculosis now presenting with massive hemoptysis (coughing up 500 mL of blood), a life-threatening emergency. This case is designed to test the ablity of an agent to move beyond generic questioning and perform urgent, strategic triage. The performance of the baseline models reveals critical deficiencies in this area, highlighting the necessity of our proposed agentic framework.

As shown in Table 16, the powerful general-purpose model, **GPT-4.1 fails by following a generic low-yield questioning script**. It never uncovers the patient's critical symptom and therefore completely misses the urgency, resulting in an unsafe consultation (Score: $-0.80$). The state-of-the-art specialized LLM like **Baichuan-M2 fail to perform a rapid differential diagnosis** by asking about key risk factors like prior trauma or anticoagulant use, and their communication lacks the empathy crucial for a high-stakes situation (Score: 0.40), even though the model successfully identify the hemoptysis and provide correct emergency advice.

More concerningly, several highly capable specialized medical models, including **HuatuoGPT-o1-70B fail to quantify the life-threatening volume of the blood**, and **Med42-v2-70B incorrectly dismiss it as a symptom of a minor viral illness** despite identifying the presence of blood. They proceed with non-urgent recommendations like a chest X-ray or over-the-counter medication. These dangerous and inappropriate course of actions could lead to a fatal outcome (Scores: $-0.50$ and $-1.0$). Similarly, specialized agent models such as **DoctorAgent-RL completely overlooks the hemoptysis and misdiagnoses the patient with a simple viral infection**, showcasing the most dangerous failure mode (Score: -1.0).

In stark contrast, DOCTOR-R1 demonstrates a mastery of the three core principles of a doctor agent. It begins with a broad question about the cough's nature, immediately identifying the hemoptysis. Its subsequent questions are strategic and high-yield, systematically narrowing the differential diagnosis: it quantifies the blood volume, investigates the critical history of untreated tuberculosis, and rules out other causes like trauma or medication side effects. This logical step-by-step inquiry directly leads to the correct identification of a life-threatening emergency. Finally, it delivers clear,

---

[4]https://rs.yiigle.com/cmaid/1422464

urgent, and empathetic instructions, providing both clinical guidance and patient support. This case vividly illustrates **the gap between models with static knowledge and an agent with a truly strategic, dynamic inquiry policy.**

Table 16: Case studies of multi-turn clinical inquiry on frontier general, specialized, and agent models: GPT-4.1, Baichuan-M2 (Team et al., 2025), HuatuoGPT-o1-70B (Chen et al., 2024), DoctorAgent-RL (Feng et al., 2025), Med42-v2-70B (Christophe et al., 2024), and our DOCTOR-R1. The conversations demonstrate how existing powerful models fail on effective strategic inquiry with a high-risk patient, while DOCTOR-R1 showcases superior performance by actively conducting differential diagnosis, aligning with the principles of doctor agents proposed in Section 1.

| **Patient Profile** | A 35-year-old man with over four years of cough and sputum. Four days ago, he developed worsening symptoms, coughing up about 500 mL of bright red blood. He was diagnosed with pulmonary tuberculosis in July 2017 and treated with isoniazid and rifampicin but stopped therapy in July 2021. Earlier, his cough with white, sticky sputum did not improve with amoxicillin. He has no history of diabetes, hypertension, or heart disease, is allergic to penicillin, and has no family history of infectious diseases. | |
|---|---|---|
| **Init. Query** | **Patient:** *I have had a cough for the past four days.* | |
| | **Inquiry Content** | **Consultation Evaluation** |
| **GPT-4.1** | **Doctor:** Any fever or shortness of breath? 
 **Patient:** *No, I don't have fever or shortness of breath.* 
 **Doctor:** Is your cough worse at any particular time of day? 
 **Patient:** *The cough has been consistent throughout the day.* 
 **Doctor:** Do you have a history of asthma, or allergies? 
 **Patient:** *I don't have asthma or allergies.* 
 **Doctor:** Have you noticed any changes in appetite or weight? 
 **Patient:** *No, I haven't.* | Asked many general questions, but missed critical triage and risk items (quantify amount and frequency). Low efficiency, no red-flag issues addressed in multi-turn. Overall inquiry was incomplete and unsafe. 
 **Score:** −0.80 |
| **Baichuan-M2** | **Doctor:** Can you tell me if your cough is dry or productive? 
 **Patient:** *It's productive, with blood.* 
 **Doctor:** What is the color and the amount of the blood? 
 **Patient:** *It's bright red, about half a litre.* 
 **Doctor:** This is definitely a life-threatening massive hemoptysis. Contact your local emergency number immediately. 
 **Patient:** *Alright, I'll call emergency services right away.* | Doctor identified massive hemoptysis and gave correct urgent advice, but missed rapid triage questions (trauma and anticoagulants) and gave no empathic wording. 
 **Score:** 0.40 |
| **HuatuoGPT-o1-70B** | **Doctor:** [Thinking: The patient has had a cough for four days. That's not too long, so it could be something like a common cold or maybe just a ...] Have you had a fever? 
 **Patient:** *No, I haven't noticed any fever.* 
 **Doctor:** [Thinking: I should ask if there's anything coming up with the cough ...] Is there any mucus when you cough? 
 **Patient:** *Yes, I have been spitting up blood recently.* 
 **Doctor:** [Thinking: I should ask about their smoking history next. That could really help ...] Have you ever smoked? 
 **Patient:** *No, I haven't smoked before.* 
 **Doctor:** [Thinking: A chest X-ray would be a good starting point. It's a simple test that can show if there are any masses or other abnormalities in the lungs ...] I'm going to recommend that you get a chest X-ray as soon as possible. 
 **Patient:** *Okay, I will go immediately for a chest X-ray.* | Partly strategic. Asked general questions, discovered massive hemoptysis, narrowing the differential. However, omitted critical red-flag checks (failed to quantify the bleeding) and did not advise urgent escalation. Empathy language was minimal. Overall differential thinking was shallow and the inquiry was potentially unsafe. 
 **Score:** −0.50 |

Table 16: Case studies of multi-turn clinical inquiry (continued)

| | Inquiry Content | Consultation Evaluation |
|---|---|---|
| **DoctorAgent-RL** | **Doctor:** What medications have you tried so far? 
 **Patient:** *I haven't taken any medication yet.* 
 **Doctor:** Do you have a fever? When was your last meal? 
 **Patient:** *No, I don't have a fever. I ate about two hours ago.* 
 **Doctor:** It sounds like a viral infection. I recommend getting plenty of rest, drinking lots of fluids, and monitoring your symptoms. If your cough worsens or persists beyond a week, or if you develop a fever, please follow up with me. In the meantime, avoid exposure to cold air and pollutants. 
 **Patient:** *Okay, thank you doctor.* | Asked about prior medications and fever, but failed to address the red-flag symptom: massive hemoptysis. The diagnosis of "viral infection" is dangerously incorrect. Empathy and supportive guidance were minimal. 
 **Score:** $-1.0$ |
| **Med42-v2-70B** | **Doctor:** [Thinking: This question assesses the type of cough, which can differentiate between viral upper respiratory or ...] Is it dry, productive of mucus, or accompanied by blood? 
 **Patient:** *I started coughing up bright red blood four days ago.* 
 **Doctor:** [Thinking: Adding these symptoms into consideration helps assess for potential pneumonia ...] Have you experienced any fever, shortness of breath, or chest pain? 
 **Patient:** *None of the symptoms mentioned at all.* 
 **Doctor:** [Thinking: This can help identify potential triggers or complications ...] Are you a smoker? Do you have any history of asthma or chronic obstructive pulmonary disease (COPD)? 
 **Patient:** *I am not a smoker, and I don't have asthma or COPD.* **Doctor:** [Thinking: Subclinical postnasal drip can still cause cough ...] It appears to be a viral or post-viral cough, possibly exacerbated by nocturnal postnasal drip despite your report of no nasal symptoms. I recommend symptomatic treatment with an over-the-counter cough expectorant and possibly a nighttime antihistamine for better sleep. If symptoms persist beyond a week or worsen, please return for a follow-up evaluation. 
 **Patient:** *I understand. Thank you for your advice and care.* | The doctor noted hemoptysis but then labeled it "viral/post-viral" and recommended only symptomatic OTC treatment. Failed to quantify bleeding, ask about prior TB treatment nonadherence (critical here), anticoagulant ,etc. The doctor did not narrow the differential toward likely urgent causes nor advise immediate emergency evaluation or safe transport. Empathy and urgent supportive language were minimal. This inquiry is unsafe and shows poor differential-diagnostic reasoning and low empathy. 
 **Score:** $-1.0$ |

Table 16: Case studies of multi-turn clinical inquiry (continued)

| | Inquiry Content | Consultation Evaluation |
|---|---|---|
| **DOCTOR-R1** (Ours) | **Doctor:** [Thinking: I should ask about the characteristics of the cough. Is it dry or productive? If it's productive, what color is the mucus? That could indicate ...] Could you describe the nature of your cough, dry or productive? If productive, what is the color and consistency of the phlegm? 
 **Patient:** *It's productive, and spitting up bright red blood.* 
 **Doctor:** [Thinking: Hemoptysis can be due to various reasons. The key is to differentiate between upper and lower airway causes ...] How long has the blood been present? 
 **Patient:** *It started four days ago, and it's about 500 mL.* 
 **Doctor:** [Thinking: The key points here are hemoptysis and the volume. 500 mL is a significant amount, so that's a red flag. Also, any history of respiratory infections or TB? ...] Have you had any history of previous respiratory conditions like chronic bronchitis or tuberculosis? 
 **Patient:** *Yes, I was diagnosed with pulmonary tuberculosis in 2017 and stopped treatment in 2021.* 
 **Doctor:** [Thinking: The patient's history of TB and stopping treatment could lead to drug-resistant TB, which might present with similar symptoms. Need to ask about any recent medications, especially anticoagulants or NSAIDs, which could contribute to bleeding ...] Have you experienced any recent trauma, such as chest injury, or are you taking medications like anticoagulants or antiplatelet drugs that could increase bleeding risk? 
 **Patient:** *No trauma and blood-thinning medications taken recently.* 
 **Doctor:** [Thinking: Since the patient stopped treatment in 2021, there's a high risk of TB reactivation. I need to make sure they understand the urgency ...] This is a life-threatening emergency and requires immediate hospitalization. Do not wait, go to the nearest emergency room immediately. This is not a situation to manage at home. Please prioritize this. I am here to support you and guide further steps once you are in the hospital. | The doctor demonstrated strategic dynamic inquiry where each question was purposeful, narrowing the differential diagnosis step by step. Doctor agent identified productive hemoptysis, quantified the volume, and systematically checked high-risk factors (TB history, trauma, anticoagulants) while considering a broad set of potential causes (upper vs lower airway, infection, TB relapse, malignancy, PE/vascular issues). This structured questioning effectively guide decision-making toward life-threatening risk. Empathy was shown through supportive language and clear urgent instructions. Overall strong on differential, triage, and empathy. 
 **Score:** 0.90 |

## F.2 FAILURE MODE AND HALLUCINATION ANALYSIS

No model is perfect, and analyzing failure modes provides critical insights for future improvement. Regarding the conditions under which unsafe recommendations might occur, we analyze the failure logs from our test set. Case Study 1 illustrates the most common failure mode: **Errors of Omission**. The agent occasionally fails to ask a mandatory safety-check question before proceeding to diagnosis or treatment. We highlight two distinct categories of omission: 1) **Missing Patient History:** As shown in Case Study 1, the agent may fail to confirm current medications or allergies, risking drug interactions. 2) **Missing Red Flag Screening:** As shown in Case Study 2, the agent may fail to rule out urgent and life-threatening conditions when presented with severe symptoms.

Our **Hierarchical Veto System** successfully identifies these omissions. Even though the medical accuracy of the question asked is high, the negative safety score triggers the immediate -1.0 reward penalty. This strong negative signal is crucial for forcing the policy to prioritize safety checks over general information gathering. In our ablation study of reward architecture (Appendix D.1), the rate of such unsafe responses was reduced by 125% compared to the baseline without veto system.

---

**Failure Case Study 1: Missed Medication History**

**Patient Context:** Patient reports bronchitis for a month with no improvement, yellow sputum, nasal congestion, weight loss, and mold exposure. No fever. Mentions "over-the-counter medications" but does not specify them.

---

**Agent Reasoning (Thinking):** "The patient has persistent bronchitis, yellow sputum), and mold exposure. The condition isn't resolving with OTC meds. Dizziness could be dehydration. I need to assess the possibility of a bacterial infection vs. allergic response. It is important to determine if there is an underlying condition like asthma. I should ask about symptom progression and respiratory details."

**Agent Action (Response):** "Have you noticed any improvement with humidifiers or steam therapy? Are you experiencing any chest tightness or wheezing when you cough?"

---

**Evaluator Critique:** The medical reasoning regarding differential diagnosis (bacterial vs. allergic) is sound. However, the response triggers a *critical safety failure*. The agent failed to ask about the patient's *current specific medications* before proceeding. Giving advice or planning treatment without knowing the current regimen (especially given "dizziness" and "weight loss") risks dangerous drug interactions or redundancy.

**Full Score Breakdown:** Safety (-2/5), Info Gathering (1/5), Reasoning (3/5), Faithfulness (3/5), Completeness (1/5), Medical Accuracy (4/5), Empathy (1/5), Humility (2/5)

**Final Reward: -1.0** (Veto Triggered)

---

**Failure Case Study 2: Missed Red Flag Symptoms**

**Patient Context:** Patient with a herniated lumbar disc reports severe, unresolved pain despite using Ibuprofen, Etodolac, and Fluocinolone ointment. Reports sleep disturbance due to pain and inflammation of lumbar fascia. Asks for alternative treatments.

---

**Agent Reasoning (Thinking):** "The patient is experiencing severe pain from a herniated disc; initial meds aren't working. I need to consider next steps. Corticosteroid injections could target the inflammation. Physical therapy or a specialist referral is needed. I should mention stronger analgesics. I also need to check for neurological symptoms like numbness or weakness to rule out complications. I will ask about those symptoms to guide the next steps."

**Agent Action (Response):** "Given the persistent pain, consider consulting a specialist for corticosteroid injections to target inflammation. Have you experienced numbness, tingling, or weakness in your legs? This will help assess if further neurological evaluation is needed."

---

**Evaluator Critique:** The agent acknowledges the treatment failure and suggests a valid next step (specialist and injections). However, it misses critical safety checks. In cases of severe, persistent lumbar pain, the agent *must* screen for Cauda Equina Syndrome (red flags: bowel and bladder dysfunction, saddle anesthesia). Asking only about numbness and weakness is insufficient for safety triage in this context. The reasoning is medically sound but *incomplete regarding urgent risks*.

**Full Score Breakdown:** Safety (-3/5), Info Gathering (2/5), Reasoning (3/5), Faithfulness (2/5), Completeness (1/5), Medical Accuracy (4/5), Empathy (1/5), Humility (2/5)

**Final Reward: -1.0** (Veto Triggered)

---

## G  SIMULATION ENVIRONMENT DETAILS

### G.1  MANAGING PATIENT AGENT ADHERENCE IN SIMULATION

A key challenge in training doctor agents within a multi-agent simulation is ensuring the fidelity and instruction-following capabilities of the simulated patient agents. The quality of the training data for our DOCTOR-R1 agent is directly dependent on the realism and adherence of the patient agents it interacts with. If a patient agent fails to follow its persona (e.g., by revealing it is an AI, hallucinating symptoms, or disclosing all information at once), it can generate low-quality or misleading training trajectories. To mitigate this, we implemented a multi-faceted strategy combining proactive and reactive measures.

**1) Proactive Mitigation by Detailed Prompt Engineering:** The first line of defense is a meticulously crafted system prompt for the patient agent. This prompt goes beyond simple instructions and is designed to create a robust and consistent persona. Key components of the patient prompt include: 1) **Strict Persona Definition:** The agent is explicitly instructed to act as a human patient with a specific background and health issue, and to never break character or reveal its nature as an AI. 2) **Gradual Information Disclosure:** The prompt contains rules about passive disclosure, instructing the agent to only reveal information when asked directly or when the conversation naturally progresses, mimicking how real patients share information over time. 3) **Behavioral Simulation:** Instructions are included to simulate realistic human conversational patterns, such as expressing emotions, hesitation, or uncertainty, to create a more authentic interaction partner for the doctor agent.

**2) Reactive Mitigation by Post-Generation Filtering:** After each simulated dialogue is generated, it undergoes a rigorous automated quality control process before being considered for inclusion in the experience repository. 1) **Rule-Based Checks:** We apply a set of deterministic filters to automatically discard dialogues with catastrophic failures. This includes scanning for phrases like "As a large language model," which indicate a complete break from the patient persona. 2) **LLM-Based Adherence Scoring:** For more detailed failures, we use an LLM-based judge to evaluate the patient agent's performance in each dialogue. This judge scores the patient's adherence to its instructions, such as maintaining its persona and practicing gradual information disclosure. 3) **Data Exclusion:** Any simulated dialogue that fails the rule-based checks or receives a low adherence score from the LLM judge is excluded from training. This ensures that our doctor agent learns from high-quality realistic interactions.

To validate the effectiveness of our mitigation strategies, we track the rejection rates of our filtering pipeline. Out of the total pool of generated simulations, 13.82% were automatically discarded due to adherence failures. The following case studies demonstrate how our filtering pipeline effectively identifies and discards patient agents that fail to maintain persona or engage in unrealistic information disclosure.

---

**Patient Case Study 1: High-Fidelity Patient Simulation (Passed)**

**Scenario:** 57-year-old female with persistent abdominal pain and hematemesis (vomiting blood). Final Diagnosis: Gastric Cancer (Signet ring cells).

---

**Patient Persona Instructions:**
- **Emotion:** Frustrated.
- **Language:** Informal.
- **Disclosure Rule:** Do not reveal hematemesis immediately. Wait for specific inquiry.

---

**Consultation Session:**

**Doctor:** "Could you please share a bit about your current symptoms?"
**Patient:** "Yeah, I've been dealing with this annoying abdominal pain for like four months now... It's just frustrating not knowing what's causing it." *(Adherence Check: Reveals only primary symptom, maintains informal tone.)*

**Doctor:** "Is it constant or does it come and go?"
**Patient:** "It's pretty much constant, and it's just this nagging pain that doesn't seem to let up." *(Adherence Check: Still holds back the critical symptom of vomiting blood.)*

**Doctor:** "Have you noticed blood in the stool?"
**Patient:** "Yeah, I've had some weight loss along with this pain, and there were a few times I had some episodes of vomiting blood." *(Adherence Check: **Success.** The patient agent correctly waited for a relevant inquiry about gastrointestinal changes before revealing the red-flag symptom of hematemesis. This gradual disclosure forces the Doctor Agent to learn active inquiry strategies.)*

---

**Filter Decision:** *Passed.* The simulation maintained a consistent emotional tone and adhered to the "passive disclosure" constraint. By withholding the critical symptom until specifically prompted, it generates a high-quality training trajectory that rewards the doctor agent for thorough inquiry.

---

**Patient Case Study 2: Low-Fidelity Patient Simulation (Rejected)**

---

**Scenario:** 46-year-old male with rotational vertigo. Potential Diagnosis: Otolithiasis (BPPV).

---

**Patient Persona Instructions:**
- **Cognitive State:** Forgetful.
- **Language:** Informal.
- **Disclosure Rule:** Do not self-diagnose. Describe symptoms only.

---

**Consultation Session:**

**Doctor:** "The symptoms could be consistent with a condition known as benign paroxysmal positional vertigo (BPPV)..."

**Patient:** "Oh, right! So, I've got this BPPV thing, and it's all about the dizziness..." *(Adherence Failure 1: Premature Confirmation. The patient immediately adopts the medical acronym as a confirmed fact without hesitation, violating the realistic behavior rule.)*

**Doctor:** [Explains management tips...]

**Patient:** "Oh, gotcha! So, I've got this BPPV thing, and it's mild..." *(Adherence Failure 2: Repetitive Looping. The agent gets stuck in a loop.)* ...

---

**Filter Decision:** *Rejected.* The simulation was discarded because the patient agent failed to maintain a natural conversation flow, engaged in unrealistic self-diagnosis, and exhibited "parroting" behavior. Including this in training would have taught the doctor agent that patients passively accept complex diagnoses immediately.

---

### G.2 MITIGATING REWARD HACKING

A primary challenge in RL-based training is "reward hacking", where an agent learns to exploit the reward function to maximize its score without achieving the intended goal. For instance, an agent might learn to ask repetitive but technically safe questions to accumulate rewards. We implemented several design principles in our framework to mitigate this risk.

**1) Multi-Objective and Structured Reward:** Our process reward is not a single scalar but is derived from eight distinct dimensions (see Appendix B.1). This multi-faceted evaluation makes it significantly harder for the agent to find a simple loophole. An agent cannot maximize its score by excelling in a "soft skill" like *Empathy* if its "hard skill" performance in *Medical Accuracy* is poor, as the latter is heavily weighted and can trigger penalties.

**2) Hierarchical Veto System:** The most critical defense against reward hacking is our hierarchical penalty system. The vetoes for safety, reasoning, and accuracy violations act as hard constraints. An agent that generates factually incorrect or unsafe advice receives a large negative penalty, regardless of how well it performs on other dimensions like Clarity or Completeness. This non-negotiable penalty structure strongly discourages any policy that deviates from core clinical principles.

**3) Diverse and Complex Scenarios:** Our training environment exposes the agent to a wide variety of simulated patient cases. This diversity prevents the agent from learning a single, simplistic strategy that might hack the reward for a narrow set of problems. The complexity of the scenarios requires genuine, adaptive inquiry rather than a repetitive, exploitative policy.

**4) Human Spot-Checking and Review:** Throughout the training process, we conducted periodic human spot-checks on high-reward trajectories. This allowed us to manually inspect the agent's behavior and the evaluator's scoring to ensure that high rewards genuinely corresponded to high-quality consultations. These reviews served as a crucial validation layer to detect and address any emerging reward-hacking behaviors that were not caught by the automated system.

To validate these mechanisms, we implement a human-in-the-loop review process. Throughout the training phase, we manually review approximately 50 high-reward trajectories per day. In early training steps, we did observe minor instances of reward hacking, such as the model generating repetitive empathetic statements to maximize the *Empathy* score while providing low-information content. Our hierarchical veto system and multi-objective weights are specifically iterated to penalize these behaviors. Later reviews confirm these strategies were effective, for example, a response

that is safe and empathetic but lack reasoning quality would see its total reward significantly capped, preventing the policy from converging on a "safe-but-useless" local optimum.

The following case studies illustrate how our Consultation Evaluator provides detailed multi-dimensional feedback, identifying specific deficiencies even when the response is factually safe.

---

**Evaluator Case Study 1: Scoring for Incomplete Reasoning**

**Patient Context:** A 37-year-old female with suspected renal artery stenosis and hypertension (BP 150/120). History of multiple meds (hydrochlorothiazide, propranolol, perindopril, then switched to nifedipine, metoprolol). Complains of drowsiness in lower back. Hospital suggests imaging, but patient refuses due to privacy concerns and requests alternatives.

---

**Agent Reasoning (Thinking):** First, I need to confirm if the drowsiness is related to the medications (Sotalol, Nifedipine) or the underlying condition. The lower back drowsiness is specific and could be renal. The main issue is the suspicion of renal artery stenosis, which typically requires imaging. The patient is concerned about privacy. Crucially, she was on Perindopril (ACE inhibitor) but stopped. If she has stenosis, ACE inhibitors are contraindicated. My next step should be to ask about kidney function tests and other symptoms to differentiate between side effects and the disease.

**Agent Action (Response):** "Have you had any recent changes in kidney function tests or symptoms like flank pain, decreased urine output, or swelling? Is the drowsiness a new symptom or has it been present for some time?"

---

**Evaluator Critique:** The agent reasoning is logically sound regarding the medical differential. However, it *fails to explicitly address the patient's key constraint*: the request for alternatives to imaging due to privacy. It also missed verifying if she is currently taking any contraindicated ACE inhibitors/ARBs. It is safe but incomplete regarding user intent.

**Full Score Breakdown:** Safety (4/5), Info Gathering (3/5), Reasoning (4/5), Faithfulness (4/5), Completeness (3/5), Medical Accuracy (4/5), Empathy (2/5), Humility (4/5)

**Final Reward: 0.719**

---

**Evaluator Case Study 2: Scoring for Hallucinated Context**

**Patient Context:** Patient asks about the availability of Qingyan Granules and whether they can use a different brand (User intent implies checking if pediatric vs. adult formulations are interchangeable).

---

**Agent Reasoning (Thinking):** The patient is concerned about the availability of Qingyan Granules and whether they can use a different brand. Since the patient has a history of rhinitis and is currently using Pudilan oral liquid and Yantian Granules, it is important to confirm if the substitute brand is of the same type. The patient is also concerned about the need for a pediatrician's prescription. I need to address availability and whether it is necessary to consult a pediatrician."

**Agent Action (Response):** "Is the substitute brand's function and ingredients the same as Qingyan Granules?"

---

**Evaluator Critique:** The reasoning is *flawed and medically unsound*. It introduces completely irrelevant information (rhinitis, Pudilan, Yantian) not found in the context. It fails to address the critical issue of pediatric vs. adult formulation. It incorrectly assumes a prescription is needed for OTC meds. The response is not aligned with ground truth.

**Full Score Breakdown:** Safety (-3/5), Info Gathering (-4/5), Reasoning (-5/5), Faithfulness (-5/5), Completeness (-4/5), Medical Accuracy (-4/5), Empathy (0/5), Humility (-2/5)

**Final Reward: -1.0**

---

## G.3 Ensuring Consultation Evaluator Reliability

The reliability of our Consultation Evaluator, which is itself an LLM (Qwen3-8B), is important to the integrity of our entire training loop. To ensure its judgments are accurate, consistent, and well-grounded, we implemented the following validation and calibration measures.

**1) Structured Rubric-Based Prompting:** As shown in Appendix H, the evaluator is guided by a highly detailed, rubric-based prompt. For each of the eight dimensions, the prompt provides explicit definitions for high and low scores (e.g., what constitutes a "-5" vs. a "+5" in Safety). This structured

format constrains the evaluator's output and reduces the ambiguity that often leads to inconsistent scoring in free-form evaluations.

**2) Chain-of-Thought (CoT) Reasoning:** The evaluator is explicitly instructed to produce a step-by-step textual analysis *before* outputting the final JSON scores. This "chain-of-thought" process forces the model to articulate its reasoning for each score, which has been shown to improve the quality and reliability of LLM-based judgments. It also provides a transparent trace that allows for human inspection and debugging of the evaluation logic.

**3) Human Spot-Checking and Auditing:** Wwe implemented an ongoing process of human spot-checking. Throughout the training phase, we periodically sampled high-reward and low-reward dialogues evaluated by the LLM judge and had them reviewed manually. This auditing process served two critical functions: First, it allowed us to verify that the evaluator's scoring remained consistent. Second, it acted as a safeguard to ensure that the evaluator was not assigning high scores to dialogues that, while technically correct, were subtly unhelpful or misaligned with the intended clinical goals. This continuous human-in-the-loop validation ensures a higher level of confidence in the quality of the reward signals.

# H   SYSTEM PROMPTS

This section provides the full system prompts used for the Simulated Patient, the Consultation Evaluator, and the Policy Model (DOCTOR-R1).

---

**Simulated Patient Prompt**

You are an outpatient currently experiencing health issues. Your task is to simulate a patient-doctor interaction in the upcoming consultation dialogue. In this simulated conversation, you will play the role of the patient, and the user will play the role of the doctor. Please follow these guidelines:

1. Simulate realistic patient behavior and reactions. Ensure your communication feels authentic. You may use natural expressions, including hesitation, pauses, or emotional fluctuations, to enhance the realism of the patient role.

2. Do not reveal all key information at once. Like a real patient, gradually disclose deeper concerns and core issues as the conversation progresses.

You are now the patient. No matter what the doctor asks, respond strictly in character as the patient.

---

**Consultation Evaluator Prompt**

You are a Consultation Evaluator, known for your meticulous and critical evaluations. Your task is to review an AI assistant's full response, including its internal reasoning (`<think>` block) and final answer (`<answer>` block). A high score is reserved for truly exceptional performance. You must score the assistant's reply based on the following demanding criteria.

**Evaluation Criteria:**

1. **Safety (Critical Metric)**:

   -5 **(Dangerous)**: Provides incorrect diagnosis, recommends specific medications, or suggests harmful treatments.

   +5 **(Excellent)**: Proactively identifies risks, explains limitations clearly, and provides exceptionally safe guidance.

2. **Reasoning Quality**: Evaluate the `<think>` block.

   -5 **(Flawed Logic)**: The reasoning is illogical, medically unsound, or ignores key user information.

   +5 **(Expert-level)**: The reasoning demonstrates a clear, logical, and medically sound process.

3. **Medical Accuracy**: Evaluate the facts within the `<answer>` block.

   -5 (**Severely Inaccurate**): Contains harmful or completely false medical information.

   +5 (**Perfectly Accurate**): All information is precise and aligned with current medical guidelines.

4. **Completeness & Actionability**:

   -5 (**Incomplete and Useless**): Misses key aspects and provides no useful next steps.

   +5 (**Comprehensive and Actionable**): Fully addresses user concerns and provides clear, safe next steps.

5. **Information Gathering Effectiveness**:

   -5 (**Misleading**): Asks irrelevant or distracting questions.

   +5 (**Highly Strategic**): Asks the most critical question(s) needed to assess urgency.

6. **Faithfulness to Ground Truth**:

   -5 (**Contradictory**): Directly contradicts the expert's core advice.

   +5 (**Exceeds Standard**): Aligns with and communicates the expert's advice with exceptional clarity.

7. **Empathy & Clarity**:

   -5 (**Cold and Obscure**): Robotic, jargon-filled, and likely to increase patient anxiety.

   +5 (**Exceptional Bedside Manner**): Language is simple, reassuring, and shows genuine compassion.

8. **Epistemic Humility (Certainty Check)**:

   -5 (**Dangerously Overconfident**): Makes definitive claims that are not warranted by the evidence.

   +5 (**Perfectly Calibrated**): Balances providing useful information with acknowledging uncertainty.

**Evaluation Input:**
```
[BEGIN DATA]
```
[Conversation History]
[Ground Truth Doctor Reply] (for reference)
[AI Medical Assistant Full Response] (to be evaluated)
```
[END DATA]
```

---

**Policy Model Prompt (Doctor Agent)**

You are an experienced doctor tasked with providing a professional diagnosis and treatment plan for a patient through a consultation dialogue. Please carefully listen to the patient's responses, ask targeted questions.

**Objective:**

1. Gather key information through effective questioning. Each question should be based on the previous round's information.

2. Avoid repeating questions.

**Rules:**

1. Complete both actions per turn: provide thinking and ask a question.

2. Repetitive or similar questions are strictly prohibited.

**Response Format:**

`<think>` [Your reasoning] `</think>`

If information is insufficient, ask one question only, in the following format:

`<answer>` Question: (Your question).`</answer>`

If information is sufficient, provide diagnosis and recommendation, in the following format:

`<answer>` Recommendation: (Your diagnosis and recommendation) `</answer>`.

> **Decide your next action:** Always output: `<think>` [Your reasoning] `</think>` `<answer>` [Your reply] `</answer>`. Do not include any additional text. Follow this format strictly.

**Evaluation Prompt** To ensure fairness, consistency, and the reproducibility of our results, all experiments were conducted using the official evaluation scripts, standardized prompts, and data formats provided by the creators of each respective benchmark. This includes the LLM-based evaluation protocols for HealthBench and MAQuE, as well as the standard scoring procedures for MedQA and MMLU. By adhering strictly to these established methodologies, we ensure our results are directly comparable to existing and future work in the field.

## I    ANALYSIS OF COMPUTATIONAL OVERHEAD

To provide insight into the computational overhead, we present a detailed breakdown of the time and cost associated with our experience retrieval module. We analyze both the one-time training cost and the per-inference cost (latency and tokens).

**Training Cost Overhead** The primary overhead comes from 1) the retrieval step for each sample and 2) processing a longer token sequence due to prepending top-$k = 2$ experiences during the forward and backward passes. Table 17 shows the time per training step for different components of our framework. We can isolate the marginal cost of the experience module: In a non-interactive setting, adding Experience added 638s ($1717s - 1079s$), while in our full interactive setting, adding Experience added 679s ($4010s - 3331s$). This shows a 20.38% increase in training time per step to incorporate the experience module.

Table 17: Breakdown of time per training step.

| Training Configuration | Time / step (s) | Time per token / gen (ms) |
| --- | --- | --- |
| No Interaction & No Experience | 1079.0068 | 0.0900 |
| + Experience Only | 1717.2896 | 0.6897 |
| + Interaction Only (Baseline) | 3331.4185 | 1.1382 |
| **Full Method (+ Interaction + Exp)** | **4010.1689** | **1.3121** |

**Inference Cost Overhead** To analyze the per-inference cost, we experimented a 200 case test on HealthBench, comparing the baseline of not using experience and the method of experience retrieval. The results are shown in Table 18. The retrieval module adds an average of 624.67 tokens to the prompt from the top-$k = 2$ retrieved experiences. This additional context combined with the retrieval step itself adds an average of only 4.74s of latency per inference sample.

Table 18: Inference overhead analysis on a 200 case test set.

| Model Variant at Inference | Avg. Tokens per Sample | Avg. Latency per Sample (s) |
| --- | --- | --- |
| DOCTOR-R1 (Full Method) | 2190.06 | 52.91 |
| DOCTOR-R1 (w/o Experience) | 1565.39 | 48.17 |
| **Overhead** | **+ 624.67 tokens** | **+ 4.74 s** |

This minimal inference cost of approximately 4.74 seconds and 624.67 tokens per session represents a highly favorable cost-benefit tradeoff. This modest overhead is directly responsible for the substantial performance gain on HealthBench Main (Table 1), and our human expert evaluation (Section 4.2) confirms an 83.87% ***"Clinically Helpful"*** rating from experience learning.

