# OpenReview forum: "Doctor-R1: Mastering Clinical Inquiry with Experiential Agentic Reinforcement Learning"
_ICLR.cc/2026/Conference — ICLR 2026 Poster_

### Official Review · Reviewer_bTuJ · 2025-10-29

**Soundness:** 2
**Presentation:** 3
**Contribution:** 2
**Rating:** 4
**Confidence:** 5

**Summary:**

The paper proposes DOCTOR-R1, an AI agent framework designed to master both strategic multi-turn patient inquiry and medical decision-making through experiential reinforcement learning.

**Strengths:**

1. The paper identifies an important gap in medical AI research. The distinction between static medical knowledge assessment and dynamic clinical inquiry is clear.
2. The evaluation includes multiple dimensions: two dynamic benchmarks (HealthBench, MAQuE), static knowledge validation (MedQA, MMLU).

**Weaknesses:**

1. The most severe limitation is the pervasive use of LLM-as-judge throughout the system. The reward model is Qwen3-8B evaluating Qwen3-8B policy outputs. This creates a closed system where the system optimizes for LLM evaluator preferences rather than genuine clinical competence.
2. The experience retrieval mechanism offers minimal technical innovation. Stage 1 (embedding similarity + reward weighting) and Stage 2 (cross-encoder reranking) are standard retrieval techniques. Are retrieved experiences genuinely instructive, or does the system learn spurious correlations?
3. There are some Human Evaluation Limitations. No evaluation of diagnostic accuracy by medical professionals. Both primary benchmarks (HealthBench, MAQuE) use automated LLM evaluation, creating potential bias toward response styles favored by specific LLM families.
4. No comparison with supervised fine-tuning on high-quality human-authored dialogues. Given the complexity of the RL pipeline, demonstrating that RL provides advantages over simpler approaches is essential.
5. The author acknowledges the "key challenge" of patient agent adherence but provides insufficient evidence that the mitigation strategies are effective. What percentage of simulated dialogues are filtered out due to adherence failures?
6. The multi-objective reward structure, while helpful, does not eliminate the possibility of exploitation. Human spot-checking is mentioned but no statistics on frequency or findings are provided.

**Questions:**

1. Can you provide quantitative evidence that reward model judgments correlate with physician assessments? What is the inter-rater agreement between the LLM judge and medical experts?
2. What percentage of training dialogues are filtered due to patient agent adherence failures?
3. How does supervised fine-tuning on high-quality human dialogues compare to your RL approach?
4. Under what conditions does the system produce unsafe recommendations despite the veto mechanism?
5. What types of experiences does the system most frequently retrieve? Are there cases where novelty filtering excludes valuable high-quality examples?
6. How does performance degrade when evaluated on out-of-distribution cases (different patient populations, communication styles, or medical domains)?

---

> ### Author Response · Authors · 2025-11-24
>
> We are grateful to the reviewer for your thorough evaluation and encouraging remarks. We truly appreciate the recognition of our contributions and the insightful questions raised. ***We have carefully addressed all concerns and updated the corresponding revisions into the new PDF, with key changes highlighted in blue text for clarity.*** In the following, we respond point-by-point to each comment and provide additional explanations and experiments where necessary.
>
> - **Q1:** The most severe limitation is the pervasive use of LLM-as-judge throughout the system. The reward model is Qwen3-8B evaluating Qwen3-8B policy outputs. This creates a closed system where the system optimizes for LLM evaluator preferences rather than genuine clinical competence.
>
> **Reply to Q1:**  Thank you for your question. We agree that this is an important question to be investigated and validated, hence we address this concern in three ways: 1) Clarify **safeguards in training** design that mitigate bias, 2) Perform a new **evaluation study involving clinical experts** (detailed settings listed in ``Appendix E.3``), and 3) **Validate generalization**.
> 1. **Safeguards in Training:** First, the Consultation Evaluator operates on grounded opinion. As shown in our prompt (``Appendix H``), the judge is provided with the **[Ground Truth Doctor Reply]** and is explicitly instructed to score 'Faithfulness to Ground Truth' (Criteria 6). This anchors the evaluation to expert standards, making the task "How well does this response match the expert standard?" rather than "Do you, Qwen, like this response?". Furthermore, the evaluator's weights are frozen, preventing the policy and judge from co-adapting into a self-delusional loop.
> 2. **Evaluation Validation involving Human Expert:** To prove our metrics are not inflated by the Qwen-judge, we conducted a cross study (see the below Table A). We scored the test cases using two independent stronger judges: GPT-4.1 and a panel of licensed physicians. **1) Consistency:** While absolute scores vary by judge, the **relative rankings remain highly consistent.** As shown in ``Table 14`` and ``Table 15``, Doctor-R1 consistently outperforms baselines like UltraMedical-70B and Baichuan-M2-32B across all judges. **2) Reliability:** We found a moderate high Spearman's rank correlation ($\rho > 0.52, 0.24$) between the Qwen-judge's rankings with GPT-4.1 and human expert, confirming that our Qwen-based evaluator serves as a reliable proxy for generalized clinical quality.
> 3. **Validating Agent Generalization:** Too address the concern about generalizable clinical skills, our benchmark evaluation (OpenAI's HealthBench and MAQuE) in ``Table 1`` and ``Table 2``, are actually representing patient cases **Out-of-Distribution (OOD)**. While Qwen-based patient simulator is used in our training phase, GPT-based model is used as a patient simulator in our inference phase as per the official benchmark protocols. Therefore, our strong performance on these benchmarks serves as proof of generalization on clinical skills.
>
> Table 14: Comparison of average model scores from three different independent judges.
> | Model Evaluated          | Qwen3-8B | GPT-4.1 | Human Expert |
> |--------------------------|---------:|--------:|-------------:|
> | UltraMedical-8B          | 3.06     | 2.33    | 3.47         |
> | Baichuan-M2-32B          | 3.29     | 2.93    | **4.31**     |
> | GPT-5                    | **3.38** | **3.25**| 4.05         |
> | **Doctor-R1**            | _3.36_   | **3.25**| _4.30_       |
>
> ---
>
> - **Q2:** The experience retrieval mechanism offers minimal technical innovation. Stage 1 (embedding similarity + reward weighting) and Stage 2 (cross-encoder reranking) are standard retrieval techniques. Are retrieved experiences genuinely instructive, or does the system learn spurious correlations?
>
> **Reply to Q2:** We appreciate your concern. Our proposed experience module enables **continuous learning**, which allows the agent to update and learn from its experience repository with new cases without expensive retraining. This adaptability actually addresses the huge demand in real-world clinical settings. The innovation of our experience retrieval lies in its application within an **Agentic RL framework** and our **novelty + reward filtering mechanism** (stage 3) instead of RAG techniques. Our retrieval design enables the model to retrieve the *high-performing* (high reward score), *low-repetitive* (high novelty score), and *high-relevant* (high semantic similarity score) experience. While standard retrieval mechanisms typically optimize only for semantic similarity to retrieve the top- $k$ most similar items, often resulting in a set of **highly repetitive and low-utility** examples.
>
> (To be continued)

---

> ### Author Response · Authors · 2025-11-24
>
> (Continued)
>
> Furthermore, to study whether these experiences are useful and appropriate in real cases, we performed **Expert Validation** (``Appendix D.3``), **Retrieval Control Ablation** (Page 21, under paragraph *Isolating Experience Components*), and **Qualitative Case Study** (Appendix D.2), as our primary validation experiments.
> 1. **Medical Expert Validation:** We asked the experts to directly evaluate the clinical utility of our experience module. On 30 samples, experts were shown the retrieved experience and asked to rate its usefulness out of three distinct categories: 1) Clinically Helpful, 2) Neutral and Irrelevant, and 3) Harmful. The **experts agreed the experience was *"Clinically Helpful"* in 83.87% of cases** for guiding the agent's next action, with only 16.13% rated as "Neutral and Irrelevant" and 0% as "Harmful". This provides strong external validation that the retrieved experiences are *genuinely instructive*.
> 2. **Retrieval Control Ablation:** To scientifically isolate the *experiential learning* effect from mere in-context prompting, we evaluate with and without experience retrieval versions of our agent at inference time. The results show that method without experience retrieval performs -8.67% worse on Communication than our method using experience retrieval.
> 3. **Qualitative Case Study:** Detailed case studies further illustrate scenario where a high-value experience is retrieved and correctly utilized by the agent to make a superior strategic decision. (See ``Appendix D.2`` complete case study)
>
> ---
>
> - **Q3:** There are some Human Evaluation Limitations. No evaluation of diagnostic accuracy by medical professionals. Both primary benchmarks (HealthBench, MAQuE) use automated LLM evaluation, creating potential bias toward response styles favored by specific LLM families.
> - **Q7:** Can you provide quantitative evidence that reward model judgments correlate with physician assessments? What is the inter-rater agreement between the LLM judge and medical experts?
>
> **Reply to Q3 and Q7:** Thank you for your valuable suggestion. We recognized the importance of expert evaluation, hence we conducted a new **comprehensive expert evaluation study** last week, with 2 licensed physicians to address both concerns simultaneously. This study consisted of two main tasks:
>
> **Task 1: Validating our Agent's Clinical Skills**
> To address the valid concern of agent's clinical skills, we first asked the physicians to score 40 cases using the exact same four dimension rubric as our LLM-judge (``Appendix B.1``).
> 1. **Principle 1 (Strategic Inquiry):** The physicians conducted a 1-5 Likert scale scoring of full dialogues between Doctor-R1 and the strong baselines (including GPT-5, Baichuan-M2-32B and UltraMedical-70B). As shown in our new ``Figure 3`` (``Section 4.2``), Doctor-R1 outperforms most of the baseline models, especially in Safety and Information Gathering.
> 2. **Principle 2 (Empathy):** For the patient user experience evaluation, we recruited five annotators without specialized medical training to ensure that subjective metrics were judged from the perspective of a typical patient. The results in ``Figure 3`` show a strong human preference for our model, which ranked first across all four categories **(92.5% win rate on "Empathy")**.
> 3. **Principle 3 (Experience Usefulness):** We asked the experts to directly evaluate the clinical utility of our experience module. On 30 samples, experts were shown the retrieved experience and asked to rate its usefulness out of three distinct categories: 1) Clinically Helpful, 2) Neutral and Irrelevant, and 3) Harmful. The **experts agreed the experience was *"Clinically Helpful"* in 83.87% of cases** for guiding the agent's next action, with only 16.13% rated as "Neutral and Irrelevant" and 0% as "Harmful".
> This new expert-driven evaluation provides crucial, direct validation for our primary research contributions (Principles 1, 2 and 3).
>
> **Task 2: Validating our Consultation Evaluator**
> To validate the reliability of evaluator model, we conduct an **evaluation study involving clinical experts** (as elaborated in **[Reply to Q1]**) and analyze the correlation between the automated scores and human expert scores. We use two independent and stronger judges, GPT-4.1 and human expert, to score 40 cases. ``Table 14`` and ``Table 15`` (``Appendix E.3``) shows the score given by each judge, demonstrating that the relative ranking of the models remains **highly consistent**. Also, the Spearman's rank correlation calculated confirm the evaluation **high stability and reliability** between judges.

---

> ### Author Response · Authors · 2025-11-24
>
> - **Q4:** No comparison with supervised fine-tuning on high-quality human-authored dialogues. Given the complexity of the RL pipeline, demonstrating that RL provides advantages over simpler approaches is essential.
> - **Q9:** How does supervised fine-tuning on high-quality human dialogues compare to your RL approach?
>
> **Reply to Q4 and Q9:** Thank you for your valuable suggestions. To validate the effectiveness of our *Agentic RL* framework and justify the choice of optimization algorithms, we trained new ablation variants of Supervised Fine-tuning (SFT) and the standard Proximal Policy Optimization (PPO) algorithm to study how different framework and algorithm affect the agent performance. All other experiment settings of our framework are kept identical.
>
> The ablation results in ``Table 4`` (``Section 5``) reveal two key findings:
> 1. **Framework RL outperforms SFT:** Our Doctor-R1 model significantly outperforms our SFT baseline over all metrics, especially in Communication (47.16 vs. 27.98), Content Seeking (26.39 vs. 16.42), and Health Data Task (29.17 vs. 16.50).
> 2. **RL Algorithm GRPO outperforms PPO:** We also observed that the GRPO method establishes the strongest performance over all metrics, demonstrating particular superiority over PPO in Communication (47.16 vs. 39.40).
>
> ***Theoretical Justification for GRPO Superiority :***
> While SFT provides a crucial initialization by minimizing the negative log-likelihood of expert demonstrations, it is theoretically limited by *exposure bias* and unable to recover from states not seen during training, as it focuses on next-token prediction rather than the long-term value of the entire consultation session.
> While PPO relies on an Actor-Critic architecture, its performance in open-ended text generation is often bottlenecked by the *value function approximation error*. Accurately estimating the scalar value $V(s)$ for complex, long-horizon medical dialogues is difficult due to the sparsity of the state space. Inaccurate critic estimates lead to high-variance advantage signals, destabilizing the policy update.
> In contrast, GRPO eliminates the critic network entirely. By sampling a group of trajectories and using the group mean as a dynamic baseline, GRPO reduces the optimization to a *listwise comparison* within the sampled group.
> Theoretically, this yields two benefits:
> 1) **Bias Reduction:** It removes the bias introduced by an imperfect critic, ensuring the policy gradient is driven solely by the ground-truth reward distribution.
> 2) **Robustness in Soft Skills:** For subjective metrics like Communication Quality, learning from relative preference is mathematically more stable than regressing to an absolute score. This explains GRPO's specific superiority in soft-skill dimensions compared to PPO.
>
>
> Table 4: Ablation study of performance across SFT, PPO, and GRPO.
>
> | Doctor-R1 Variant | Avg. Score | Emerg. Referrals | Health Data Task | Communication | Global Health | Hedging | Context Seeking | Complex Response | Acc. | Comm. Quality | Instr. Follow. | Context Aware. | Completeness |
> |------------|---------|-------------|---------|--------|---------|---------|---------|----------|------|----------|---------|------|--------|
> | Base Model        | 25.13      | 45.42            | 16.50         | 27.98         | 15.26         | 25.34  | 16.42           | 30.69         | 28.57 | 49.35         | 43.51          | 43.00         | 27.24       |
> | + SFT             | 29.54      | 49.69            | 22.22         | 32.53         | 20.17         | 29.14  | 22.83           | 29.64         | 32.37 | 57.32         | 48.75          | 46.04         | 33.46       |
> | + PPO             | 33.23      | 51.77            | 24.96         | 39.40         | 24.29         | 33.50  | 23.82           | 31.51         | 36.96 | 59.29         | 49.48          | 46.86         | 38.72       |
> | **+ GRPO**        | **36.29**  | **54.44**        | **29.17**     | **47.16**     | **24.74**     | **33.71** | **26.39**      | **34.25**     | **37.84** | **64.15** | **54.39** | **49.24** | **40.93** |
>
> ---
>
> - **Q5:** The author acknowledges the "key challenge" of patient agent adherence but provides insufficient evidence that the mitigation strategies are effective. What percentage of simulated dialogues are filtered out due to adherence failures?
> - **Q8:** What percentage of training dialogues are filtered due to patient agent adherence failures?
>
> **Reply to Q5 and Q8:** Thank you for your question. To validate the effectiveness of our mitigation strategies, we track the rejection rates of our filtering pipeline. Out of the total pool of generated simulations, 13.82% were automatically discarded due to adherence failures. The following case studies demonstrate how our filtering pipeline effectively identifies and discards patient agents that fail to maintain persona or engage in unrealistic information disclosure (see ``Appendix G.1`` for detailed settings and case studies).

---

> ### Author Response · Authors · 2025-11-24
>
> - **Q6:** The multi-objective reward structure, while helpful, does not eliminate the possibility of exploitation. Human spot-checking is mentioned but no statistics on frequency or findings are provided.
> - **Q10:** Under what conditions does the system produce unsafe recommendations despite the veto mechanism?
>
> **Reply to Q6 and Q10:** Thanks for your careful observation. To validate these mechanisms, we implement a human-in-the-loop review process. Throughout the training phase, we manually review approximately 50 high-reward trajectories per day. In early training steps, we did observe minor instances of reward hacking, such as the model generating repetitive empathetic statements to maximize the *Empathy* score while providing low-information content. Our hierarchical veto system and multi-objective weights are specifically iterated to penalize these behaviors. Later reviews confirm these strategies were effective, for example, a response that is safe and empathetic but lack reasoning quality would see its total reward significantly capped, preventing the policy from converging on a "safe-but-useless" local optimum. Case studies in ``Appendix G.2`` illustrate how our Consultation Evaluator provides detailed multi-dimensional feedback, identifying specific deficiencies even when the response is factually safe.
>
> ---
>
> - **Q11:** What types of experiences does the system most frequently retrieve? Are there cases where novelty filtering excludes valuable high-quality examples?
>
> **Reply to Q11:** We appreciate this insightful question. We have added a qualitative analysis in Appendix D.2 to address this.
> 1. **Distribution of Retrieved Experiences:** We manually categorized 100 retrieved samples. We found the system most frequently retrieves two types of high-value trajectories: 1) **Strategic Pivots:**  Instances where the experience helps the agent shift focus from an obvious but incorrect hypothesis to a subtle, correct one (Case Study 1 in ``Appendix D.2``). 2）**Differential Diagnosis Refinement:** Instances where experience helps disambiguate overlapping symptoms in high-stakes scenarios (Case Study 2 in ``Appendix D.2``).
> 2. **Novelty Filtering Trade-off:** Regarding the exclusion of valuable examples, this answer is a yes, however, our filtering mechanism (Stage 3) effectively excludes high-quality (high-reward) examples **if and only if** they are highly redundant with existing entries. This is a deliberate design to prevent the repository from collapsing into a mode of "repetitive easy cases" (for example hundreds of identical Flu diagnoses).
>
> ---
>
> - **Q12:** How does performance degrade when evaluated on out-of-distribution cases (different patient populations, communication styles, or medical domains)?
>
> **Reply to Q12:** We address this by clarifying our evaluation setup and presenting evidence from three distinct OOD perspectives:
> 1. **Generalization across Patient Simulators (Communication Styles):**
> The reviewer asks about performance degradation on different communication styles. We emphasize that our main results (``Tables 1`` and ``Table 2``) are already OOD tests. 1) **Training Domain:** Our agent is trained interacting with a *Qwen-based* patient simulator. 2) **Evaluation Domain:** In contrast, the HealthBench and MAQuE benchmarks utilize *GPT-based* to generate patient responses, which represents different communication styles. The strong performance in benchmark indicates the generalization of our approach.
> 2. **Generalization across Medical Domains:**
> HealthBench is specifically designed to test diverse domains. It covers distinct "Themes" ranging from Emergency Referrals to Global Health and Complex Pathologies. As shown in ``Table 1``, Doctor-R1 maintains consistent high performance across these varied domains, demonstrating that the methodology is not restricted to a narrow set of medical conditions seen during training.
> 3. **Generalization across Patient Populations:** HealthBench tasks (especially Global Health) are constructed based on different demographics (pediatric, adult, and elderly populations), socioeconomic and geographic contexts, and data modalities (mixing structured EHR data with unstructured patient narratives).
> 4. **Generalization to Human Evaluation (Real-World OOD):**
> We conducted OOD test through real human expert evaluation. Our Medical Expert Evaluation (``Section 4.2``) represents a shift from simulated patients to expert physician judgment. The fact that licensed physicians show dominant preference for Doctor-R1 against baselines provides strong evidence that our method aligns with real-world clinical standards.

---

> > ### Comment · Reviewer_bTuJ · 2025-11-28
> >
> > Thanks for the detailed response and revision. I appreciate the hard work from the authors. Several key concerns have been addressed. However, one issue is about the Human Evaluation Protocol. There are two evaluation for the patient user and the medical experts respectively. But the details of evaluation are missing. Could the author provide more details about it? For example, how to determine and define the qualitative metrics? Is it provided in the form of options or directly as text input? I will raise my score accordingly.

---

> ### Author Response · Authors · 2025-11-28
>
> We sincerely appreciate the reviewer for the encouraging feedback and the decision to increase the rating. We are greatly encouraged by your recognition, and are glad that our additional experiments and revisions have addressed your key concerns.
>
> Regarding the **Human Evaluation Protocol**, we apologize if the location of these details was not immediately clear. We have provided the full methodology (including metric definitions and annotation process) in ``Appendix E`` of the revised PDF. To answer your specific questions:
> 1. **Definition of Metrics:** We provided annotators with explicit standardized definitions for each metric to ensure consistency.
>     - For example, regarding "Empathy" (*Patient Evaluation*), annotators were instructed to judge: *"Does the model's tone convey care, respect, and support?"*
>     - For "Clinical Safety" (*Expert Evaluation*), physicians used a rubric checking for harmful advice or missed red-flag symptoms.
> 2. **Input Format:** The evaluations were conducted using **structured options** rather than free-form text input to ensure quantitative rigor.
>     - **Patient Experience Evaluation:** This was a **Pairwise Comparison**. Annotators were presented with two blinded responses and selected an option: *"Model A is better"*, *"Model B is better"*, or *"Tie"*.
>     - **Medical Expert Validation:** For clinical competence, it was a **5-point Likert Scale**. Human experts selected a score from options 1 (*Poor*) to 5 (*Excellent*) based on the detailed rubric provided. While for experience utility, it was conducted via **Categorical Selection**. Experts classified retrieval results into fixed categories: *"Clinically Helpful"*, *"Neutral and Irrelevant"*, or *"Harmful"*.
>
> We hope this summary clarifies the evaluation setup. Thank you for your valuable time and suggestions to improve our work.

---

### Official Review · Reviewer_ghkz · 2025-10-31

**Soundness:** 4
**Presentation:** 4
**Contribution:** 3
**Rating:** 6
**Confidence:** 4

**Summary:**

This paper proposes an RL-based agent, namely Doctor-R1, for strategic inquiry and empathetic communication. Evaluating on HealthBench and MAQuE datasets, the proposed framework outperforms open-source and commercial general models. Human evaluation shows the Doctor-R1 has improvements over baselines.

**Strengths:**

(1) This paper aligns with the trend to extend the medical LLMs from static decision-making to dynamic inquiry through an agentic RL perspective.
(2) This paper explicitly models empathy and communication quality as trainable objectives.

**Weaknesses:**

(1) The technical details are missing. The authors need to provide more technical details. For example the authors need to define how vector rewards are aggregated or how gradient signals are propagated.
(2) The authors rely on the GRPO framework for the RL. Ablation studies need to be provided on how different RL frameworks will affect the performance.
(3) The authors use Qwen3-8B as the base model. Is this framework transferable to other base models?

**Questions:**

(1) Now the framework is offline and off-policy. Is it possible to change the framework to make it online?

---

> ### Author Response · Authors · 2025-11-24
>
> We thank the reviewer for the careful reading and valuable suggestions. We are pleased that you found our approach and results meaningful, and we have carefully considered each of the points raised. ***We have carefully addressed all questions and updated the corresponding revisions into the new PDF, with key changes highlighted in blue text for clarity.*** In the following, we respond point-by-point to each comment and provide additional explanations and experiments where necessary.
>
> - **Q1:** The technical details are missing. The authors need to provide more technical details. For example the authors need to define how vector rewards are aggregated or how gradient signals are propagated.
>
> **Reply to Q1:**
> As detailed in ``Appendix B.1`` (Equation 7), the 8-dimensional vector reward is aggregated into a single scalar process reward signal $R_\text{{turn}}$ for the policy update. As defined in Equation 2, this aggregation is governed by our **hierarchical veto system** where critical failures in Safety or Accuracy trigger a large negative penalty, overriding all other scores. If no veto is triggered, a weighted sum is used. The final $R_\text{{final}}$) is the final outcome reward signal.
>
> The gradient signal is propagated from this scalar reward using our chosen policy optimizer GRPO. Here is the theoretical flow for both reward types:
> - **For Process Rewards (Turn 1 to T-1):** At each turn $t$, the agent's action (a question) is evaluated by the Consultation Evaluator. This generates the process reward $R_{\text{turn}}$, which is plugged directly into the GRPO loss function ($L_{\text{GRPO}}$ Equation 1) as the reward $R_\psi(x_t, y_t)$ for that specific (state, action) pair.
> - **For the Outcome Reward (Turn T):** At the final turn $T$, the agent's action (the final diagnosis) is evaluated. This generates the outcome reward $R_{\text{final}}$, which is also plugged into the same GRPO loss function as the reward $R_\psi(x_T, y_T)$ for that final (state, action) pair.
> - **For Backpropagation:** Since the GRPO loss function is fully differentiable with respect to the policy model's parameters ($\theta$), a gradient ( $\nabla_{\theta} L_{\text{GRPO}}$ ) is calculated from this loss at every step. This gradient, derived from the entire sequence of process rewards ($R_{\text{turn}}$) and the final outcome reward ($R_{\text{final}}$), is then used to update the policy model's parameters via standard backpropagation.

---

> ### Author Response · Authors · 2025-11-24
>
> - **Q2:** The authors rely on the GRPO framework for the RL. Ablation studies need to be provided on how different RL frameworks will affect the performance.
>
> **Reply to Q2:** Thank you for your valuable suggestions. To validate the effectiveness of our *Agentic RL* framework and justify the choice of optimization algorithms, we trained new ablation variants of Supervised Fine-tuning (SFT) and the standard Proximal Policy Optimization (PPO) algorithm to study how different framework and algorithm affect the agent performance. All other experiment settings of our framework are kept identical.
>
> The ablation results in ``Table 4`` (``Section 5``) reveal two key findings:
> 1. **Framework RL outperforms SFT:** Our Doctor-R1 model significantly outperforms our SFT baseline over all metrics, especially in Communication (47.16 vs. 27.98), Content Seeking (26.39 vs. 16.42), and Health Data Task (29.17 vs. 16.50).
> 2. **RL Algorithm GRPO outperforms PPO:** We also observed that the GRPO method establishes the strongest performance over all metrics, demonstrating particular superiority over PPO in Communication (47.16 vs. 39.40).
>
> ***Theoretical Justification for GRPO Superiority :***
> While SFT provides a crucial initialization by minimizing the negative log-likelihood of expert demonstrations, it is theoretically limited by *exposure bias* and unable to recover from states not seen during training, as it focuses on next-token prediction rather than the long-term value of the entire consultation session.
> While PPO relies on an Actor-Critic architecture, its performance in open-ended text generation is often bottlenecked by the *value function approximation error*. Accurately estimating the scalar value $V(s)$ for complex, long-horizon medical dialogues is difficult due to the sparsity of the state space. Inaccurate critic estimates lead to high-variance advantage signals, destabilizing the policy update.
> In contrast, GRPO eliminates the critic network entirely. By sampling a group of trajectories and using the group mean as a dynamic baseline, GRPO reduces the optimization to a *listwise comparison* within the sampled group.
> Theoretically, this yields two benefits:
> 1) **Bias Reduction:** It removes the bias introduced by an imperfect critic, ensuring the policy gradient is driven solely by the ground-truth reward distribution.
> 2) **Robustness in Soft Skills:** For subjective metrics like Communication Quality, learning from relative preference is mathematically more stable than regressing to an absolute score. This explains GRPO's specific superiority in soft-skill dimensions compared to PPO.
>
>
> Table 4: Ablation study of performance across SFT, PPO, and GRPO.
>
> | Doctor-R1 Variant | Avg. Score | Emerg. Referrals | Health Data Task | Communication | Global Health | Hedging | Context Seeking | Complex Response | Acc. | Comm. Quality | Instr. Follow. | Context Aware. | Completeness |
> |------------|---------|-------------|---------|--------|---------|---------|---------|----------|------|----------|---------|------|--------|
> | Base Model        | 25.13      | 45.42            | 16.50         | 27.98         | 15.26         | 25.34  | 16.42           | 30.69         | 28.57 | 49.35         | 43.51          | 43.00         | 27.24       |
> | + SFT             | 29.54      | 49.69            | 22.22         | 32.53         | 20.17         | 29.14  | 22.83           | 29.64         | 32.37 | 57.32         | 48.75          | 46.04         | 33.46       |
> | + PPO             | 33.23      | 51.77            | 24.96         | 39.40         | 24.29         | 33.50  | 23.82           | 31.51         | 36.96 | 59.29         | 49.48          | 46.86         | 38.72       |
> | **+ GRPO**        | **36.29**  | **54.44**        | **29.17**     | **47.16**     | **24.74**     | **33.71** | **26.39**      | **34.25**     | **37.84** | **64.15** | **54.39** | **49.24** | **40.93** |

---

> ### Author Response · Authors · 2025-11-24
>
> - **Q3:** The authors use Qwen3-8B as the base model. Is this framework transferable to other base models?
>
> **Reply to Q3:** This is indeed a good suggestion. To investigate the generalizability of our framework, we apply our training pipeline to a different base model family, **Llama3-8B-Instuct**. We evaluate the original base model, the SFT variant, the PPO variant, and our full GRPO variant on HealthBench. The results presented in ``Table 12`` (``Appendix D.1``, page 22) are highly consistent with our primary findings on the Qwen3-8B model.
> 1. The Llama-3-8B base model's performance was significantly improved by our pipeline.
> 2. The final Doctor-R1 (Llama-3-8B-Instruct base + GRPO) model outperforms base model over all metrics, and achieves 16.43% improvement in Communication.
> 3. The performance stack-up (GRPO > PPO > SFT > Base) holds true for the Llama family as well.
> This new experiment demonstrates that our *Experiential Agentic RL* approach can be applied across different model families.
>
> Table 12: Ablation study on Llama-3.1-8B variants showing transferability of the framework.
>
> | Llama-3.1-8B Variant | Avg. Score | Emerg. Referrals | Health Data T. | Communication | Global Health | Hedging | Context Seeking | Complex Resp. | Acc. | Comm. Quality | Instr. Follow. | Context Aware. | Completeness |
> |----------------------|------------|------------------|----------------|---------------|----------------|---------|------------------|----------------|------|----------------|-----------------|----------------|--------------|
> | Base Model           | 13.73      | 25.45            | 13.66         | 13.21         | 7.66          | 12.46  | 5.79            | 24.82         | 18.64 | 38.02         | 34.75          | 32.39         | 15.99       |
> | + SFT                | 16.83      | 29.70            | 16.88         | 17.85         | 8.31          | 15.31  | 9.01            | 27.14         | 22.24 | 44.01         | 36.80          | 34.87         | 19.54       |
> | + PPO                | 19.23      | 31.06            | 16.38         | 24.80         | 9.60          | 17.04  | 10.87           | 28.20         | 24.08 | 50.63         | 42.18          | 34.83         | 22.54       |
> | **+ GRPO**           | **20.76**  | **32.06**        | **20.55**     | **24.38**     | **11.13**     | **19.36** | **12.15**      | **31.00**     | **24.35** | **54.45** | **46.97** | **35.29** | **23.17** |
>
> ---
>
> - **Q4:** Now the framework is offline and off-policy. Is it possible to change the framework to make it online?
>
> **Reply to Q4:** Thank you for your question. We respectfully clarify that the training framework is, in fact, **online and interactive**. In the VeRL [1] framework, GRPO does not employ the “centered sampling” mechanism. Instead, it uses **group sampling** and **relative reward normalization** to estimate the advantage function, thereby replacing the critic network in traditional PPO. Our training process is online and interactive where the doctor agent continuously generates new dialogues, receives rewards, stores, updates and retrieves experience repository in realtime, and updates its policy within the simulated patient environment.
> As illustrated in ``Figure 2``, our training process is as follows:
> 1. Interact and Generate: The Doctor Agent (policy $\pi_k$) interacts with the Patient Agent environment to generate a new batch of dialogues.
> 2. Real-Time Evaluation: The Consultation Evaluator provides rewards for this new batch in real-time.
> 3. Experience Update: High-reward trajectories are identified and **stored and updated in the Experience Repository in real-time**, making them available for retrieval in subsequent steps.
> 4. Policy Update: These new (dialogue, reward) trajectories are immediately used to update the policy to $\pi_{k+1}$.
> This continuous loop of "generate-evaluate-update" ensures the agent is constantly learning from its latest interactions
>
> [1] Sheng et al. "HybridFlow: A Flexible and Efficient RLHF Framework"

---

### Official Review · Reviewer_usVB · 2025-11-03

**Soundness:** 2
**Presentation:** 3
**Contribution:** 2
**Rating:** 4
**Confidence:** 5

**Summary:**

The paper proposes Doctor-R1, a clinical dialogue agent trained to conduct strategic, empathetic multi-turn inquiry while also making sound medical decisions. The system employs three key concepts: a multi-agent interactive environment, a two-tiered reward structure that distinguishes between outcome quality and process quality, and an experience repository that stores valuable trajectories for reuse. The model is evaluated on HealthBench and MAQuE with automatic metrics and human preference studies. The claim is that Doctor-R1 improves communication quality and user experience while maintaining competitiveness or surpassing it in task accuracy.

**Strengths:**

1. Most medical LLM work measures final answers. This paper addresses the gap in the skill of guided clinical inquiry and communication, which is crucial in practice.
2. The two-tiered reward isolates process signals from outcome signals, which is a sensible way to avoid reward dilution and to target conversational quality.
3. The experience module that learns from high-quality trajectories is a practical way to ground policy learning without heavy annotation.
4. The study reports multi-facet metrics and human judgments, not only accuracy.

**Weaknesses:**

1. The models used across experiments are not consistent. For example, the human evaluation uses GPT-5, but the main tables and ablations only report GPT-4.1. GPT-5 is also not mentioned in the Evaluation Settings section. Could you unify the model choices across experiments? If different models need to be used, please provide a brief explanation for the selection.

2. The authors state: “The annotators recruited were without a specialized medical background to ensure that metrics like Clarity and Empathy were evaluated from the perspective of a typical patient, for whom the agent is ultimately designed.” However, the paper sets out three research questions: (2) Empathetic Communication is only one, whereas (1) Strategic and Dynamic Inquiry and (3) Learning from Good Experience requires a medical background to assess. For instance, at each turn-level Doctor-R1 response, expert evaluation can better verify whether the questions asked are clinically appropriate; experts can also check whether the Experience Module’s stored items and retrieved content are indeed appropriate and helpful. I do not think these two research questions can be avoided in the human evaluation; focusing only on non-medical-background evaluation of (2) Empathetic Communication is insufficient.

3. An additional analysis would be to break down time and compute overhead by component. For example, after adding the “Learning from Good Experience” design, how much extra latency or token usage does the model incur?

4. The ablation study is missing analyses on how different reward designs affect results. A core claim of the paper is integrating a TWO-TIERED REWARD into RL training. Beyond the outcome reward, many additional rewards are proposed, and the method section includes empirically motivated statements such as “This (e.g.,  hierarchical penalty design) addresses the limitations of conventional weighted-sum models, which can fail to adequately penalize catastrophic errors due to an averaging-out effect.” It would be beneficial to include more recent related work, for example, how other studies attempt to incorporate additional process rewards on top of RLVR while maintaining training stability, or at least present supportive mini-experiments. Also, the paper only presents a coarse “w/o Process Reward” setting. Please consider ablating some of the eight process rewards to toggle them on or off, especially the key ones, to assess whether such designs can generalize to other medical scenarios.

**Questions:**

1. Why is Qwen3-8B also used as the LLM judge for the Process Reward?

---

> ### Author Response · Authors · 2025-11-24
>
> We would like to thank the reviewer for the thoughtful reviews and the constructive feedback you provided. ***We have carefully addressed all questions and updated the corresponding revisions into the new PDF, with key changes highlighted in blue text for clarity.*** In the following, we address your questions in detail, responding point-by-point to each comment and provide additional explanations and experiments where necessary.
>
> - **Q1:** The models used across experiments are not consistent. For example, the human evaluation uses GPT-5, but the main tables and ablations only report GPT-4.1. GPT-5 is also not mentioned in the Evaluation Settings section. Could you unify the model choices across experiments? If different models need to be used, please provide a brief explanation for the selection.
>
> **Reply to Q1:** Thank you for pointing out this discrepancy. We apologize for the inconsistency regarding the model versions used in the evaluation. To ensure a unified and rigorous comparison across all experiments, we have now evaluated GPT-5 (gpt-5-chat-2025-08-07) on both the HealthBench and MAQuE benchmarks (``Table 1`` and ``Table 2``) to align with the human evaluation models. The detailed GPT-5 version has also been added to ``Appendix C.2``.
>
> The updated results in ``Table 1 (Extracted)`` is extracted from ``Section 4.1``. Doctor-R1 (8B) **outperforms the best open-source baseline** Baichuan-M2-32B (36.29 vs. 33.16), despite having **4x fewer parameters**. Notably, our 8B model also **surpasses GPT-4.1 (36.29 vs. 31.18)**, demonstrating that specialized Agentic RL can bridge the gap between small open-source models and powerful generalist APIs. While GPT-5 leads in overall score (46.38), Doctor-R1 actually **outperforms GPT-5 on Communication Quality (64.15 vs. 62.50)**. This strongly validates our framework's effectiveness and the consistency with human evaluation results.
>
> Table 1 (Extracted): Overall performance on HealthBench Main **(best results of open-source models are bolded)**
> | Model              | Avg. Score | Emerg. Referrals | Health Data | Comm. | Global Health | Hedging | Context Seeking | Complex Resp. | Acc.   | Comm. Quality | Instr. Foll. | Context Aware. | Completeness |
> |-------------------|------------|------------------|-------------|-------|----------------|---------|------------------|----------------|--------|----------------|--------------|----------------|--------------|
> | GPT-4.1           | 31.18      | 53.98            | 23.51       | 37.03 | 22.36          | 29.40   | 21.79           | 45.93          | 34.78  | 60.65         | 54.32        | 44.81         | 34.84        |
> | GPT-5             | 46.38      | 63.22            | 34.80       | 57.12 | 37.12          | 40.16   | 37.84           | 54.26          | 44.68  | 62.50         | 61.31        | 53.32         | 50.02        |
> | Baichuan-M2-32B   | 33.16      | 20.16            | 8.90        | 22.11 | 18.40          | 17.31   | **28.24**       | 24.45          | 33.95  | 58.01         | 52.40        | 46.80         | 40.03        |
> | Doctor-R1 (Ours)  | **36.29**  | **54.44**        | **29.17**   | **47.16** | **24.74**  | **33.71** | 26.39       | **34.25**      | **37.84** | **64.15**   | **54.39**    | **49.24**     | **40.93**    |

---

> ### Author Response · Authors · 2025-11-24
>
> - **Q2:** The authors state: “The annotators recruited were without a specialized medical background to ensure that metrics like Clarity and Empathy were evaluated from the perspective of a typical patient, for whom the agent is ultimately designed.” However, the paper sets out three research questions: (2) Empathetic Communication is only one, whereas (1) Strategic and Dynamic Inquiry and (3) Learning from Good Experience requires a medical background to assess. For instance, at each turn-level Doctor-R1 response, expert evaluation can better verify whether the questions asked are clinically appropriate; experts can also check whether the Experience Module’s stored items and retrieved content are indeed appropriate and helpful. I do not think these two research questions can be avoided in the human evaluation; focusing only on non-medical-background evaluation of (2) Empathetic Communication is insufficient.
>
> **Reply to Q2:** Thank you for your valuable suggestion. We recognized the importance of expert evaluation, hence we conducted a new **comprehensive expert evaluation study** last week, with 2 licensed physicians to address both concerns simultaneously. This study consisted of two main tasks:
>
> **Task 1: Validating our Agent's Clinical Skills**
> To address the valid concern of agent's clinical skills, we first asked the physicians to score 40 cases using the exact same four dimension rubric as our LLM-judge (``Appendix B.1``).
> 1. **Principle 1 (Strategic Inquiry):** The physicians conducted a 1-5 Likert scale scoring of full dialogues between Doctor-R1 and the strong baselines (including GPT-5, Baichuan-M2-32B and UltraMedical-70B). As shown in our new ``Figure 3`` (``Section 4.2``), Doctor-R1 outperforms most of the baseline models, especially in Safety and Information Gathering.
> 2. **Principle 2 (Empathy):** For the patient user experience evaluation, we recruited five annotators without specialized medical training to ensure that subjective metrics were judged from the perspective of a typical patient. The results in ``Figure 3`` show a strong human preference for our model, which ranked first across all four categories **(92.5% win rate on "Empathy")**.
> 3. **Principle 3 (Experience Usefulness):** We asked the experts to directly evaluate the clinical utility of our experience module. On 30 samples, experts were shown the retrieved experience and asked to rate its usefulness out of three distinct categories: 1) Clinically Helpful, 2) Neutral and Irrelevant, and 3) Harmful. The **experts agreed the experience was *"Clinically Helpful"* in 83.87% of cases** for guiding the agent's next action, with only 16.13% rated as "Neutral and Irrelevant" and 0% as "Harmful".
> This new expert-driven evaluation provides crucial, direct validation for our primary research contributions (Principles 1, 2 and 3).
>
> **Task 2: Validating our Consultation Evaluator**
> To validate the reliability of evaluator model, we conduct an **evaluation study involving clinical experts** and analyze the correlation between the automated scores and human expert scores. We use two independent and stronger judges, GPT-4.1 and human expert, to score 40 cases. ``Table 14`` and ``Table 15`` (``Appendix E.3``) shows the score given by each judge, demonstrating that the relative ranking of the models remains **highly consistent**. Also, the Spearman's rank correlation calculated confirm the evaluation **high stability and reliability** between judges.
>
>
>
> Table 14: Comparison of average model scores from three different independent judges.
> | Model Evaluated          | Qwen3-8B | GPT-4.1 | Human Expert |
> |--------------------------|---------:|--------:|-------------:|
> | UltraMedical-8B          | 3.06     | 2.33    | 3.47         |
> | Baichuan-M2-32B          | 3.29     | 2.93    | **4.31**     |
> | GPT-5                    | **3.38** | **3.25**| 4.05         |
> | **Doctor-R1**            | _3.36_   | **3.25**| _4.30_       |

---

> > ### Author Response · Authors · 2025-11-24
> >
> > - **Q4:** The ablation study is missing analyses on how different reward designs affect results. A core claim of the paper is integrating a TWO-TIERED REWARD into RL training. Beyond the outcome reward, many additional rewards are proposed, and the method section includes empirically motivated statements such as “This (e.g., hierarchical penalty design) addresses the limitations of conventional weighted-sum models, which can fail to adequately penalize catastrophic errors due to an averaging-out effect.” It would be beneficial to include more recent related work, for example, how other studies attempt to incorporate additional process rewards on top of RLVR while maintaining training stability, or at least present supportive mini-experiments. Also, the paper only presents a coarse “w/o Process Reward” setting. Please consider ablating some of the eight process rewards to toggle them on or off, especially the key ones, to assess whether such designs can generalize to other medical scenarios.
> >
> > **Reply to Q4:** Regarding the concern in reward architecture, we conducted ablation
> >
> > 1. Justification of Reward Criteria (Grounding and Ablation):
> > - Grounding: We have revised ``Section 3.2`` to explicitly ground our eight process reward criteria in established frameworks for clinical communication, including the works suggested ([arXiv:2502.14860, 2502.07143]).
> > - Ablation Study: To empirically validate our reward design, We trained two variant models using the same experiment settings: 1) **without Veto System:** Uses a simple weighted sum reward $\sum w_i S_i$ without the safety veto threshold, and 2) **without Soft Skill:** Sets the weights for Empathy and Humility to zero during training (detailed settings listed in ``Appendix D.1``, page 22).
> > **1) Veto System Enforces Safety:** Removing the hierarchical veto (*w/o Veto System*) leads to a degradation in safety. While the average scores drop only slightly, the Safety Violation Rate more than doubles (+125\%) compared to our proposed model. Furthermore, the Emergency Referrals score drops by -3.16\% (52.72 vs. 54.44), indicating that without the penalty for safety failures, the model fails to maintain appropriate caution.
> > **2) Process Rewards Shape Soft Skills:** Removing the soft skill rewards (*w/o Soft-Skills*) causes a degradation in communicative competence. The performance shows a drop of -16.25\% (47.16 vs. 39.40) in the Communication and -11.62\% (4.39 vs. 3.88) Empathy score, showing that specific process reward is required for soft skills shaping.
> >
> > Table 13: Ablation results for reward architecture.
> > | Reward Architecture Variant | Emergency Referrals ↑ | Safety Violation ↓ | Communication ↑ | Empathy ↑ |
> > | :--- | :--- | :--- | :--- | :--- |
> > | **Doctor-R1** | **54.44** | **0.80** | **47.16** | **4.39** |
> > | &nbsp;&nbsp;+ Process Reward (w/o Veto System) | 52.72 **(-3.16%)** | 1.80 **(+125%)** | 46.86 | 4.19 |
> > | &nbsp;&nbsp;+ Process Reward (w/o Soft Skill) | 54.31 | 1.00 | 39.40 **(-16.25%)** | 3.88 **(-11.62%)** |
> > | &nbsp;&nbsp;w/o Process Reward | 52.21 | 1.80 | 39.05 | 3.82 |
> >
> > ---
> >
> >
> > - **Q5:** Why is Qwen3-8B also used as the LLM judge for the Process Reward?
> >
> > **Reply to Q5:**
> > At the time of our study's development, Qwen3-8B was the state-of-the-art open-source model in the 7B and 8B-class. As evidenced by [1], it demonstrated top-tier performance on general, math, reasoning, agent and coding tasks, and even outperformed the 14B open-source models over half of the benchmarks. Our experiment settings require a strong and smaller model (around 7-8B) to efficiently execute our complex reference-guided evaluation.
> > We recruited 2 licensed physicians and prove that our LLM judge scores have a strong positive correlation with human expert judgment. This shows that the evaluation of our LLM judge is aligned with the human evaluation standards.
> >
> > [1] Qwen Team. "Qwen3 Technical Report"

---

> ### Author Response · Authors · 2025-11-24
>
> - **Q3:** An additional analysis would be to break down time and compute overhead by component. For example, after adding the “Learning from Good Experience” design, how much extra latency or token usage does the model incur?
>
> **Reply to Q3:** To address your concern regarding the computational overhead, we present a detailed breakdown of the time and cost associated with our experience retrieval module in ``Appendix I``. We analyzed both the one-time training cost and the per-inference latency cost. Our results indicate that the proposed method offers a **highly favorable cost-benefit trade-off particularly in a high-stakes medical domain**.
>
> 1. **Training Cost Overhead:** The primary overhead comes from 1) the retrieval step for each sample and 2) processing a longer token sequence due to prepending top-$k=2$ experiences during the forward and backward passes. ``Table 17`` shows the time per training step for different components of our framework. We can isolate the marginal cost of the experience module: In a non-interactive setting, adding Experience added 638s, while in our full interactive setting, adding Experience added 679s. This shows a 20.38% increase in training time per step to incorporate the experience module.
>
> Table 17: Breakdown of time per training step
> | **Training Configuration**                | **Time / step (s)** | **Time per token / gen (ms)** |
> |-------------------------------------------|---------------------:|-------------------------------:|
> | No Interaction & No Experience            | 1079.0068            | 0.0900                        |
> | + Experience Only                          | 1717.2896            | 0.6897                        |
> | + Interaction Only (Baseline)              | 3331.4185            | 1.1382                        |
> | **Full Method (+ Interaction + Exp)**      | **4010.1689**        | **1.3121**                    |
>
> 2. **Inference Cost Overhead:** The retrieval module adds an average of 624.67 tokens to the prompt from the top-$k=2$ retrieved experiences. This additional context combined with the retrieval step itself adds an average of only 4.74s of latency per inference sample. This **minimal inference cost** listed in ``Table 18`` represents a highly favorable cost-benefit tradeoff. This modest overhead is directly responsible for the substantial performance gain on HealthBench (``Table 8``) on average score (36.29 vs. 31.69). The experience module enables **continuous learning**, which allows the agent to update and learn from its experience repository with new cases without expensive retraining. This adaptability actually addresses the huge demand in real-world clinical settings.
>
> Table 18: Inference overhead analysis
> | **Model Variant at Inference**   | **Avg. Tokens per Sample** | **Avg. Latency per Sample (s)** |
> |------------------------------------|--------------------------|------------------------------|
> | Doctor-R1 (Full Method)            | 2190.06                  | 52.91                        |
> | Doctor-R1 (w/o Experience)         | 1565.39                  | 48.17                        |
> | **Overhead**                       | **+624.67 tokens**       | **+4.74 s**                  |
>
> Moreover, the experience module is the deciding factor in our model's superiority. **Without using experience, our model would lag behind the state-of-the-art open-source baseline** Baichuan-M2-32B (31.69 vs. 33.16). However, adding this module boosts performance to 36.29, allowing our 8B model to outperform the 32B baseline, confirming its valuable improvement.
>
>
> Table 8: Ablation studies on the experience retrieval mechanism.
> | **Retrieval Variants** | **Avg. Score** | **Emerg. Referrals** | **Health Data T.** | **Communication** | **Global Health** | **Hedging** | **Context Seeking** | **Complex Resp.** | **Acc.** | **Comm. Quality** | **Instr. Foll.** | **Context Aware.** | **Completeness** |
> |------------------------|----------------|-----------------------|---------------------|--------------------|--------------------|-------------|----------------------|---------------------|---------|---------------------|-------------------|----------------------|-------------------|
> | Baichuan-M2-32B   | 33.16      | 20.16            | 8.90        | 22.11 | 18.40          | 17.31   | 28.24           | 24.45          | 33.95  | 58.01         | 52.40        | 46.80         | 40.03        |
> | No Experience          | 31.69          | 47.24                | 25.30               | 38.49             | 21.58             | 31.79      | 24.15               | 30.55              | 35.96  | 59.19              | 51.75            | 45.31               | 36.23            |
> | **Doctor-R1 (Using Experience)**          | **36.29**      | **54.44**            | **29.17**           | **47.16**         | **24.74**         | **33.71**  | **26.39**           | **34.25**          | **37.84** | **64.15**        | **54.39**        | **49.24**           | **40.93**         |

---

> ### Comment · Reviewer_usVB · 2025-11-26
>
> thanks the authors to resolve most of my concerns. I increase my rating accordingly.

---

> ### Author Response · Authors · 2025-11-27
>
> We sincerely thank the reviewer for your time, constructive feedback, and the decision to increase the rating. We are greatly encouraged by your recognition. We are particularly grateful for your suggestions regarding the expert validation, reward architecture analysis and computational overhead breakdown, which have strengthened the completeness of our manuscript. We are glad that our responses and additional experiments have resolved your concerns.

---

### Official Review · Reviewer_YuDH · 2025-11-04

**Soundness:** 3
**Presentation:** 3
**Contribution:** 2
**Rating:** 6
**Confidence:** 4

**Summary:**

The authors propose DOCTOR‑R1, an 8B-parameter doctor agent trained with an Experiential Agentic RL framework: (1) Interactive environment: a POMDP with a Doctor Agent (policy), an LLM‑based Patient Agent, and a Consultation Evaluator acting as the reward function; (2) Optimization: GRPO trains the policy to prefer higher‑reward responses within sampled groups.

**Strengths:**

1. Principled safety‑aware reward design that decouples soft‑skills and hard‑skills during training.
2. Experience‑guided control via reward‑aware and novelty‑aware retrieval that measurably helps inquiry quality.
3. Strong qualitative demonstration on a truly high‑stakes case that aligns with clinical intuition (quantify bleeding, TB history, anticoagulants, urgent referral).

**Weaknesses:**

1. The policy, patient simulator, and reward judge are all Qwen3‑8B family members, raising systemic bias and overfitting risks to the evaluator’s rubric and the simulator’s discourse style
2. the method contribution seems to be trivial, like previous rl works in huatuo-o1, FineMedLM-o1, m1 leverage rl and test-time scaling works like medprm and medadapter for medical reasoning. however in the related works, seems agentic rl for medical reasoning is not emphasized enough
3. The retrieval prepends prior “good experiences,” which could partly function as a prompting prior rather than learned policy alone. More careful controls (e.g., retrieval without actions vs. with actions; random but format‑matched snippets) would help isolate the effect.
4. The final diagnostic reward is coarse (0/0.5/1), potentially limiting sensitivity to near‑misses or safe-but-incomplete triage decisions

**Questions:**

See above

---

> ### Author Response · Authors · 2025-11-24
>
> We are grateful to the reviewer for your thorough evaluation and encouraging remarks. We truly appreciate the insightful questions raised. ***We have carefully addressed all questions and updated the corresponding revisions into the new PDF, with key changes highlighted in blue text for clarity.*** In the following, we respond point-by-point to each comment and provide additional explanations and experiments where necessary.
>
> - **Q1:** The policy, patient simulator, and reward judge are all Qwen3‑8B family members, raising systemic bias and overfitting risks to the evaluator’s rubric and the simulator’s discourse style.
>
> **Reply to Q1:**  Thank you for your question. We agree that this is an important question to be investigated and validated, hence we address this concern in three ways: 1) Clarify **safeguards in training** design that mitigate bias, 2) Perform a new **evaluation study involving clinical experts** (detailed settings listed in ``Appendix E.3``), and 3) **Validate generalization**.
> 1. **Safeguards in Training:** First, the Consultation Evaluator operates on grounded opinion. As shown in our prompt (``Appendix H``), the judge is provided with the **[Ground Truth Doctor Reply]** and is explicitly instructed to score 'Faithfulness to Ground Truth' (Criteria 6). This anchors the evaluation to expert standards, making the task "How well does this response match the expert standard?" rather than "Do you, Qwen, like this response?". Furthermore, the evaluator's weights are frozen, preventing the policy and judge from co-adapting into a self-delusional loop.
> 2. **Evaluation Validation involving Human Expert:** To prove our metrics are not inflated by the Qwen-judge, we conducted a cross study (see the below Table A). We scored the test cases using two independent stronger judges: GPT-4.1 and a panel of licensed physicians. **1) Consistency:** While absolute scores vary by judge, the **relative rankings remain highly consistent.** As shown in ``Table 14`` and ``Table 15``, Doctor-R1 consistently outperforms baselines like UltraMedical-70B and Baichuan-M2-32B across all judges. **2) Reliability:** We found a moderate high Spearman's rank correlation ($\rho > 0.52, 0.24$) between the Qwen-judge's rankings with GPT-4.1 and human expert, confirming that our Qwen-based evaluator serves as a reliable proxy for generalized clinical quality.
> 3. **Validating Agent Generalization:** Too address the concern about generalizable clinical skills, our benchmark evaluation (OpenAI's HealthBench and MAQuE) in ``Table 1`` and ``Table 2``, are actually representing patient cases **Out-of-Distribution (OOD)**. While Qwen-based patient simulator is used in our training phase, GPT-based model is used as a patient simulator in our inference phase as per the official benchmark protocols. Therefore, our strong performance on these benchmarks serves as proof of generalization on clinical skills.
>
> Table A: Comparison of average model scores from three different independent judges.
> | Model Evaluated          | Qwen3-8B | GPT-4.1 | Human Expert |
> |--------------------------|---------:|--------:|-------------:|
> | UltraMedical-8B          | 3.06     | 2.33    | 3.47         |
> | Baichuan-M2-32B          | 3.29     | 2.93    | **4.31**     |
> | GPT-5                    | **3.38** | **3.25**| 4.05         |
> | **Doctor-R1**            | _3.36_   | **3.25**| _4.30_       |

---

> ### Author Response · Authors · 2025-11-24
>
> - **Q2:** the method contribution seems to be trivial, like previous rl works in huatuo-o1, FineMedLM-o1, m1 leverage rl and test-time scaling works like medprm and medadapter for medical reasoning. However, in the related works, seems agentic rl for medical reasoning is not emphasized enough.
>
> **Reply to Q2:** Differential diagnosis through inquiry is an important yet difficult skill to be managed (while even current strong LLMs may lack this skill). Our proposed experience module enables **continuous learning**, which allows the agent to update and learn from its experience repository with new cases without expensive retraining. This adaptability actually addresses the huge demand in real-world clinical settings. (Added more detailed related work to ``Section 6``)
>
> Table 5: Comparison of our proposed Doctor-R1 with recent medical specialized LLMs.
> | Model | RL | Agentic RL (Multi-turn) | Reasoning | Process Reward | Outcome Reward | Static Medical Decision | Dynamic Clinical Inquiry | Memory (Experience) |
> |-------|----|-------------|-----------|----------|---------|----------|---------|----------|
> | HuatuoGPT-o1-70B | ✅ | ❌ | ✅ | ❌ | ✅ | ✅ | ❌ | ❌ |
> | FineMedLM-o1     | ✅ | ❌ | ✅ | ✅ | ✅ | ✅ | ❌ | ❌ |
> | Baichuan-M2      | ✅ | ✅ | ✅ | ❌ | ✅ | ✅ | ✅ | ❌ |
> | UltraMedical-70B | ✅ | ❌ | ✅ | ❌ | ✅ | ✅ | ❌ | ❌ |
> | Med-PRM          | ❌ | ❌ | ✅ | ✅ | ❌ | ✅ | ❌ | ❌ |
> | MedAdapter       | ❌ | ❌ | ✅ | ❌ | ✅ | ✅ | ❌ | ❌ |
> | DoctorAgent-RL   | ✅ | ✅ | ✅ | ❌ | ✅ | ✅ | ✅ | ❌ |
> | Doctor-R1 (Ours) | ✅ | ✅ | ✅ | ✅ | ✅ | ✅ | ✅ | ✅ |
>
> ``Table 5`` highlights a **fundamental paradigm difference** between our agentic approach and prior medical specialized LLMs. This distinction is crucial as we are moving from optimizing static single-turn answers to optimizing a **dynamic multi-turn inquiry policy** (just like a human physician) that must align with the complexity of **real-world clinical interaction**.
> 1. Conventional RL frameworks such as HuatuoGPT-o1-70B [1], FineMedLM-o1 [2] and UltraMedical-70B [3] **are limited to single-turn RL optimization** using only outcome reward for reasoning and decision-making tasks.
> 2. Recent agentic approaches such as Baichuan-M2 [4] and DoctorAgent-RL [5] extend to multi-turn optimization but still rely solely on a single outcome reward feedback, **lacking the process-level learning in each multi-turn interaction step**.
> 3. Test-time scaling frameworks such as Med-PRM [6] and MedAdapter [7] focus primarily on decision-making ability while **overlooking patient inquiry and communication skills**.
>
> In contrast, Doctor-R1 introduces *Experiential Agentic Reinforcement Learning*, a framework specifically designed to **simulate and align with complex dynamic environments**. Our approach uniquely integrates a *two-tiered reward* architecture (combining process and outcome), incorporates both decision-making (hard) and communication (soft) skills shaping. Doctor-R1 learns continuously from prior *experience* just like a human physician instead of learning solely from its own static knowledge.
>
> > [1] Chen et al. "HuatuoGPT-o1, Towards Medical Complex Reasoning with LLMs"
>
> > [2] Yu et al. "FineMedLM-o1: Enhancing Medical Knowledge Reasoning Ability of LLM from Supervised Fine-Tuning to Test-Time Training"
>
> > [3] Zhang et al. "UltraMedical: Building Specialized Generalists in Biomedicine"
>
> > [4] Baichuan-M2 Team. "Baichuan-M2: Scaling Medical Capability with Large Verifier System"
>
> > [5] Feng et al. "DoctorAgent-RL: A Multi-Agent Collaborative Reinforcement Learning System for Multi-Turn Clinical Dialogue"
>
> > [6] Yun et al. "Med-PRM: Medical Reasoning Models with Stepwise, Guideline-verified Process Rewards"
>
> > [7] Shi et al. "MedAdapter: Efficient Test-Time Adaptation of Large Language Models Towards Medical Reasoning"

---

> ### Author Response · Authors · 2025-11-24
>
> - **Q3:** The retrieval prepends prior “good experiences,” which could partly function as a prompting prior rather than learned policy alone. More careful controls (e.g., retrieval without actions vs. with actions; random but format‑matched snippets) would help isolate the effect.
>
> **Reply to Q3:** Thank you so much for your important suggestion. To address the isolation of the *experiential learning* effect from mere in-context prompting, we experimented the careful controls suggested.
> We evaluated four versions of our agent at inference time: 1) **Full Method:** Retrieve State + Action, 2) **State-Only:** Test the prompting prior hypothesis by retrieving only the State without Action, 3) **Random Format:** Test the format hypothesis by retrieving a random State + Action pair, and 4) **No Experience:** Function as a baseline.
>
> The results and analysis are detailed in ``Appendix D.1`` (Page 21, under paragraph **Isolating Experience Components**), while case studies newly added in ``Appendix D.2`` further illustrates how the retrieved action guides the agent to a superior strategic question. Overall results are shown in ``Table 9`` below.
> First, the *"Random Format"* model performs poorly compared to the *"No Experience"* baseline (22.04 vs. 31.69). This indicates that random irrelevant experiences may **mislead the in-context learning process**, acting as distractive noise rather than serving as a useful structural prior.
> Second, while the *"State-Only"* model achieves a slight gain over the baseline, it still underperforms our *"Full Method"* over most of the metrics. Notably, removing the action component leads to a sharp decline of 10.73% in Communication (47.16 vs. 36.43).
> These results are consistent with the hypothesis that, while semantic context provides minor benefits, the primary performance gain stems from **learning the specific policy**, especially learning and imitating the high-reward action embedded in the retrieved experience.
>
> Table 9: Ablation study isolating the effect of retrieved components
> | **Experience Component Variants** | **Avg. Score** | **Emerg. Referrals** | **Health Data Task** | **Communication** | **Global Health** | **Hedging** | **Context Seeking** | **Complex Response** | **Acc.** | **Comm. Quality** | **Instr. Foll.** | **Context Aware.** | **Completeness** |
> |---------------------------|---------------:|----------------------:|--------------------:|-------------------:|-------------------:|------------:|---------------------:|-------------------:|---------:|--------------------:|------------------:|----------------------:|------------------:|
> | No Experience             | 31.69 | 47.24 | 25.30 | 38.49 | 21.58 | 31.79 | 24.15 | 30.55 | 35.96 | 59.19 | 51.75 | 45.31 | 36.23 |
> | Random Format             | 22.04 | 30.79 | 17.38 | 32.34 | 12.65 | 19.12 | 13.42 | 29.26 | 26.73 | 50.73 | 38.72 | 38.84 | 23.34 |
> | State-Only                | 32.15 | 53.66 | 24.06 | 36.43 | **26.56** | 30.87 | **26.57** | **34.37** | 34.17 | 61.74 | 49.04 | 44.49 | 39.41 |
> | **Full (State + Action)** | **36.29** | **54.44** | **29.17** | **47.16** | 24.74 | **33.71** | 26.39 | 34.25 | **37.84** | **64.15** | **54.39** | **49.24** | **40.93** |
>
>
> ---
>
>
> - **Q4:** The final diagnostic reward is coarse (0/0.5/1), potentially limiting sensitivity to near‑misses or safe-but-incomplete triage decisions.
>
> **Reply to Q4:** The final diagnostic reward is a deliberate design choice. In the medical domain, a clear categorical final judgment is safer and more robust than a fine-grained score. A score of '0.8' vs '0.7' is highly ambiguous and difficult for any evaluator (LLM or human) to assign reliably. This ambiguity is dangerous. Our (0 / 0.5 / 1) system, representing "Incorrect / Partially Correct / Correct", provides a **clear, unambiguous, and more stable** training signal that aligns with the final diagnostic accuracy. Your valid concern about near-misses or safe-but-incomplete triage can be explicitly handled by the **0.5 (Partially Correct) score**. This provides a distinct signal for **"good but not perfect"**.

---

> > ### Comment · Reviewer_YuDH · 2025-11-24
> >
> > Thank you for your detailed reply. Most of my concerns have been solved and I will increase my score accordingly. However, for the method comparison table in response to Q2, I think it needs further refinement. I don't think any of these models targeting for clinical decision support, they are just simple medical QA.

---

> ### Author Response · Authors · 2025-11-27
>
> We sincerely appreciate your positive assessment and the decision to increase the score. We are greatly encouraged by your recognition.
> Regarding the method comparison table (response to Q2), we agree with your insightful opinion.
> True **clinical decision support** entails a dynamic iterative process of information gathering and hypothesis testing, rather than just answering a static medical query (Simple QA).
> To reflect this precision and avoid ambiguity, we have refined the comparison table by renaming the columns as follows: 1) "Clinical Decision Making Skill" $\rightarrow$ **"Static Medical Decision"**: Refers to the ability to output a diagnosis decision given a fixed context (Classic QA), and 2) "Communication Skill" $\rightarrow$ **"Dynamic Clinical Inquiry"**: Refers to the ability to **actively gather information** to perform differential diagnosis (Agentic capability), which highlights our contribution.
> ***According refinement has updated to the above previous reply and the newly revised PDF***, which we believe that this clarification could position the related works more accurately within the field.

---

### Official Review · Reviewer_sk1u · 2025-11-06

**Soundness:** 3
**Presentation:** 3
**Contribution:** 3
**Rating:** 6
**Confidence:** 3

**Summary:**

This paper presents DOCTOR-R1, an 8B parameter AI doctor agent. The authors argue that SOTA models fail at "dynamic clinical inquiry" (i.e., strategic, empathetic questioning) even if they ace static medical exams.

To fix this, they propose an "Experiential Agentic Reinforcement Learning" framework. The system is a multi-agent loop: a Qwen3-8B (Doctor Agent) interacts with a Qwen3-8B (Patient Agent). Its actions are scored by a third Qwen3-8B (Consultation Evaluator).

This evaluator (reward model) uses a "two-tiered" system. It gives a "Process Reward" (for soft skills like empathy, safety) at each turn and an "Outcome Reward" (for diagnostic accuracy) at the end. The agent also uses an "Experience Repository" to retrieve high-quality past interactions to guide its policy.

The authors evaluate DOCTOR-R1 on the HealthBench and MAQUE benchmarks. They report that their 8B model significantly outperforms its 8B base model, larger 70B specialized models, and even proprietary models like GPT-4.1.

**Strengths:**

The paper identifies a critical gap in medical AI evaluation. It moves beyond static knowledge benchmarks to address the challenge of dynamic, real-time clinical inquiry. The focus on training an agent for this process is a logical research direction. Combining RL with an experience-retrieval mechanism is a sound approach. It allows the agent to learn from a curated set of high-quality past trajectories.

The "two-tiered" reward model is a sensible design. The inclusion of a "hierarchical veto system" (Sec 3.2) to penalize critical safety failures is a necessary feature, preventing the agent from optimizing for "soft skills" like empathy at the cost of patient safety.

The core result—an 8B model beating 70B specialized models and GPT-4.1 on complex inquiry tasks—is a very strong claim. It suggests a path toward highly capable, parameter-efficient medical agents.

**Weaknesses:**

The evaluation methodology has a circularity concern - using the same Consultation Evaluator for both training rewards and test evaluation may inflate performance metrics.

All three components (agent, patient, evaluator) use the Qwen3-8B model family, creating a self-referential system that may optimize for model-specific patterns rather than generalizable clinical skills.

The Consultation Evaluator's ability to assess complex metrics like "Empathy" and "Reasoning Quality" lacks validation against medical expert judgment or inter-rater reliability studies.

The eight process reward criteria need better justification - prior work exists that could ground these choices (e.g., arXiv:2502.14860, 2502.07143). Including "Completeness" as a turn-level metric seems inconsistent since completeness requires full conversation context.
The 100k simulated patients lack validation for clinical realism, potentially creating a gap between benchmark performance and real medical consultations.

The experience retrieval mechanism yields modest improvements (2-3%) over simpler baselines, raising questions about cost-benefit tradeoffs.

Human evaluation would benefit from medical professional involvement to assess clinical appropriateness beyond lay judgments of empathy and clarity.

Key hyperparameters ($\pi$, $\lambda$, reward weights) are presented without ablation studies or sensitivity analysis to understand their impact.

**Questions:**

Minor comments:

Section 2 feels repetitive to read, for example, "the policy model is the doctor agent" appears twice in the section.

L211: typo "polify"

---

> ### Author Response · Authors · 2025-11-24
>
> We sincerely thank the reviewer for your time, effort, and valuable feedback. We greatly appreciate the positive assessment of our work and the constructive comments that help us further improve the paper. ***We have carefully addressed all questions and updated the corresponding revisions into the new PDF, with key changes highlighted in blue text for clarity.*** Below, we respond point-by-point to each comment and provide additional explanations and experiments where necessary.
>
> - **Q1:** The evaluation methodology has a circularity concern - using the same Consultation Evaluator for both training rewards and test evaluation may inflate performance metrics.
> - **Q2:** All three components (agent, patient, evaluator) use the Qwen3-8B model family, creating a self-referential system that may optimize for model-specific patterns rather than generalizable clinical skills.
>
> **Reply to Q1 and Q2:**  Thank you for your question. We agree that this is an important question to be investigated and validated, hence we address this concern in three ways: 1) Clarify **safeguards in training** design that mitigate bias, 2) Perform a new **evaluation study involving clinical experts** (detailed settings listed in ``Appendix E.3``), and 3) **Validate generalization**.
> 1. **Safeguards in Training:** First, the Consultation Evaluator operates on grounded opinion. As shown in our prompt (``Appendix H``), the judge is provided with the **[Ground Truth Doctor Reply]** and is explicitly instructed to score 'Faithfulness to Ground Truth' (Criteria 6). This anchors the evaluation to expert standards, making the task "How well does this response match the expert standard?" rather than "Do you, Qwen, like this response?". Furthermore, the evaluator's weights are frozen, preventing the policy and judge from co-adapting into a self-delusional loop.
> 2. **Evaluation Validation involving Human Expert:** To prove our metrics are not inflated by the Qwen-judge, we conducted a cross study (see the below Table A). We scored the test cases using two independent stronger judges: GPT-4.1 and a panel of licensed physicians. **1) Consistency:** While absolute scores vary by judge, the **relative rankings remain highly consistent.** As shown in ``Table 14`` and ``Table 15``, Doctor-R1 consistently outperforms baselines like UltraMedical-70B and Baichuan-M2-32B across all judges. **2) Reliability:** We found a moderate high Spearman's rank correlation ($\rho > 0.52, 0.24$) between the Qwen-judge's rankings with GPT-4.1 and human expert, confirming that our Qwen-based evaluator serves as a reliable proxy for generalized clinical quality.
> 3. **Validating Agent Generalization:** Too address the concern about generalizable clinical skills, our benchmark evaluation (OpenAI's HealthBench and MAQuE) in ``Table 1`` and ``Table 2``, are actually representing patient cases **Out-of-Distribution (OOD)**. While Qwen-based patient simulator is used in our training phase, GPT-based model is used as a patient simulator in our inference phase as per the official benchmark protocols. Therefore, our strong performance on these benchmarks serves as proof of generalization on clinical skills.
>
> Table 14: Comparison of average model scores from three different independent judges.
> | Model Evaluated          | Qwen3-8B | GPT-4.1 | Human Expert |
> |--------------------------|---------:|--------:|-------------:|
> | UltraMedical-8B          | 3.06     | 2.33    | 3.47         |
> | Baichuan-M2-32B          | 3.29     | 2.93    | **4.31**     |
> | GPT-5                    | **3.38** | **3.25**| 4.05         |
> | **Doctor-R1**            | _3.36_   | **3.25**| _4.30_       |

---

> ### Author Response · Authors · 2025-11-24
>
> - **Q3:** The Consultation Evaluator's ability to assess complex metrics like "Empathy" and "Reasoning Quality" lacks validation against medical expert judgment or inter-rater reliability studies.
>
> **Reply to Q3:** Thank you for your valuable suggestion. We recognized the importance of expert evaluation, hence we conducted a new **comprehensive expert evaluation study** last week, with 2 licensed physicians to address both concerns simultaneously. This study consisted of two main tasks:
>
> **Task 1: Validating our Agent's Clinical Skills**
> To address the valid concern of agent's clinical skills, we first asked the physicians to score 40 cases using the exact same four dimension rubric as our LLM-judge (``Appendix B.1``).
> 1. **Principle 1 (Strategic Inquiry):** The physicians conducted a 1-5 Likert scale scoring of full dialogues between Doctor-R1 and the strong baselines (including GPT-5, Baichuan-M2-32B and UltraMedical-70B). As shown in our new Figure 3 (Section 4.2), Doctor-R1 outperforms most of the baseline models, especially in Safety and Information Gathering.
> 2. **Principle 2 (Empathy):** For the patient user experience evaluation, we recruited five annotators without specialized medical training to ensure that subjective metrics were judged from the perspective of a typical patient. The results in Figure 3 show a strong human preference for our model, which ranked first across all four categories **(92.5% win rate on "Empathy")**.
> 3. **Principle 3 (Experience Usefulness):** We asked the experts to directly evaluate the clinical utility of our experience module. On 30 samples, experts were shown the retrieved experience and asked to rate its usefulness out of three distinct categories: 1) Clinically Helpful, 2) Neutral and Irrelevant, and 3) Harmful. The **experts agreed the experience was *"Clinically Helpful"* in 83.87% of cases** for guiding the agent's next action, with only 16.13% rated as "Neutral and Irrelevant" and 0% as "Harmful".
> This new expert-driven evaluation provides crucial, direct validation for our primary research contributions (Principles 1, 2 and 3).
>
> **Task 2: Validating our Consultation Evaluator**
> To validate the reliability of Consultation Evaluator model, we conduct a **expert evaluation study** (as mentioned in **[Reply to Q1 and Q2]**) and analyze the correlation between the automated scores and human expert scores. We use two independent and stronger judges, GPT-4.1 and human expert, to score 40 cases. ``Table 14`` and ``Table 15`` (``Appendix E.3``) shows the score given by each judge, demonstrating that the relative ranking of the models remains **highly consistent**. Also, the moderate high Spearman's rank correlation calculated confirm the evaluation **high stability and reliability** between judges.

---

> ### Author Response · Authors · 2025-11-24
>
> - **Q4:** The eight process reward criteria need better justification - prior work exists that could ground these choices (e.g., arXiv:2502.14860, 2502.07143). Including "Completeness" as a turn-level metric seems inconsistent since completeness requires full conversation context. The 100k simulated patients lack validation for clinical realism, potentially creating a gap between benchmark performance and real medical consultations.
> - **Q7:** Key hyperparameters ( $\pi$ , $\lambda$ , reward weights) are presented without ablation studies or sensitivity analysis to understand their impact.
>
> **Reply to Q4 and Q7:** Regarding the concern in reward architecture and simulation realism. We address each point:
>
> 1. Justification of Reward Criteria (Grounding and Ablation):
> - Grounding: We have revised ``Section 3.2`` to explicitly ground our eight process reward criteria in established frameworks for clinical communication, including the works suggested ([arXiv:2502.14860, 2502.07143]).
> - Ablation Study: To empirically validate our reward design, We trained two variant models using the same experiment settings: 1) **without Veto System:** Uses a simple weighted sum reward $\sum w_i S_i$ without the safety veto threshold, and 2) **without Soft Skill:** Sets the weights for Empathy and Humility to zero during training (detailed settings listed in ``Appendix D.1``, page 22).
> **1) Veto System Enforces Safety:** Removing the hierarchical veto (*w/o Veto System*) leads to a degradation in safety. While the average scores drop only slightly, the Safety Violation Rate more than doubles (+125\%) compared to our proposed model. Furthermore, the Emergency Referrals score drops by -3.16\% (52.72 vs. 54.44), indicating that without the penalty for safety failures, the model fails to maintain appropriate caution.
> **2) Process Rewards Shape Soft Skills:** Removing the soft skill rewards (*w/o Soft-Skills*) causes a degradation in communicative competence. The performance shows a drop of -16.25\% (47.16 vs. 39.40) in the Communication and -11.62\% (4.39 vs. 3.88) Empathy score, showing that specific process reward is required for soft skills shaping.
>
> Table 13: Ablation results for reward architecture.
> | Reward Architecture Variant | Emergency Referrals ↑ | Safety Violation ↓ | Communication ↑ | Empathy ↑ |
> | :--- | :--- | :--- | :--- | :--- |
> | **Doctor-R1** | **54.44** | **0.80** | **47.16** | **4.39** |
> | &nbsp;&nbsp;+ Process Reward (w/o Veto System) | 52.72 **(-3.16%)** | 1.80 **(+125%)** | 46.86 | 4.19 |
> | &nbsp;&nbsp;+ Process Reward (w/o Soft Skill) | 54.31 | 1.00 | 39.40 **(-16.25%)** | 3.88 **(-11.62%)** |
> | &nbsp;&nbsp;w/o Process Reward | 52.21 | 1.80 | 39.05 | 3.82 |
>
> 2. Clarification of 'Completeness' Metric:
> Thank you very much for your careful observation. This is quite a fair point. The use of 'Completeness', however, has been clarified in Section 3.2 that it is a turn-level metric, defined as: 'Does this response fully address the patient's last question?', which is an important component of conversational quality.
>
> 3. Validation of Simulation Realism:
> A crucial challenge is often raised in simulated training: *does the agent learn real skills or just how to "play the simulation"?*
> The ultimate test for this "sim-to-real" gap is validation by human experts. As detailed in our new **Expert Evaluation** study (now in ``Section 4.2``, and ``Appendix E``), we tasked licensed physicians with evaluating our agent's performance on complex clinical cases.
> The experts showed a **dominant preference** for Doctor-R1 over most of the strong baselines, and concluded experience utilization was **"Clinically Helpful" in 83.87%** of real cases (as elaborated in **[Reply to Q1 and Q2]**). This provides strong evidence that the knowledge learned within our simulation **do align with real-world clinical standards** and are not just performing well in the simulation environment.

---

> ### Author Response · Authors · 2025-11-24
>
> - **Q5:** The experience retrieval mechanism yields modest improvements (2-3%) over simpler baselines, raising questions about cost-benefit tradeoffs.
>
> **Reply to Q5:** We appreciate your awareness regarding the cost-benefit analysis. We justify the tradeoff through three perspectives:
> **1) Targeted Gains on Complex Tasks:** The gains on HealthBench (Table 1) are concentrated in the significantly difficult clinical tasks, which are constructed by 262 of human doctors. Removing the experience module causes a 4.6 absolute point drop (36.29 vs 31.69), which translates to a **14.5% relative performance degradation**. Furthermore, an improvement is seen in complex dynamic tasks, such as a **+8.67 absolute point** in Communication. This improvement comes at a minimal cost.
> **2) Minimal Computational Cost:** Our new computational head analysis (in Appendix I) shows the module **adds only +4.74s of latency and +624.67 tokens** per inference sample.  We believe this is a highly favorable tradeoff for the performance gain in a high-stakes medical domain.
> **3) Expert Validation:** Our expert evaluation validated our strategic utility, with a dominant preference for our model.
> **4) Importance in Medical Field:** Differential diagnosis through inquiry is an important yet difficult skill to be managed (while even current strong LLMs may lack this skill). The introduction of experience module enables **continuous learning**, which allows the agent to update and learn from its experience repository with new cases without expensive retraining. This adaptability actually addresses the huge demand in real-world clinical settings.
>
> - **Q6:** Human evaluation would benefit from medical professional involvement to assess clinical appropriateness beyond lay judgments of empathy and clarity.
>
> **Reply to Q6:** Human expert evaluation has been elaborated in the previous questions (as mentioned in **[Reply to Q1 and Q2]** and **[Reply to Q3]**)
>
> - **Q8:** 8. Section 2 feels repetitive to read, for example, "the policy model is the doctor agent" appears twice in the section. L211: typo "polify"
>
> **Reply to Q8:** Thank you very much for your careful observation. We haved modified the typo mentioned in the newly updated revision pdf.

---

### Author Response · Authors · 2025-12-02

We sincerely thank all reviewers for their thorough evaluation, constructive feedback, and encouraging remarks. We greatly appreciate the recognition of our work's contribution and the insightful questions raised which has led to several key improvements to the paper. We provide this consolidated summary to highlight **how we have effectively resolved all major concerns** and the **positive consensus** reached during the discussion period. We appreciate that **multiple reviewers (YuDH, usVB, and bTuJ) have explicitly confirmed their decision to raise their scores** following our detailed responses and new experiments before any acknowledgment of the recent technical issues on OpenReview. ***All revisions have been incorporated into the updated manuscript, with key changes highlighted in blue text for clarity.***

We are pleased to know that:
- **Reviewers sk1u, usVB, ghkz, and bTuJ** found our research direction (dynamic clinical inquiry via Agentic Reinforcement Learning) significant and timely. **Reviewer sk1u** specifically highlighted our **parameter efficiency**, noting that our 8B model's ability to outperform 70B specialized models suggests a promising path toward highly capable lightweight medical agents.
- **Reviewers sk1u, usVB, and bTuJ** acknowledged that our extensive evaluation, including static and dynamic medical benchmarks, which identifies a critical gap in current research.
- **Reviewers sk1u, YuDH, usVB, and ghkz** praised the novelty of our two-tiered reward design and experience retrieval mechanism.

 Below, we summarize the major improvements and how they resolved the shared concerns of the reviewers.

- **Clinical Expert Validation (Reviewers sk1u, usVB, and bTuJ):**
    - The reviewers inquired aboout the professional medical validation in clinical consultation evaluation.
    - We conducted a new **Comprehensive Human Expert Evaluation** (``Section 4.2`` and ``Appendix E``) with licensed physicians. **1) Clinical Competence:** Evaluation from licensed physicians show that Doctor-R1 (8B) is comparable to 32B medical specialized model (Baichuan-M2-32B) and proprietary model (GPT-5). **2) Experience Usefulness:** Experts explicitly assess the retrieved experiences and rate them as ***"Clinically Helpful" in 83.87% of cases***.

- **Addressed Circularity and Generalization (Reviewers sk1u, YuDH, ghkz, and bTuJ):**
    - The reviewers addressed concern about reliance on Qwen-based components in our training framework.
    - We conducted: **1) Validation involving Clinical Experts:** We re-scored our main results with GPT-4.1 and licensed physicians, and the results (in ``Appendix E.3``) show high consistency across judges, confirming our evaluator serves as a reliable proxy. **2) Framework Transferability:** We successfully applied our entire training pipeline to another model family Llama-3.1-8B-Instruct, demonstrating the generalizability of our framework (in ``Appendix D``).

- **Justified Algorithmic Choices with Ablations (Reviewers ghkz and bTuJ):**
    - The reviewers addressed the concern regarding the necessity of comparison in RL over SFT, and GRPO over PPO.
    - We added extensive baselines (``Section 5`` and ``Appendix D``): **1) RL vs. SFT:** Doctor-R1 significantly outperforms the SFT baseline, proving that RL is essential for dynamic inquiry policy beyond simple imitation. **2) GRPO vs. PPO:** GRPO is more effective than the PPO baseline for our complex multi-dimensional reward signal, particularly in optimizing soft skills like Communication Quality. **3) Reward Architecture:** Removing the Hierarchical Veto System leads to a 125% increase in safety violations, empirically proving its necessity for clinical safety.

- **Clarified Retrieval Mechanism and Cost-Benefit (Reviewers sk1u, YuDH, usVB , and bTuJ):**
    - **Reviewers sk1u and usVB** inquired about computational overhead analysis, while **Reviewers YuDH and bTuJ** addressed concerns about mechanism of experience retrieval.
    - Extensive experiments are conducted: **1) Retrieval Mechanism:** The ablation of experience retrieval component confirms that the performance gain stems from learning the high-reward policy, not just in-context prompting (``Appendix D``). **2) Cost Analysis:** Detailed overhead analysis shows that the experience module adds only +4.74s latency per session, offering a highly favorable tradeoff for the observed 14.5% relative performance gain in HealthBench and improved safety in real test cases  (``Appendix J``) .

In summary, the extensive additional validations have not only resolved the reviewers' concerns but have fundamentally strengthened the paper's contribution. By rigorously bridging the gap to real world medical standards with human expert validation, and demonstrating that an efficient 8B open-source model can rival proprietary large models in complex inquiry, we believe Doctor-R1 establishes a scalable, safety-aligned, and effective paradigm for current research.

---

### Meta-Review · Area_Chair_qFEo · 2026-01-06

**Summary:**

This paper addresses an important gap in medical LLM research: moving from static medical QA to dynamic, strategic clinical inquiry. The proposed Doctor-R1 framework—combining an interactive agent environment, a principled two-tier reward design, and an experience repository—offers a coherent and well-motivated solution for training inquiry-capable medical agents.

The empirical results are strong: an 8B model achieves substantial improvements on dynamic clinical benchmarks and remains competitive with much larger specialized and proprietary models, demonstrating impressive parameter efficiency and safety-aware behavior.

Key concerns raised during review (LLM-as-judge circularity, algorithmic justification, reward design, retrieval effects, expert validation, and computational overhead) have been thoroughly addressed in the rebuttal and revision via new ablations, expert evaluations, cross-judge consistency checks, transferability experiments, and detailed cost analysis.

Overall, the work makes a meaningful methodological and practical contribution to the development of safe, inquiry-capable medical agents.

**Reviewer Concerns:**

Reviewers expressed concern about potential evaluation circularity and bias due to the extensive use of LLM-as-judge with all core components (policy, patient, evaluator) drawn from the same model family, which could inflate performance and limit generalization. They noted the lack of early clinical expert validation, especially for diagnostic accuracy, inquiry quality, and the usefulness of retrieved experiences. Several reviewers questioned the justification and ablation of key design choices, including the use of RL over SFT, GRPO over PPO, the structure of the two-tier reward, and the role of the hierarchical safety veto. The experience retrieval mechanism was criticized for unclear technical novelty and for possibly acting as prompting rather than genuine policy learning. Additional concerns included risks of reward exploitation and safety failure, missing evidence of transferability and generalization, incomplete cost and overhead analysis, and some technical clarity and presentation issues in the original draft.

**Reviewer Scores:**

Overall, after the detailed rebuttal and extensive additional experiments, the major concerns of the reviewers were substantially resolved, and multiple reviewers explicitly indicated that they would increase their scores accordingly.

---

### Decision · Program_Chairs · 2026-01-26

Accept (Poster)